# Decomposed Attention FredFormer: Large Time-series Prediction Model for Satellite Orbit Prediction

## Abstract

Accurate satellite orbit prediction is critical for collision avoidance and sustainable space operations. However, conventional methods are constrained by coarse update intervals, orbit discontinuities, and other factors. Additionally, building separate prediction models for each satellite is computationally expensive, making large-scale accurate forecasting increasingly impractical. To address the aforementioned challenges, we propose Decomposed Attention FredFormer (DAF), a large time-series prediction model that uses efficient Real Fast Fourier Transform (RFFT)/Inverse RFFT in favor of positional embeddings. Our DAF also integrates Tensorized Multi-Head Attention based on Tensor Train Decomposition for parameter-efficient compression and improved performance. We pre-trained on a large-scale Starlink dataset and evaluated zero-shot performance on seven cross-domain satellite orbit datasets and three real-world datasets. DAF achieves up to 34.85% reduction in mean squared error and 16.01% reduction in mean absolute error over the second-best model, using only 0.05% of its parameters and maintaining inference time as fast as the conventional neural network baselines. These results demonstrate that DAF enables zero-shot, high-precision orbit prediction not only for Starlink satellites, but also for other satellites. The code is available here: https://anonymous.4open.science/r/DAF-0D75

## 1 Introduction

The number of operational spacecraft in Earth orbit has increased dramatically over the last decade, now exceeding 11,700 active payloads by March 2025 (France-Presse, 2025). The dominant growth of large constellations primarily drives the increase in the number of satellites. For example, as of August 1, 2025, the Starlink constellation consists of 8,074 active satellites, with the goal of expanding the network into a 'megaconstellation' comprising a total of 42,000 satellites (Pultarova, 2025). The increasing number of satellites imposes a significant mass burden on Earth's orbital environment, leading to the generation of numerous debris fragments caused by collisions, explosions, and degradation in the harsh conditions of space (Boley & Byers, 2021). The Kessler Syndrome (Kessler & Cour-Palais, 1978) describes a scenario in which the growth of space debris not only poses a serious threat to satellite operations, but also increases the likelihood of cascading collisions among space objects. Over time, this results in an exponential increase in the amount of debris. Such space debris poses critical dangers to satellites and the International Space Station (ISS), threatening the long-term sustainability of space activities. A notable example occurred in February 2009, when the Russian satellite Cosmos 2251 collided with the U.S. Iridium 33 satellite, producing a large amount of debris (Iannotta, 2009). As a result, the ISS had to perform a total of five collision avoidance maneuvers over two years, up to early April 2011 (NASA Orbital Debris Program Office, 2011).

To preserve the sustainability of the space environment, many studies have focused on accurately predicting satellite orbits to prevent such collisions (Choumos et al., 2024; Caldas & Soares, 2024b; Reiland et al., 2021; Grau et al., 2025). Current operational orbit prediction typically relies on the propagation of Two-Line Element (TLE) sets using the analytical model Simplified General Perturbation-4 (SGP4) (Hoots & Roehrich, 1980). Although the SGP4 offers computational efficiency, its kilometer-level prediction error can accumulate over long-term predictions, resulting in differences of hundreds of kilometers from the actual orbit (Vallado et al., 2006). Recent machine

learning (ML)-based approaches for satellite orbit prediction have been proposed to address the accumulated prediction errors of the SGP4 model over long-term prediction. While such methods have demonstrated improved performance over using SGP4 alone, their evaluations were limited to a small number and a narrow variety of satellites. As a result, these approaches exhibit limited generalizability and constrained applicability in real-world operational scenarios (Pihlajasalo et al., 2018; San-Juan et al., 2018; Curzi et al., 2022; Jadala et al., 2022). Furthermore, with the rapid increase in the number of satellites, developing prediction models for each satellite becomes prohibitively time-consuming and computationally expensive. Therefore, large time-series models that are trained on massive datasets, can offer the advantage of zero-shot prediction across various domains without the need for individual model training (Das et al., 2024; Jin et al., 2024). However, large models typically require an increase in inference time. This characteristic makes them unsuitable for satellite orbit prediction, where real-time inference and limited hardware resources are critical constraints.

Therefore, there is a growing need for large models to provide improved zero-shot prediction performance for various satellites, while improving computational efficiency and conserving hardware resources. In this work, we aim to design the first large time-series model for satellite orbit prediction. In particular, we propose Decomposed Attention FredFormer (DAF) model, which is pre-trained on a large-scale Starlink dataset and achieves high prediction performance with a small number of parameters, while yielding fast inference speed. The DAF model leverages FredFormer, which has shown a high prediction performance on time-series datasets. Additionally, we utilize Real Fast Fourier Transform (RFFT) layer, while removing the patching and layer normalization. Furthermore, we develop Tensor Train Decomposition (TTD) to the attention weight matrix of FredFormer, reducing model parameters while simultaneously improving performance. Our DAF model demonstrates its effectiveness compared to existing large time-series models (e.g., TimesFM, Time-MoE) as well as general time-series models (e.g., GRU, LSTM). The main contributions of our work are summarized as follows:

- In this work, we propose the novel Decomposed Attention FredFormer (DAF), which is capable of training on large-scale satellite data and achieving high-performance zero-shot evaluation on various satellites. To the best of our knowledge, this research is the first large time-series model proposed in the field of satellite orbit prediction.

- The DAF model enhances prediction performance, while reducing model parameters by replacing FFT to RFFT, adopting positional embedding instead of patching, and applying TTD to Multi-Head Attention.

- We evaluate the generalizability and deployment potential of the DAF model through zero-shot evaluation on seven constellation datasets and three real-world satellite datasets. In addition, we achieve parameter reduction through TTD and demonstrate its effectiveness by comparing inference time with existing large models and conventional neural networks.

## 2 DATASET PROCESSING PIPELINE

In Figure 1 (a), we first illustrate the data collection process used in our study and explain how Orekit (maisonobe et al., 2024) method was applied for interpolation. Next, in Section 2.1, we provide a detailed description of the data collection process used in our study. Finally, in Section 2.2, we describe the interpolation methods and the corresponding validation procedures.

### 2.1 DATASET DESCRIPTION

Table 1 provides an overview of the nine satellite orbit datasets used in our experiments. We collected all TLE datasets (except for KOMPSAT) from Space-Track via its public API (Space-Track, 2025). The real-world datasets (KOMPSAT-3, 3A, and 5) were collected from analytical computation of orbital elements Jeong et al. (2024). Additionally, all datasets span from Communication and Observation to Multipurpose missions. This diversity in temporal resolution, dataset scale, and mission profile enables a comprehensive evaluation of our prediction models across a wide range of environments.

Satellites used in our research complete an orbit in approximately 90-120 minutes. However, TLEs are typically collected by humans only every eight hours, creating significant temporal gaps in the

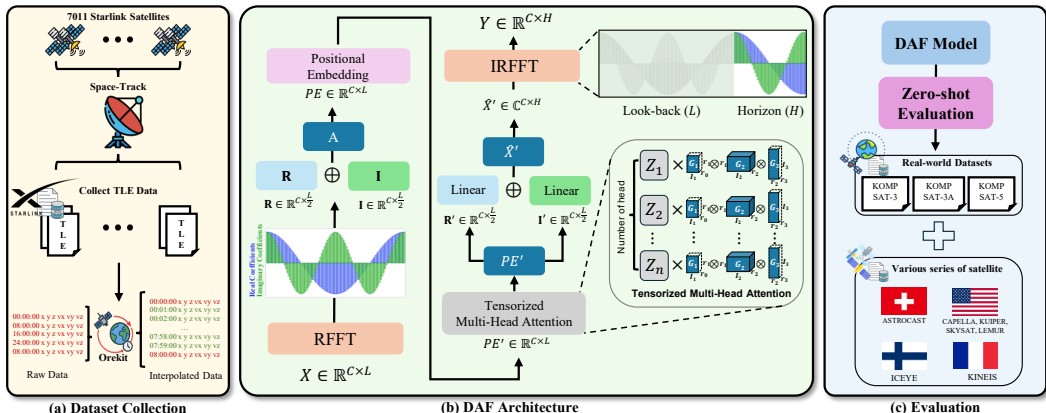

Figure 1: Overall framework of our proposed approach. **(a)** For data preprocessing and interpolation, we collected TLE data using the Space-Track's public API and interpolated them using the Orekit method. **(b)** Input sequences are transformed to the frequency domain via RFFT and concatenated with positional embeddings. The transformed data pass through TMHA, undergo linear projection and IRFFT, and are finally used to predict the next sequence. **(c)** We evaluate zero-shot performance on seven cross-domain satellite orbit datasets and three real-world datasets.

Table 1: Summary of satellite orbit datasets, including name, mission type, number of rows and satellites, observation period, and average/minimum/maximum revisit frequency.

| Dataset | Mission Type | Rows | Satellites | Period | Avg. Freq. | Min. Freq. | Max. Freq. |
|---------|-------------|------|-----------|--------|-----------|-----------|-----------|
| Starlink | Communication | 13,234,406 | 7,011 | 19.11.14–25.02.25 (1,930 d) | 11 h 21 m | 0.864 ms | 786 h 36 m |
| ASTROCAST | Communication | 5,864 | 16 | 25.02.26–25.07.15 (140 d) | 11 h 34 m | 2.592 ms | 85 h 34 m |
| CAPELLA | Observation | 1,562 | 6 | 25.02.26–25.07.15 (140 d) | 12 h 20 m | 12.096 ms | 78 h 50 m |
| ICEYE | Observation | 18,431 | 42 | 25.02.26–25.07.15 (140 d) | 8 h 47 m | 6.048 ms | 83 h 10 m |
| KINEIS | Communication | 12,854 | 25 | 25.02.26–25.07.15 (140 d) | 7 h 46 m | 0.864 ms | 91 h 14 m |
| KUIPER | Communication | 6,865 | 55 | 25.02.26–25.07.15 (140 d) | 9 h 26 m | 82.080 ms | 63 h 34 m |
| LEMUR | Observation | 16,922 | 63 | 25.02.26–25.07.15 (140 d) | 12 h 52 m | 2.592 ms | 114 h 11 m |
| SKYSAT | Observation | 4,852 | 15 | 25.02.26–25.07.15 (140 d) | 12 h 23 m | 1 h 32 m | 109 h 43 m |
| KOMPSAT | Multipurpose | 7,892,646 | 3 | 16.01.01–21.01.01 (1,828 d) | 1 m | 1 m | 1 m |
| Total | | 21,194,402 | 7,236 | 16.01.01-25.07.15 (3,484 d) | | - | |

raw data. Such gaps make it impossible to reflect perturbations such as atmospheric drag and solar radiation pressure, rendering the data unsuitable for training continuous prediction models. To address this challenge, we interpolate between gaps using the propagation method, filling in missing positions and velocities to produce a physically consistent orbit.

## 2.2 DATASET PREPROCESSING AND INTERPOLATION METHOD SELECTION

We model satellite orbits as time-series by integrating ground-station observations and TLE data. To determine the most accurate interpolation method among various satellite orbit interpolation techniques, we conducted a validation procedure. First, we collected ground observation data for 7,216 active Starlink satellites as of April 14, 2025, at one-minute resolution, in a modified ephemeris format from the International Telecommunication Center (ITC). After excluding 337 satellites launched after the Starlink period, which were used in our study, we obtained a final dataset of 6,879 satellites for which both ephemeris and TLE data matched.

Using the TLE datasets corresponding to these 6,879 satellites, we generated interpolated epochs and state vectors ($x$, $y$, $z$, $vx$, $vy$, $vz$) at one-minute intervals using the SGP4 (Hoots & Roehrich, 1980), Skyfield (Rhodes, 2019), and Orekit (maisonobe et al., 2024) methods. The ephemeris dataset served as ground truth (Khairallah & Kassas, 2021) and was used to evaluate interpolation error by comparing interpolated vectors to the ground truth using MSE, MAE, mean difference, and standard deviation. Before conducting the error analysis, we validated the statistical significance between the datasets using the Friedman test (Friedman, 1937) and Nemenyi post-hoc test (Nemenyi, 1963).

We used the Friedman test to examine whether there were statistically significant differences across the three datasets generated by the SGP4, Skyfield, and Orekit methods. When the Friedman test indicated significance, the Nemenyi post-hoc test was applied to perform pairwise comparisons and identify which interpolation pairs showed significant differences. As shown in Table 2, the Friedman

Table 2: Statistical analysis results validating the significance of interpolation performance across SGP4, Skyfield, and Orekit. The table reports results for four metrics across the *x, y*, and *z* features, including Friedman test statistics and *p*-values, Nemenyi post-hoc *p*-values for pairwise comparisons, and the corresponding mean values for each method. The Nemenyi test was only applied to features with statistically significant Friedman test results. The best result for each metric is highlighted in **bold**.

| Metric | Feat. | Friedman test | | Nemenyi test (*p*-val.) | | | Mean Values | | |
|---|---|---|---|---|---|---|---|---|---|
| | | stat. | *p*-val. | SGP4&Skyfield | SGP4&Orekit | Orekit&Skyfield | SGP4 | Skyfield | Orekit |
| MSE | *x* | 8448.9772 | < 0.05 | < 0.05 | < 0.05 | < 0.05 | 10645.732 | 9816.147 | **9816.146** |
| | *y* | 9174.1494 | < 0.05 | < 0.05 | < 0.05 | < 0.05 | 8133.012 | 7369.783 | **7369.780** |
| | *z* | 3947.891 | < 0.05 | < 0.05 | < 0.05 | 0.9 | 10259.165 | **10174.8805** | 10174.8813 |
| MAE | *x* | 8497.8197 | < 0.05 | < 0.05 | < 0.05 | < 0.05 | 31.88368 | 13.587937 | **13.587854** |
| | *y* | 9230.8524 | < 0.05 | < 0.05 | < 0.05 | < 0.05 | 30.527394 | 13.181023 | **13.180938** |
| | *z* | 4552.0648 | < 0.05 | < 0.05 | < 0.05 | 0.9 | 18.179287 | 13.355324 | **13.355318** |
| Mean Diff. | *x* | 396.8254 | < 0.05 | 0.9 | < 0.05 | < 0.05 | -0.038654 | -0.022224 | **-0.022223** |
| | *y* | 30.4268 | < 0.05 | 0.9 | < 0.05 | < 0.05 | -0.002987 | **-0.000447** | -0.000447 |
| | *z* | 4.7382 | 0.093 | - | - | - | 0.090636 | **0.091607** | 0.091607 |
| Mean Std. | *x* | 8448.9772 | < 0.05 | < 0.05 | < 0.05 | < 0.05 | 37.83216 | 18.310809 | **18.310721** |
| | *y* | 9174.1494 | < 0.05 | < 0.05 | < 0.05 | < 0.05 | 36.089137 | 17.615347 | **17.615256** |
| | *z* | 3947.891 | < 0.05 | < 0.05 | < 0.05 | 0.9 | 22.715372 | 17.910431 | **17.909425** |

test revealed statistically significant differences ($p < 0.05$) across all metrics except for the mean difference *z* feature. The Nemenyi post-hoc test showed that Orekit generally outperformed the others, whereas comparisons between SGP4 and Skyfield were often not significant. Accordingly, we adopted Orekit with one-minute interpolation for all subsequent experiments. A detailed discussion on the choice of interpolation method is provided in Appendix B.

For pre-training, we used 6,955 satellites. The test set consists of 56 satellites, with one randomly selected per launch month. We split the remaining satellites into training and validation sets in an 8:2 ratio; to maintain distributional balance, the validation set was sampled at regular intervals. For normalization, we computed the mean and standard deviation from the pre-training training split and applied the same z-score normalization to the training, validation, and test sets.

# 3 METHOD

## 3.1 DECOMPOSED ATTENTION FREDFORMER (DAF)

In our research, based on the Frequency Debiased Transformer model (FredFormer) (Piao et al., 2024) as the backbone, we propose the Decomposed Attention FredFormer (DAF) model with several improvements. First, we replace FredFormer's DFT layer with a RFFT layer. Next, given the periodic nature of satellite orbits, we remove patching and add positional embeddings so that the attention mechanism can account for the input sequence order. Furthermore, since the inputs are z-scaled, we remove layer normalization. In particular, we replace FredFormer's attention with Tensorized Multi-Head Attention (TMHA) based on TTD. Lastly, the inverse DFT (IDFT) layer was replaced with an inverse RFFT (IRFFT) layer. The overall architecture of the DAF model is shown in Figure 1 (b).

## 3.2 RATIONALES OF APPLYING RFFT AND IRFFT

First, FredFormer employed the DFT to convert discrete-time signals from the time domain to the complex frequency domain. However, when the input data is real-valued, as in our study, the Hermitian symmetry property allows the use of the RFFT, which preserves the same information as the full DFT while representing it in $\frac{L}{2} + 1$ complex components (Zhou et al., 2022; Xu et al., 2023). Therefore, we discard both the DFT and IDFT, and adopt the RFFT–IRFFT pair in our architecture. In particular, we apply an RFFT to the input $X \in \mathbb{R}^{C \times L}$, transforming it into the frequency domain:

$$\widehat{X} = RFFT(X) \in \mathbb{C}^{C \times (\frac{L}{2} + 1)}, \tag{1}$$

where $C$ denotes the number of input channels, and $L$ denotes the historical look-back window. Subsequently, we separate the real part $R \in \mathbb{R}^{C \times \frac{L}{2}}$ and the imaginary part $I \in \mathbb{R}^{C \times \frac{L}{2}}$ of $\widehat{X}$, and then stack them to obtain $A \in \mathbb{R}^{C \times L}$, which is passed through a positional embedding layer to produce $PE \in \mathbb{R}^{C \times L}$ and then fed into the TMHA layer.

---

**Algorithm 1** Algorithm of Decomposed Attention FredFormer (DAF). $C$ denotes the number of channels, $L$ the look-back window length, $H$ the prediction horizon, $\mathcal{H}$ the number of heads, $D$ the weight matrix dimension, and $\mathbf{r}$ the TT ranks. $\mathcal{T}_h(\cdot)$ refers the Tensorized Multi-Head Attention.

---

1: **Input:** Historical data $X \in \mathbb{R}^{C \times L}$
2: **Output:** prediction result $Y \in \mathbb{R}^{C \times H}$
3: **(i) RFFT–IRFFT Backbone**
4: $\widehat{X} \leftarrow \mathrm{RFFT}(X) \in \mathbb{C}^{C \times (L/2)}$
5: $R \leftarrow \mathrm{Re}(\widehat{X}) \in \mathbb{R}^{C \times (L/2)}, \quad I \leftarrow \mathrm{Im}(\widehat{X}) \in \mathbb{R}^{C \times (L/2)}$
6: $A \leftarrow Concat(R, I) \in \mathbb{R}^{C \times L}$
7: $PE \leftarrow A + Positional\ Embedding$
8: **(ii) Tensorized Multi-Head Attention**
9: $W \leftarrow \mathrm{stack}(W_Q, W_K, W_V) \in \mathbb{R}^{3 \times D \times D}$
10: $(G_1, G_2, G_3) \leftarrow \mathrm{TTD}(W, \mathbf{r})$
11: **for** $h = 1$ to $\mathcal{H}$ **do**
12: $\quad (Q_h, K_h, V_h) \leftarrow \mathcal{T}_h(Z, G_1, G_2, G_3)$
13: $\quad$ **Conduct scaled dot-product attention**
14: **end for**
15: **(iii) Frequency Summarization**
16: $(R' \in \mathbb{R}^{C \times \frac{L}{2}}, I' \in \mathbb{R}^{C \times \frac{L}{2}}) \leftarrow \mathrm{Split}(PE') \in \mathbb{R}^{C \times L}$
17: $\widehat{X}' \leftarrow R'W_R + i\,I'W_I \in \mathbb{C}^{C \times H}, \quad W_R, W_I \in \mathbb{R}^{(L/2) \times H}$
18: $Y \leftarrow \mathrm{IRFFT}(\widehat{X}') \in \mathbb{R}^{C \times H}$
19: **return** $Y$

---

Subsequently, after passing through the TMHA layer, the resulting $PE' \in \mathbb{R}^{C \times L}$ is again separated into the real component $\mathrm{R}' \in \mathbb{R}^{C \times \frac{L}{2}}$ and imaginary component $\mathrm{I}' \in \mathbb{R}^{C \times \frac{L}{2}}$. Each component passes through a linear layer $W_R = W_I \in \mathbb{R}^{\frac{L}{2} \times H}$ to obtain:

$$\widehat{X}' = \mathrm{R}'W_R + i\mathrm{I}'W_I \in \mathbb{C}^{C \times H}. \tag{2}$$

Here, $H$ denotes the prediction horizon. Finally, by applying an IRFFT to transform the data back from the frequency domain to the time domain, we obtain the final output $Y \in \mathbb{R}^{C \times H}$.

## 3.3 POSITION-AWARE LEARNING

Initially, the FredFormer applies patching to $\mathrm{A} \in \mathbb{R}^{C \times L}$, performs layer normalization, and then feeds the result into the attention layer. Also, FredFormer adopts patching to enable the model to learn complex temporal patterns at both global and local scales. However, in contrast, satellite orbital data exhibit strong periodicity and a relatively simple structure, thereby reducing the importance of local patterns. Therefore, we remove patching and instead add a learnable positional embedding before passing the input to the attention layer. Additionally, since the validation and test inputs are z-scored using the global mean and standard deviation estimated from 5,564 pre-training series, we omit layer normalization to avoid double normalization. Finally, the input to the attention is defined as follows:

$$Z = A + PE \in \mathbb{R}^{C \times L}. \tag{3}$$

## 3.4 TENSORIZED MULTI-HEAD ATTENTION

Similar to matrix decomposition methods such as Singular Value Decomposition (SVD) (Klema & Laub, 1980), Tensor Decomposition decomposes a high-dimensional tensor into smaller core tensors. There are various types of Tensor Decomposition; in our study, we adopt the Tensor Train Decomposition (TTD) (Oseledets, 2011), which is known to be efficient for model compression, helping reduce the number of parameters while maintaining, or even improving, the model's performance. For an $d$-dimensional tensor $\mathcal{X} \in \mathbb{R}^{i_1 \times i_2 \times \cdots \times i_d}$, where $i_k$ denotes the size of dimension $k$, the formula for TTD is given as follows:

$$\mathcal{X}(i_1, i_2, \ldots, i_d) \approx G_1(:, i_1, :)G_2(:, i_2, :)\cdots G_d(:, i_d, :), \tag{4}$$

where $G_k \in \mathbb{R}^{r_{k-1} \times i_k \times r_k}$. Here, in order to keep the result of all matrix multiplications as a scalar, $r_0 = r_d = 1$ and $\{r_k\}_{k=0}^d$ refers to TT-ranks. In the conventional Attention, given the input $Z$ and the weight matrices $W_Q \in \mathbb{R}^{D \times D}, W_K \in \mathbb{R}^{D \times D}$, and $W_V \in \mathbb{R}^{D \times D}$, we obtain $Q = ZW_Q, K = ZW_K$, and $V = ZW_V$. To apply TTD to the Attention, which refers to Tensorized Attention (TA), we stack the weight matrices $W_Q, W_K$, and $W_V$ to form $W \in \mathbb{R}^{3 \times D \times D}$. Then, applying Equation 4 with the TT ranks $\{r_k\}_{k=0}^4$, we obtain $G_1 \in \mathbb{R}^{1 \times 3 \times r_1}, G_2 \in \mathbb{R}^{r_1 \times D \times r_2}$, and $G_3 \in \mathbb{R}^{r_2 \times D \times 1}$. In TA, new $Q', K'$, and $V'$ are generated from the input $Z$ by using $G_1, G_2$, and $G_3$ instead of $W_Q, W_K$, and $W_V$, and the final output is obtained by performing scaled-dot product attention. The equations for TA are as follows.

$$\text{TA}(Z, G_1, G_2, G_3) = \text{softmax}\left(\frac{Q'K'}{\sqrt{d_h}}\right)V', \quad s.t.$$

$$[Q', K', V']_{ndj} = \sum_{a=1}^{D}\sum_{\alpha=1}^{r_1}\sum_{\beta=1}^{r_2} Z_{na}\, G_1(1, j, \alpha)\, G_2(\alpha, a, \beta)\, G_3(\beta, d, 1), \tag{5}$$

where $a, d \in \{1, \ldots, D\}$ are weight dimension index, and $n \in \{1, \ldots, N\}$ is the sequence index, $j \in \{1, 2, 3\}$ selects $Q', K', V', \alpha \in \{1, \ldots, r_1\}, \beta \in \{1, \ldots, r_2\}$ are the middle ranks of TT rank $\{r_0, r_1, r_2, r_3\}$ and $d_h$ denotes attention key dimension per number of heads.

Extending Tensorized Attention to TMHA, we compose the heads multiplicatively and compute in batched form, resulting in a high-dimensional configuration. We denote this high-dimensional tensor as $Z \in \mathbb{R}^{3 \times B \times C \times D}$. For the operations of TMHA, we divide the second dimension of $Z \in \mathbb{R}^{3 \times B \times C \times D}$ proportionally to the number of heads $h$, resulting in a tensor to be restructured to $Z \in \mathbb{R}^{3 \times \frac{B}{h} \times h \times C \times D}$, where $B$ represents the batch size, $C$ denotes the dimension of the input data. The resulting $Z$, along with $G_1, G_2, G_3$, is calculated using the following Equation (6):

$$\text{TMHA}(Z, G_1, G_2, G_3) = \text{Concat}_{i=1}^{\text{head}}(\text{TA}_i(Z_i, G_1, G_2, G_3)), \tag{6}$$

where $\text{TA}_i$ denotes the $i$-th head's TA, as defined in Equation 5. The output from the TMHA layer is passed through separate linear layers for the real and imaginary parts to generate the corresponding predictions, which are then fed into the IRFFT layer for decoding.

## 4 EXPERIMENTS

In this section, we describe the experimental setup and results for 14 models, including our model. Section 4.1 outlines the experimental settings for both state-of-the-art large time-series prediction models and the commonly used baseline approach. Section 4.2 presents results on the Starlink dataset for nine fine-tuned large time-series baselines and six pre-trained models (including ours). As shown in Figure 1 (c), Section 4.3 reports zero-shot evaluation on seven cross-domain datasets and three real-world datasets. Section 4.4 provides an ablation study on 11 datasets (including Starlink) for FredFormer and its three variants, along with an analysis of parameter reduction achieved by applying the tensor decompositions. Finally, Section 4.4 investigates the effect of PE in the frequency domain.

### 4.1 EXPERIMENTS SETUP AND BASELINE

To evaluate the practicality of the proposed method, we conduct a comparative analysis against 14 baselines using the Starlink test set and 10 cross-domain datasets. The cross-domain setting is used to measure zero-shot performance, and we also consider real-world applicability using three KOMPSAT datasets (3, 3A, 5). Evaluation is based on MAE and MSE metrics. In the space domain, an error as small as 0.001 can translate to a physical deviation of several to tens of kilometers; therefore, we report our results to the sixth decimal place for a more precise comparison. All tests

are conducted on an NVIDIA GeForce RTX 3090 GPU under identical conditions to measure model inference time for orbit prediction.

We use the five conventional time-series prediction models and nine large time-series models as baselines. The conventional models include GRU (Dey & Salem, 2017), LSTM (Hochreiter & Schmidhuber, 1997), DLinear (Zeng et al., 2023), SparseTSF (Lin et al., 2024), and Fred-Former (Piao et al., 2024). We trained these models from scratch on the pre-training dataset. The large time-series models include Time-MoE, Timer, Timer-XL, Time-LLM, TTM, MOMENT, MOIRAI, and TimesFM. For fine-tuning, we used 1/100 of the pre-training dataset. The look-back window is $L = 512$, and the prediction horizon is $H = 90$. We adopt a sliding-window approach with a stride of $L + H$ to ensure non-overlapping segments. All models are trained for 100 epochs, starting from the optimal hyperparameters reported in the respective papers, which were further tuned to ensure loss convergence on the Starlink dataset. For univariate models, we report the average performance across the six features, while inference time is measured as the total over all six. We perform all tests under the same settings and conditions. Detailed experimental configurations are provided in the Appendix D.

## 4.2 Main Result

Table 3 reports the mean error over three runs on the Starlink dataset. The DAF model achieved the best performance across both MSE and MAE. TimesFM ranked second in both metrics. However, our model achieved relative improvements of $34.85\%$ in MSE and $16.01\%$ in MAE compared to TimesFM, while being approximately $2,221\times$ smaller in parameter count. In contrast, although SparseTSF uses only 94 parameters, it recorded an MSE of $0.004167$, which is significantly worse than that of our model.

Table 3: Prediction results on Starlink dataset in MSE and MAE of each model. The rightmost column shows the total parameter count of each model, and the bottom row "Improvements" highlights the performance gain of our model over the best baseline for each metric. The best result is highlighted in **bold**, and the second-best is highlighted with underline.

| Metric | MSE | MAE | Parameter |
|---|---|---|---|
| Time-MoE | $2.265333 \pm 0.026844$ | $1.263860 \pm 0.009000$ | 113,352,192 |
| Timer | $0.327132 \pm 0.004084$ | $0.484511 \pm 0.003660$ | 67,397,728 |
| Timer-XL | $0.081020 \pm 0.050731$ | $0.220013 \pm 0.096210$ | 67,398,880 |
| Time-LLM | $1.025519 \pm 0.077263$ | $0.876721 \pm 0.046888$ | 105,321,802 |
| TTM | $0.499960 \pm 0.378112$ | $0.555809 \pm 0.302664$ | 16,344,891 |
| TimesFM | $\underline{0.001506 \pm 0.000157}$ | $\underline{0.028408 \pm 0.001259}$ | 498,828,960 |
| MOMENT | $0.010566 \pm 0.000856$ | $0.086805 \pm 0.003665$ | 16,085,594 |
| MOIRAI | $0.027327 \pm 0.017312$ | $0.118248 \pm 0.045438$ | 15,855,786 |
| GRU | $0.002177 \pm 0.000162$ | $0.033817 \pm 0.000262$ | 121,884 |
| LSTM | $0.003582 \pm 0.001138$ | $0.043888 \pm 0.007628$ | 139,292 |
| DLinear | $0.002516 \pm 0.000032$ | $0.036839 \pm 0.000260$ | 554,040 |
| SparseTSF | $0.004167 \pm 0.000014$ | $0.052828 \pm 0.000070$ | 94 |
| FredFormer | $0.034392 \pm 0.002362$ | $0.163769 \pm 0.005493$ | 290,194 |
| **DAF (Ours)** | $\mathbf{0.000981 \pm 0.000030}$ | $\mathbf{0.023853 \pm 0.000072}$ | 224,556 |
| Improvements (%) | $34.85 \pm 3.79$ | $16.01 \pm 4.44$ | - |

## 4.3 Zero-shot Evaluation

Furthermore, Table 4 presents the zero-shot performance of the models trained on the Starlink dataset when evaluated on seven cross-domain satellite datasets. The proposed model delivers the best results on most datasets despite their disparate specifications and operational purposes. It also achieves a $62.81\%$ improvement on the LEMUR dataset. Finally, Table 5 presents a further comparison on three high-precision datasets with analytically computed orbits rather than via interpolation. Despite being real-world datasets, the proposed model achieves the lowest prediction errors in nearly every case.

## 4.4 Ablation Study

In this section, we evaluate the effectiveness of the two core structures of the DAF model. The core structure of the DAF model can be grouped into two components: (1) a layer variant that replaces DFT with RFFT and removes patching and layer normalization, and (2) applying TTD to the Multi-Head Attention. Additionally, we consider a third case that applies Canonical Polyadic Decomposition (CPD) (Hitchcock, 1924) in place of TTD. The explanation of CPD and its application to TMHA is provided in Appendix C. Across 11 datasets, including Starlink, the experimental results are reported in Table 6, where we denote the four configurations as: FredFormer, FredFormer (layer variant), FredFormer (layer variant + CPD), and Fred-Former (layer variant + TTD, DAF). Our DAF model achieved the best performance in 6 out of 11

Table 4: Zero-shot evaluation on seven cross-domain datasets in MSE and MAE of each model. The bottom row "Improvements" highlights the performance gain of our model over the best baseline for each dataset and metric. The best result is highlighted in **bold**, and the second-best is highlighted with underline.

| Dataset | ASTROCAST | | CAPELLA | | ICEYE | | KINEIS | | KUIPER | | LEMUR | | SKYSAT | |
|---|---|---|---|---|---|---|---|---|---|---|---|---|---|---|
| Metric | MSE | MAE | MSE | MAE | MSE | MAE | MSE | MAE | MSE | MAE | MSE | MAE | MSE | MAE |
| Time-MoE | 2.209372 | 1.171837 | 2.295843 | 1.268301 | 2.251507 | 1.195950 | 2.384254 | 1.242035 | 2.213490 | 1.246838 | 2.259241 | 1.187898 | 2.237188 | 1.191708 |
| Timer | 0.090851 | 0.217582 | 0.535211 | 0.600972 | 0.285239 | 0.397307 | 0.941015 | 0.808324 | 0.109496 | 0.272394 | 0.295066 | 0.417342 | 0.157977 | 0.303088 |
| Timer-XL | 0.036722 | 0.139096 | 0.110913 | 0.252742 | 0.071666 | 0.192768 | 0.177071 | 0.321643 | 0.044190 | 0.159048 | 0.075421 | 0.198898 | 0.051863 | 0.168050 |
| Time-LLM | 0.964805 | 0.803355 | 1.061852 | 0.886094 | 1.016547 | 0.830448 | 1.148139 | 0.883287 | 0.963177 | 0.851150 | 1.023576 | 0.826542 | 0.990677 | 0.821108 |
| TTM | 0.225226 | 0.347945 | 0.642272 | 0.614609 | 0.430893 | 0.476646 | 0.951737 | 0.752327 | 0.260675 | 0.404975 | 0.454790 | 0.493409 | 0.321774 | 0.428811 |
| TimesFM | 0.002258 | 0.028439 | 0.006887 | 0.062037 | 0.002702 | 0.036138 | 0.024786 | 0.124216 | 0.004377 | 0.031227 | 0.002977 | 0.033964 | 0.002506 | 0.030870 |
| MOMENT | 0.012305 | 0.085201 | 0.010939 | 0.085381 | 0.011772 | 0.084199 | 0.016214 | 0.093539 | 0.016863 | 0.093891 | 0.012093 | 0.083998 | 0.012131 | 0.086694 |
| MOIRAI | 0.011396 | 0.068974 | 0.042769 | 0.144607 | 0.025507 | 0.103988 | 0.078260 | 0.200425 | 0.014616 | 0.080221 | 0.026792 | 0.107371 | 0.017238 | 0.087685 |
| GRU | 0.002624 | 0.038205 | 0.004105 | 0.046738 | 0.003493 | 0.042391 | 0.010210 | 0.074563 | 0.001242 | 0.022845 | 0.003085 | 0.038363 | 0.001888 | 0.032268 |
| LSTM | 0.002476 | 0.036840 | 0.006965 | 0.061469 | 0.004873 | 0.049932 | 0.016452 | 0.096129 | 0.001765 | 0.028110 | 0.004761 | 0.049704 | 0.002855 | 0.038525 |
| DLinear | 0.000448 | 0.013369 | 0.007117 | 0.060827 | 0.002692 | 0.031739 | 0.018088 | 0.099608 | 0.000520 | **0.010687** | 0.002533 | 0.030934 | 0.001126 | 0.016291 |
| SparseTSF | 0.000597 | 0.017675 | 0.013467 | 0.092643 | 0.004657 | 0.046133 | 0.033884 | 0.152013 | 0.000553 | 0.012730 | 0.004201 | 0.043359 | 0.001798 | 0.020233 |
| FredFormer | 0.035184 | 0.156715 | 0.031599 | 0.155111 | 0.034384 | 0.155599 | 0.030640 | 0.144051 | 0.035653 | 0.016794 | 0.034760 | 0.155006 | 0.036094 | 0.159925 |
| **DAF (Ours)** | **0.000302** | **0.011767** | **0.002860** | **0.037055** | **0.001004** | **0.021340** | **0.009251** | **0.066928** | **0.000474** | 0.012720 | **0.000942** | **0.020974** | **0.000628** | **0.015339** |
| Improvements (%) | 32.59 | 11.97 | 59.81 | 39.08 | 62.70 | 32.76 | 9.40 | 10.22 | 8.85 | -19.00 | 62.81 | 32.20 | 44.25 | 5.84 |

Table 6: Ablation Study in MSE and MAE for four variants of FredFormer on 11 datasets, including Starlink, cross-domain, and real-world datasets. The bottom row shows the number of datasets on which each variant achieved the best performance. The best result is highlighted in **bold**, and the second-best is highlighted with underline.

| Model | FredFormer | | FredFormer (Layer Variant) | | FredFormer (Layer Variant + CPD) | | FredFormer (Layer Variant + TTD, DAF) | |
|---|---|---|---|---|---|---|---|---|
| Metric | MSE | MAE | MSE | MAE | MSE | MAE | MSE | MAE |
| Starlink | 0.034392 | 0.163769 | 0.001061 | 0.025258 | 0.001013 | 0.024449 | **0.000981** | **0.023853** |
| K3 | 0.032467 | 0.147525 | 0.014338 | 0.089235 | **0.010207** | **0.071267** | 0.014248 | 0.084121 |
| K3A | 0.035877 | 0.158858 | 0.000748 | 0.021877 | **0.000714** | **0.020978** | 0.000738 | 0.020709 |
| K5 | 0.034810 | 0.156403 | 0.001203 | 0.027767 | 0.001078 | 0.025591 | **0.001042** | **0.024979** |
| ASTROCAST | 0.035184 | 0.156715 | 0.000340 | 0.012795 | 0.000343 | 0.012819 | **0.000302** | **0.011767** |
| CAPELLA | 0.031599 | 0.155111 | 0.002685 | 0.038391 | **0.002288** | **0.034798** | 0.002860 | 0.037055 |
| ICEYE | 0.034384 | 0.155599 | 0.001184 | 0.023629 | 0.001043 | 0.022069 | **0.001004** | **0.021340** |
| KINEIS | 0.030640 | 0.144051 | 0.008261 | 0.068009 | **0.006156** | **0.055699** | 0.009251 | 0.066928 |
| KUIPER | 0.035653 | 0.016794 | 0.000487 | 0.012890 | 0.000527 | 0.014434 | **0.000474** | **0.012720** |
| LEMUR | 0.034760 | 0.155006 | 0.001111 | 0.023197 | 0.000981 | 0.021694 | **0.000942** | **0.020974** |
| SKYSAT | 0.036094 | 0.159925 | 0.000633 | 0.015860 | **0.000605** | **0.016552** | 0.000628 | **0.015339** |
| 1st Count | 0 | 0 | 0 | 0 | 5 | 3 | 6 | 8 |

datasets based on MSE and in 8 datasets based on MAE. Furthermore, except for the CAPELLA dataset, it consistently ranked second-best even when it did not achieve the top performance.

The FredFormer (Layer Variant + CPD) model was the only model, besides DAF, to achieve the best performance in at least one case. Specifically, it achieved the best performance across 5 datasets based on MSE and across 3 datasets based on MAE. These results demonstrate that modifying the layers of the FredFormer model and applying TTD to the MHA significantly enhances performance. The results in Table 7 show the parameter counts for FredFormer and its three variants. FredFormer has the most parameters, totaling 290,194.

Table 5: Zero-shot evaluation on three real-world datasets in MSE and MAE of each model. "Improvements" presents the relative performance of our model compared to the best baseline for each dataset and metric. The best result is highlighted in **bold**, and the second-best is highlighted with underline.

| Dataset | KOMPSAT-3 | | KOMPSAT-3A | | KOMPSAT-5 | |
|---|---|---|---|---|---|---|
| | MSE | MAE | MSE | MAE | MSE | MAE |
| Time-MoE | 2.418150 | 1.250692 | 2.276393 | 1.198948 | 2.296285 | 1.207763 |
| Timer | 1.180190 | 0.905896 | 0.292992 | 0.447280 | 0.390626 | 0.517673 |
| Timer-XL | 0.211756 | 0.350651 | 0.078914 | 0.211551 | 0.094826 | 0.232320 |
| Time-LLM | 1.179187 | 0.894232 | 1.034300 | 0.835688 | 1.056063 | 0.845372 |
| TTM | 1.061232 | 0.797521 | 0.493810 | 0.537106 | 0.582112 | 0.583277 |
| TimesFM | 0.044738 | 0.169868 | 0.001129 | 0.024734 | 0.001127 | **0.024651** |
| MOMENT | 0.023447 | 0.116151 | 0.010463 | 0.083579 | 0.010273 | 0.081888 |
| MOIRAI | 0.098327 | 0.228442 | 0.026376 | 0.113609 | 0.033509 | 0.128145 |
| GRU | **0.014227** | 0.089150 | 0.001693 | 0.030145 | 0.002543 | 0.036848 |
| LSTM | 0.020632 | 0.107698 | 0.003804 | 0.046117 | 0.005183 | 0.053863 |
| DLinear | 0.027910 | 0.124509 | 0.001662 | 0.029771 | 0.003101 | 0.040915 |
| SparseTSF | 0.053629 | 0.191637 | 0.002394 | 0.040606 | 0.005025 | 0.058457 |
| FredFormer | 0.032467 | 0.147525 | 0.035877 | 0.158858 | 0.034810 | 0.156403 |
| **DAF (Ours)** | 0.014248 | **0.084121** | **0.000738** | **0.020709** | **0.001042** | 0.024979 |
| Improvements (%) | -0.15 | 5.64 | 34.63 | 16.27 | 7.54 | -1.33 |

The FredFormer (Layer Variant), which applies RFFT, halves the size of the linear layer after the TMHA layer, then applies IRFFT to R and I, resulting in 224,992 parameters. Our DAF model further reduces the parameter count to 224,556 by applying TTD to FredFormer (Layer Variant), demonstrating improved performance with fewer parameters. Meanwhile, FredFormer (Layer Vari-

Table 7: Comparison of parameter counts across FredFormer and its variants.

| Model | Parameter |
|---|---|
| FredFormer | 290,194 |
| FredFormer (Layer Variant) | 224,992 |
| FredFormer (Layer Variant + CPD) | 193,324 |
| FredFormer (Layer Variant + TTD, DAF (Ours)) | 224,556 |

Table 9: Total inference time (in seconds) on 11 datasets, including cross-domain and real-world datasets. Additionally, the computational cost is provided in GFLOPs (per sequence) calculated with a batch size of 16 for each model. The lowest values (fastest time and lowest computational cost) are highlighted in **bold**, and the second-lowest are highlighted with underline.

| Dataset | Inference Time (s) | | | | | | | | | | | GFLOPs |
|---|---|---|---|---|---|---|---|---|---|---|---|---|
| | Starlink | ASTROCAST | CAPELLA | ICEYE | KINEIS | KUIPER | LEMUR | SKYSAT | KOMPSAT-3 | KOMPSAT-3A | KOMPSAT-5 | |
| Time-MoE | 9496.95 | 402.97 | 123.69 | 972.14 | 634.40 | 458.96 | 1373.46 | 389.80 | 345.77 | 343.28 | 338.78 | 5353.2000 |
| Timer | 30.24 | 1.79 | 0.89 | 3.75 | 2.57 | 2.00 | 4.97 | 1.77 | 1.65 | 1.66 | 1.63 | 25.1553 |
| Timer-XL | 40.13 | 2.21 | 1.01 | 4.78 | 3.26 | 2.46 | 6.39 | 2.16 | 1.97 | 2.00 | 1.98 | 25.1557 |
| Time-LLM | 671.55 | 30.03 | 9.55 | 73.41 | 47.07 | 33.85 | 100.38 | 28.79 | 25.75 | 25.63 | 25.79 | 954.4704 |
| TTM | 149.38 | 6.54 | 2.14 | 15.82 | 10.25 | 7.43 | 22.07 | 6.38 | 5.65 | 5.63 | 5.65 | 62.1760 |
| TimesFM | 756.06 | 35.76 | 13.62 | 83.76 | 54.40 | 39.90 | 116.59 | 35.04 | 31.04 | 31.18 | 31.04 | 517.2000 |
| MOMENT | 13.64 | 1.27 | 0.78 | 2.05 | 1.56 | 1.30 | 2.70 | 1.15 | 1.05 | 1.12 | 1.10 | 7.0992 |
| MOIRAI | 16.05 | 1.27 | 0.60 | 2.02 | 1.52 | 1.25 | 2.73 | 1.08 | 1.03 | 1.01 | 0.98 | 7.1008 |
| GRU | 3.86 | 0.23 | 0.04 | 0.38 | **0.02** | 0.17 | 0.53 | 0.18 | 0.12 | 0.12 | 0.12 | 0.4368 |
| LSTM | **3.84** | **0.15** | 0.04 | 0.41 | 0.24 | 0.16 | 0.58 | 0.15 | 0.12 | 0.12 | 0.12 | 0.5824 |
| DLinear | 5.12 | 0.45 | 0.04 | 0.38 | 0.25 | 0.17 | 0.64 | 0.15 | 0.14 | 0.13 | 0.12 | 0.0096 |
| SparseTSF | 4.62 | 0.64 | **0.03** | **0.31** | 0.29 | **0.14** | **0.42** | **0.12** | **0.11** | **0.11** | **0.11** | **0.0016** |
| FredFormer | 5.45 | 0.46 | 0.05 | 0.47 | 0.31 | 0.23 | 0.65 | 0.15 | 0.14 | 0.14 | 0.14 | 0.0240 |
| **DAF (Ours)** | 3.86 | 0.16 | 0.05 | 0.39 | 0.24 | 0.17 | 0.55 | 0.14 | 0.12 | 0.17 | 0.13 | 0.0144 |

ant + CPD) exhibits even fewer parameters than DAF and achieves better performance than Fred-Former (Layer Variant). However, since FredFormer (Layer Variant + CPD) does not outperform the DAF model, the DAF model is more suitable for satellite large time-series prediction tasks.

## 4.5 EFFECT OF POSITIONAL EMBEDDING IN FREQUENCY DOMAIN

To address the role of PE as a replacement for patching and the effectiveness of frequency-domain representation, we conduct experiments comparing three variants: (1) DAF-timePE with time-domain input and PE (DAF (no RFFT)), (2) DAF-noPE with RFFT and original patching (DAF (no PE)), and (3) DAF with RFFT and PE (DAF (ours)).

The results demonstrate that DAF (ours) reduces MSE by 3.1% compared to DAF (no PE) while using 3,328 fewer parameters, validating that PE is a more parameter-efficient alternative to patching. Furthermore, DAF (ours) outperforms DAF (no RFFT) by 23.5%, confirming that the frequency-domain representation enabled by RFFT is effective for accurate orbit prediction. The slight parameter increase of DAF (ours) over DAF (no RFFT) (256 parameters) arises from the RFFT output dimension: an input sequence of length $N$ produces $N/2 + 1$ complex frequency components, which are represented as $N + 2$ real values after concatenating real and imaginary parts. In summary, DAF (ours) achieves the best performance with comparable parameter count, validating that the combination of RFFT and PE is efficient for satellite orbit prediction.

Table 8: Ablation study on PE and frequency transformation

| Model Variant | MSE | MAE | Parameter |
|---|---|---|---|
| DAF (no RFFT) | 0.001282 | 0.027066 | 224,300 |
| DAF (no PE) | 0.001012 | 0.024515 | 227,884 |
| DAF (ours) | **0.000981** | **0.023853** | 224,556 |

## 5 DISCUSSION AND FUTURE WORK

Our research shows that the proposed model DAF, trained solely on the Starlink dataset, outperforms existing large time-series models. As shown in Table 9, it delivers faster inference through a smaller parameter count and achieves GFLOPs per sequence comparable to lightweight models, making it highly efficient and readily deployable. DAF's consistent performance across diverse satellites confirms both its robustness and suitability for real mission operation.

However, we acknowledge specific scenarios where the model may face challenges. First, regarding sudden orbital maneuvers, the model may struggle to predict orbit changes resulting from collision avoidance maneuvers if such dynamic events are absent from the training data. Second, extreme environmental variations can impact performance. Severe fluctuations in the space environment, such as rapid changes in atmospheric density caused by intense solar activity, can create out-of-distribution scenarios that may increase prediction errors.

For future work, we aim to address these limitations and move beyond the proposed large model to develop a foundation model for satellite orbit prediction that generalizes across Low, Medium, and Geostationary Earth Orbits. This model can be pre-trained on the large-scale Starlink dataset, followed by transfer learning to accurately predict orbits across satellites with different orbital regimes

and mission types, without additional supervision. Furthermore, we plan to extend the model to support uncertainty quantification. In mission-critical operations such as collision avoidance, estimating the reliability of a predicted position is as important as the prediction itself. To address this, we will explore methods to transition from deterministic to probabilistic forecasting, such as modifying the output layer to estimate Gaussian distribution parameters (mean and variance) using negative log-likelihood loss or employing quantile loss to provide confidence intervals. Finally, we will validate the model's practical applicability by conducting experiments on a broader set of high-precision, real-world satellite orbit datasets. Moreover, we aim to expand the domain scope and evolve the proposed framework into a large-scale model applicable to general time-series prediction tasks beyond orbital dynamics. Given the model's capability to capture inherent periodicity and long-range dependencies via frequency-domain analysis, we anticipate its applicability to other quasi-periodic domains, such as meteorology, energy consumption, and signal processing.

## 6 CONCLUSION

In this work, we propose the Decomposed Attention FredFormer (DAF) model, a large time-series prediction model for satellite orbit prediction. DAF incorporates Tensorized Multi-Head Attention and RFFT for better frequency modeling and removes layer normalization and positional embedding to replace patching. Our design choices yield improved accuracy and efficiency, while significantly reducing model size. To validate our design, we train the model solely on the large-scale Starlink dataset and conduct comparative evaluations, including zero-shot prediction, on eleven satellite orbit datasets with varying specifications. Our DAF achieves the best performance by a large margin in most experiments. Despite being pre-trained as a large model on high-volume satellite data, our method achieves the broad generalizability across diverse orbital regimes with an optimally compressed parameter space, resulting in a compact and deployable architecture. We believe our work paves the way for demonstrating the potential of large models in satellite orbit prediction.

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

## GenAI Usage Disclosure

Text rephrasing and grammar checks were performed using GPT-5 and Grammarly. Code debugging was assisted by GPT-5. All outputs were reviewed and modified by the authors. All ideas and content were originally written and verified by the authors.

## A  Related Work

### A.1  Orbit Prediction using Machine Learning

Traditional orbit prediction processes are based on physics-based propagators, such as SGP4, which integrate orbital equations under simplifying assumptions. Although effective for short-term predictions, these methods degrade over time due to unmodeled environmental forces and imprecise initial conditions. ML-based approaches have emerged to correct these deficiencies by learning error patterns from historical data (Caldas & Soares, 2024a; Kazemi et al., 2024). One common strategy is hybrid error correction, in which a model learns the deviation of a physics-based prediction from ground truth. Peng and Bai applied SVR to correct SGP4 errors and demonstrated 20–30% reductions in prediction error (Peng & Bai, 2018). They later extended this framework to Gaussian processes and neural networks (Peng & Bai, 2019), and combined Gaussian process error fixes into a Kalman filter (Peng & Bai, 2021).

Temporal models such as RNNs and LSTMs have also been used to predict. Zhou et al. embedded an LSTM within an orbit determination process to maneuver satellites and demonstrated improved robustness in long-horizon predictions (Zhou et al., 2023). Latent force models treat unmodeled accelerations as Gaussian process priors, combining Newtonian mechanics with statistical inference (Hartikainen & Särkkä, 2010). More recently, neural operators and differentiable programming have been proposed to learn orbital dynamics directly from partial differential equations (Chen et al., 2024). Although ML-based approaches to orbit prediction have made progress, open challenges remain in generalizing models across orbital regimes and ensuring robustness with limited training data.

### A.2  Large Time-series Model

In time-series, large models have recently been actively proposed to learn general time representations through large-scale pre-training and apply them to various domains in a zero-shot manner.

**TimesFM** (Das et al., 2024) adopts a decoder-only Transformer architecture and is pre-trained on 100 billion time points. It is evaluated across heterogeneous benchmarks such as energy, finance, and weather without fine-tuning. **MOIRAI** (Woo et al., 2024) is a masked encoder-based universal time-series prediction transformer that addresses universal prediction challenges through multiple patch-size projection layers, Any-variate Attention, and a flexible mixture-distribution output layer. **TinyTimeMixers (TTM)** (Ekambaram et al., 2024) is an extremely lightweight model based on the MLP-based TSMixer architecture. It handles multivariate time-series through Adaptive Patching and Multi-Resolution Sampling. **MOMENT** (Goswami et al., 2024) builds a large-scale public time-series dataset called the "Time-Series Pile" and proposes a task-agnostic foundation model applicable to classification, prediction, and anomaly detection. **Timer-XL** (Liu et al., 2025) combines a time-series self-attention mechanism ("TimeAttention") with a Kronecker-based mask to efficiently process long-context sequences. It supports univariate, multivariate, and 1D/2D inputs within a single architecture. **Time-MoE** (Shi et al., 2025) introduces a sparse Mixture-of-Experts (MoE) structure into the Transformer to enable token-level specialization. It is scalable up to 2.4 billion parameters and is trained on the Time-300B dataset (300 billion time points). **Time-LLM** (Jin et al., 2024) reprograms time-series as text prototypes using existing large language models (LLMs) and applies the Prompt-as-Prefix method without modifying the base architecture.

While all of these models demonstrate strong capabilities in zero-shot learning across various time-series tasks, their large model sizes make fine-tuning computationally demanding. Moreover, it remains challenging to achieve high performance on specific datasets consistently.

## B    FULL RESULTS FOR INTERPOLATION METHOD SELECTION

Table 10: Statistical analysis results for validating the statistical significance of the interpolated dataset and mean values for different features. The table includes Friedman test statistics and $p$-values for MSE, MAE, mean difference, and mean standard deviation across the $x, y, z, vx, vy$, and $vz$ features, Nemenyi post-hoc $p$-values for pairwise comparisons among SGP4, Skyfield, and Orekit methods, and the corresponding mean values for each method. Nemenyi test was not performed for features that were not statistically significant in the Friedman test. The best result is highlighted in **bold**.

| Metric | Feat. | Friedman test | | Nemenyi test ($p$-val.) | | | Mean Values | | |
|---|---|---|---|---|---|---|---|---|---|
| | | stat. | $p$-val. | SGP4&Skyfield | SGP4&Orekit | Orekit&Skyfield | SGP4 | Skyfield | Orekit |
| MSE | $x$ | 8448.9772 | < 0.05 | < 0.05 | < 0.05 | < 0.05 | 10645.732 | 9816.147 | **9816.146** |
| | $y$ | 9174.1494 | < 0.05 | < 0.05 | < 0.05 | < 0.05 | 8133.012 | 7369.783 | **7369.780** |
| | $z$ | 3947.891 | < 0.05 | < 0.05 | < 0.05 | 0.9 | 10259.165 | **10174.8805** | 10174.8813 |
| | $vx$ | 8365.3877 | < 0.05 | < 0.05 | < 0.05 | < 0.05 | 0.014363 | **0.013367** | 0.013367 |
| | $vy$ | 9018.8088 | < 0.05 | < 0.05 | < 0.05 | < 0.05 | 0.010629 | **0.009711** | 0.009711 |
| | $vz$ | 4071.345 | < 0.05 | < 0.05 | < 0.05 | 0.9 | 0.013902 | **0.013801** | 0.013801 |
| MAE | $x$ | 8497.8197 | < 0.05 | < 0.05 | < 0.05 | < 0.05 | 31.88368 | 13.587937 | **13.587854** |
| | $y$ | 9230.8524 | < 0.05 | < 0.05 | < 0.05 | < 0.05 | 30.527394 | 13.181023 | **13.180938** |
| | $z$ | 4552.0648 | < 0.05 | < 0.05 | < 0.05 | 0.9 | 18.179287 | 13.355324 | **13.355318** |
| | $vx$ | 8521.2458 | < 0.05 | < 0.05 | < 0.05 | < 0.05 | 0.035155 | **0.015061** | 0.015061 |
| | $vy$ | 9140.794 | < 0.05 | < 0.05 | < 0.05 | < 0.05 | 0.033603 | **0.014575** | 0.014575 |
| | $vz$ | 4654.464 | < 0.05 | < 0.05 | < 0.05 | 0.880152 | 0.020182 | **0.014809** | 0.014809 |
| Mean Diff. | $x$ | 396.8254 | < 0.05 | 0.9 | < 0.05 | < 0.05 | -0.038654 | -0.022224 | **-0.022223** |
| | $y$ | 30.4268 | < 0.05 | 0.9 | < 0.05 | < 0.05 | -0.002987 | **-0.000447** | -0.000447 |
| | $z$ | 4.7382 | 0.093 | - | - | - | 0.090636 | **0.091607** | 0.091607 |
| | $vx$ | 2.9315 | 0.231 | - | - | - | 0.000001 | **0.000000** | **0.000000** |
| | $vy$ | 0.3282 | 0.849 | - | - | - | **-0.000005** | -0.000005 | -0.000005 |
| | $vz$ | 0.0038 | 0.998 | - | - | - | **-0.000003** | -0.000003 | -0.000003 |
| Mean Std. | $x$ | 8448.9772 | < 0.05 | < 0.05 | < 0.05 | < 0.05 | 37.83216 | 18.310809 | **18.310721** |
| | $y$ | 9174.1494 | < 0.05 | < 0.05 | < 0.05 | < 0.05 | 36.089137 | 17.615347 | **17.615256** |
| | $z$ | 3947.891 | < 0.05 | < 0.05 | < 0.05 | 0.9 | 22.715372 | 17.910431 | **17.909425** |
| | $vx$ | 8365.3877 | < 0.05 | < 0.05 | < 0.05 | < 0.05 | 0.041777 | **0.020314** | 0.020314 |
| | $vy$ | 9019.5578 | < 0.05 | < 0.05 | < 0.05 | < 0.05 | 0.039778 | **0.019489** | 0.019489 |
| | $vz$ | 4071.345 | < 0.05 | < 0.05 | < 0.05 | 0.9 | 0.02525 | **0.019863** | 0.019863 |

We present a detailed performance comparison of the three interpolation methods (SGP4, Skyfield, and Orekit) for the six features ($x, y, z, vx, vy$, and $vz$), as shown in Table 10. For each satellite, we calculated the MSE, MAE, mean difference, and mean standard deviation between the ground truth and the interpolated results. These values were then averaged across all satellites to obtain **Mean Values** for each method. Statistical significance tests follow the same analysis methods described in Section 2.2.

First, in the case of error-based metrics (MSE, MAE), there was a statistically significant difference between the three methods in all features. In the Nemenyi test, SGP4 showed a significant difference with Skyfield and Orekit in all features, and there was no difference between Skyfield and Orekit except for the features $z$ and $vz$. In the mean value, Orekit was the lowest for position features, and Skyfield and Orekit recorded the lowest value for velocity features. These results indicate that Skyfield and Orekit perform better than SGP4.

Second, the mean difference showed statistically significant differences only in the features $x$ and $y$. In these features, Skyfield and Orekit outperform SGP4, and there was no significant difference in other features.

Third, the mean standard deviation significantly outperformed SGP4 in Skyfield and Orekit in all features. In particular, the standard deviation of the two techniques in velocity features is half that of SGP4, indicating that the prediction stability is significantly superior to that of SGP4.

In summary, Skyfield and Orekit overwhelmed SGP4 in all metrics. Although the performance gap between Skyfield and Orekit is insignificant, Orekit was slightly lower in MSE and MAE. Therefore, we selected Orekit as our final interpolation method. Building on this result, the final interpolated Starlink dataset comprises approximately 7.3 billion rows (7,295,420,002). Using this dataset, we created the training, validation, and test splits as described in Section 2.2 and used them to train and evaluate the large time-series prediction models.

## C  EFFECTIVENESS OF TENSORIZED MULTI-HEAD ATTENTION

In our study, we adopted tensor decomposition for TMHA. However, as discussed in Section 4.4, we examine the effectiveness of TMHA by applying CPD instead of TTD. Accordingly, we describe CPD in Section C.1, and explain the TMHA method using CPD in Section C.2.

### C.1  CANONICAL POLYADIC DECOMPOSITION (CPD)

CPD is a method for decomposing a tensor into a sum of rank-1 tensors, each represented by the outer product of factor vectors, which are organized as columns in factor matrices. The formulation of the CPD for an N-way tensor is given in Equation 7:

$$\mathcal{X}(i_1, i_2, \ldots, i_N) \approx \sum_{r=1}^{R} A^{(1)}(i_1, r) A^{(2)}(i_2, r) \cdots A^{(N)}(i_N, r). \tag{7}$$

For instance, when CPD with rank $R$ is applied to a 3-way tensor $\mathcal{X} \in \mathbb{R}^{i_1 \times i_2 \times i_3}$, obtains factor matrix $A^{(1)} \in \mathbb{R}^{i_1 \times R}, A^{(2)} \in \mathbb{R}^{i_2 \times R}$, and $A^{(3)} \in \mathbb{R}^{i_3 \times R}$. Here, $R$, referred to as the CP rank, indicates how many rank-1 tensors are required to approximate the original tensor. In our work, CPD is applied to the weight matrices for the $Q, K$, and $V$ of MHA, denoted as $W_Q \in \mathbb{R}^{D \times D}, W_K \in \mathbb{R}^{D \times D}$, and $W_V \in \mathbb{R}^{D \times D}$, respectively. Here, $D$ denotes the dimension of each weight matrix. These weight matrices are stacked to form a 3-dimensional tensor, $W \in \mathbb{R}^{3 \times D \times D}$. We then apply Equatin 7 to $W$, resulting in the following factor matrices: $A^{(1)} \in \mathbb{R}^{3 \times R}$, $A^{(2)} \in \mathbb{R}^{D \times R}$, and $A^{(3)} \in \mathbb{R}^{D \times R}$.

### C.2  TENSORIZED MULTI-HEAD ATTENTION USING CPD

Similar to applying TTD to TMHA, we apply CPD to the conventional Attention weight matrices, yielding factor matrices $A^{(1)}, A^{(2)}$, and $A^{(3)}$. Subsequently, we apply Equation 8 to the input $Z$, and then compute the final TA value using Equation 8.

$$\text{TA}(Z, A^{(1)}, A^{(2)}, A^{(3)}) = \text{softmax}\left(\frac{Q'K'}{\sqrt{d_h}}\right) V', \quad s.t.$$

$$[Q', K', V']_{ndj} = \sum_{a=1}^{D} Z_{na} W(j, a, d)$$

$$\approx \sum_{a=1}^{D} Z_{na} \sum_{r=1}^{R} A^{(1)}(j, r) A^{(2)}(a, r) A^{(3)}(d, r) \tag{8}$$

where $a, d \in \{1, \ldots, D\}$ are weight dimension index, and $n \in \{1, \ldots, N\}$ is the sequence index, $j \in \{1, 2, 3\}$ selects $Q', K', V'$, $r \in \{1, \ldots, R\}$ is the rank of CPD and $d_h$ denotes attention key dimension per number of heads.

$$\text{TMHA}(Z, A^{(1)}, A^{(2)}, A^{(3)}) = \text{Concat}_{i=1}^{\text{head}}(\text{TA}_i(Z_i, A^{(1)}, A^{(2)}, A^{(3)})). \tag{9}$$

The process of extending TA with CPD to TMHA is similar to the method used for extending TA with TTD. We first reshape the high-dimensional tensor $Z \in \mathbb{R}^{3 \times B \times C \times D}$ by splitting the second dimension into multiple heads, resulting in $Z \in \mathbb{R}^{3 \times \frac{B}{h} \times h \times C \times D}$. Then, we apply Equation 9 to obtain the final TMHA value. Here, $\text{TA}_i$ denotes the $i$-th head's TA, as defined in Equation 8.

## D  EXPLANATION OF DETAIL EXPERIMENTS SETUP

As described in Section 4.1, we fine-tuned large time-series models on 58,488,461 training rows and 15,012,604 validation rows from the Starlink dataset. Unless otherwise noted, models use the

Adam optimizer and MSE loss (TimesFM uses Quantile Loss). A CosineAnnealingLR scheduler with $T_{\max} = 10$ and $\eta_{\min} = 10^{-8}$ is applied to MOIRAI, Time-LLM, MOMENT, TTM, Timer, and Timer-XL. The remaining hyperparameters are:

- **DAF (Ours)**: learning rate $1 \times 10^{-3}$; number of layers 2; number of heads 4; model dimension 128; hidden size 128; Tensor Train Rank [1, 3, 32, 1].
- **FredFormer**: learning rate $1 \times 10^{-3}$; number of layers 2; number of heads 4; model dimension 128; patch length 16; hidden size 128.
- **GRU, LSTM**: learning rate $1 \times 10^{-3}$; hidden size 128; number of layers 1.
- **DLinear**: learning rate $1 \times 10^{-3}$.
- **SparseTSF**: learning rate $1 \times 10^{-3}$; period length 32, dimension of model 3; model type 'mlp'.
- **Time-MoE**: learning rate $1 \times 10^{-4}$; hidden size 4,096; intermediate size 22,016; 32 transformer layers; 32 attention heads; 32 key-value heads; RoPE $\theta = 10,000$; number of experts 1; experts per token 2; aux loss (router) 0.02; max position embeddings 32,768.
- **TimesFM**: learning rate $1 \times 10^{-4}$; weight decay 0.01; batch size (train/inference) 200/20,000; 50 transformer layers; hidden size 1,024; intermediate size 2,816; 8 attention heads; 8 key-value heads; head dimension 128; patch length 128; quantiles [0.1, 0.5, 0.9]; RMSNorm $\epsilon = 10^{-6}$; tolerance $10^{-6}$.
- **MOIRAI**: learning rate $3 \times 10^{-4}$; batch size 256; dropout 0.1; 5 encoder layers; model dimension 512; 8 attention heads; feedforward dimension 2,048; node num 100; node list [23, 37, 40].
- **Time-LLM**: learning rate $1 \times 10^{-5}$; batch size 64; dropout 0.2; model dimension 2; feedforward dimension 8; 4 attention heads; 2 layers; dimension 768; TS vocab size 1,000; patch number 54; head NF 3,456; top-$k$ 5.
- **MOMENT**: learning rate $1 \times 10^{-4}$; batch size 128; dropout 0.1; 5 encoder layers; model dimension 512; 8 attention heads; feedforward dimension 2,048; patch length/stride/padding 90; head NF 3,072.
- **TTM**: learning rate $1 \times 10^{-4}$; batch size 128; dropout 0.2; 3 encoder and 2 decoder layers; model dimension 1,024; hidden dimension 64; patch size 64; number of patches 8.
- **Timer** and **Timer-XL**: learning rate $5 \times 10^{-6}$; batch size 2,048; dropout 0.1; 8 encoder layers; model dimension 1,024; 8 attention heads; feedforward dimension 2,048.

# E    DETAILED ZERO-SHOT RESULTS WITH STANDARD DEVIATIONS

Table 11 presents the MSE and MAE performance of the baselines and DAF for seven cross-domain datasets and three real-world datasets. We repeated the experiment three times and reported the results **mean ± standard deviation.** We describe the experiments on individual satellites for each constellation in Appendix F.

DAF recorded the best performance on 9 MSE and 8 MAE indicators of 10 datasets. In particular, DAF showed the best performance in KOMPSAT-3A, ASTROCAST, CAPELA, IC-EYE, KINEIS, LEMUR, and SKYSAT datasets in both MSE and MAE, and showed overall good performance. For MSE, DAF recorded 34.63±11.98% in KOMPSAT-3A, 7.54±13.36% in KOMPSAT-5, 32.59±1.49% in ASTROCAST, 59.81±0.93% in CAPELLA, 62.70±1.44% in ICEYE, 9.40±29.06% in KINEIS, 8.85±0.72% in KUIPER, 62.81±1.55% in LEMUR, and 44.25±1.63% in SKYSAT for relative improvement. For MAE, DAF shows the improvement 5.64±10.14% in KOMPSAT-3, 16.28±5.90% in KOMPSAT-3A, 11.97±2.01% in ASTRO-CAST, 39.08±0.54% in CAPELLA, 32.76±0.81% in ICEYE, and 10.22±15.57% in KINEIS, 32.20±0.88% in LEMUR, and 5.84±1.14% in SKTSAT.

On the KOMPSAT-3, KOMPSAT-5, and KUIPER datasets, other models performed best in some indicators, but DAF also showed the second-best performance in those indicators. Overall, DAF recorded the highest performance in most datasets, and the performance gap is also numerically clear. The standard deviation also showed a very low value in most cases, confirming stable results in terms of prediction consistency.

# F  ZERO-SHOT EVALUATION ON INDIVIDUAL SATELLITES

The datasets used in Table 11, excluding KOMPSAT dataset (3, 3A, and 5), are satellite constellations, where multiple satellites are grouped into a single network. Some of the constellations consist of dozens of individual satellites, while others do not. Accordingly, we conducted experiments on each satellite within these constellations and evaluated how many satellites our model performed best on.

Tables 12 to 21 present the experimental results for individual satellites excluding Starlink and KOMPSAT, as listed in Table 1. Among them, one satellite from each of the LEMUR and IC-EYE constellations was excluded because of an insufficient number of data samples, which were not even enough to construct a single input sequence. As a result, we conducted experiments on a total of 220 individual satellites. We repeated the experiment three times and reported the results as **mean ± standard deviation**. For "Improvements," we used a '-' when our model did not achieve the best or second-best performance.

Our DAF model achieved the best performance on 142 satellites in terms of MSE and on 135 satellites in terms of MAE. In contrast, the DLinear model achieved the best performance on 67 satellites for MSE and 79 satellites for MAE. The SparseTSF model showed the best performance on 6 satellites for MSE and 5 satellites for MAE. Lastly, the GRU model achieved the best performance on 5 satellites in terms of MSE, and none in terms of MAE. These results indicate that our model outperformed DLinear, which was the second-best model in terms of the number of satellites where it performed best, by 52.82% more satellites for MSE and 41.48% more satellites for MAE.

Table 11: Zero-shot evaluation on seven cross-domain datasets and three real-world datasets with mean and standard deviation in MSE and MAE. "Improvements" presents the relative performance of our model compared to the best baseline for each dataset and metric. The best result is highlighted in **bold**, and the second best is highlighted with underline.

| Dataset | KOMPSAT-3 | | KOMPSAT-3A | | KOMPSAT-5 | | ASTROCAST | | CAPELLA | |
|---|---|---|---|---|---|---|---|---|---|---|
| Metric | MSE | MAE | MSE | MAE | MSE | MAE | MSE | MAE | MSE | MAE |
| Time-MoE | 2.418150±0.059595 | 1.250692±0.020702 | 2.276393±0.028283 | 1.198948±0.010379 | 2.296285±0.026046 | 1.207763±0.011188 | 2.209372±0.023402 | 1.171837±0.007552 | 2.295843±0.032649 | 1.268301±0.012375 |
| Timer | 1.180190±0.014982 | 0.905896±0.005751 | 0.292992±0.004309 | 0.447280±0.003614 | 0.390626±0.004545 | 0.517673±0.003462 | 0.090851±0.002890 | 0.217582±0.004626 | 0.535211±0.004983 | 0.600972±0.003417 |
| Timer-XL | 0.211756±0.131739 | 0.350651±0.148071 | 0.078914±0.046482 | 0.211551±0.084767 | 0.094826±0.056760 | 0.232320±0.094856 | 0.036722±0.018414 | 0.139096±0.044310 | 0.110913±0.069572 | 0.252742±0.108593 |
| Time-LLM | 1.179187±0.137077 | 0.894232±0.040137 | 1.034300±0.085953 | 0.835688±0.047337 | 1.056063±0.052572 | 0.845372±0.032717 | 0.964805±0.186308 | 0.803355±0.095774 | 1.061852±0.022265 | 0.886094±0.023674 |
| TTM | 1.061232±0.776646 | 0.797521±0.403180 | 0.493810±0.372450 | 0.537106±0.290132 | 0.582112±0.439469 | 0.583277±0.315377 | 0.225226±0.165130 | 0.347945±0.177865 | 0.642272±0.483496 | 0.614609±0.329821 |
| TimesFM | 0.044738±0.002202 | 0.169868±0.004287 | 0.001129±0.000132 | 0.024734±0.001425 | 0.001127±0.000125 | **0.024651±0.001463** | 0.002258±0.000142 | 0.028439±0.001176 | 0.006887±0.000456 | 0.062037±0.001598 |
| MOMENT | 0.023447±0.004235 | 0.116151±0.011141 | 0.010463±0.000850 | 0.083579±0.003531 | 0.010273±0.000929 | 0.081888±0.003755 | 0.012305±0.000581 | 0.085201±0.002481 | 0.010939±0.001053 | 0.085381±0.003757 |
| MOIRAI | 0.098327±0.043900 | 0.228442±0.064739 | 0.026376±0.017098 | 0.113609±0.044194 | 0.033509±0.020832 | 0.128145±0.048832 | 0.011396±0.007114 | 0.068974±0.027276 | 0.042769±0.024198 | 0.144607±0.052026 |
| GRU | **0.014227±0.001405** | 0.089150±0.004511 | 0.001693±0.000413 | 0.030145±0.004764 | 0.002543±0.000613 | 0.036848±0.005442 | 0.002624±0.000394 | 0.038205±0.002786 | 0.004105±0.000522 | 0.046738±0.002307 |
| LSTM | 0.020632±0.001919 | 0.107698±0.005494 | 0.003804±0.000676 | 0.046117±0.005220 | 0.005183±0.000846 | 0.053863±0.005502 | 0.002476±0.000700 | 0.036840±0.006638 | 0.006965±0.001123 | 0.061469±0.005579 |
| DLinear | 0.027910±0.000475 | 0.124509±0.001049 | 0.001662±0.000023 | 0.029771±0.000217 | 0.003101±0.000045 | 0.040915±0.000308 | 0.000448±0.000004 | 0.013369±0.000067 | 0.007117±0.000105 | 0.060827±0.000458 |
| SparseTSF | 0.053629±0.000155 | 0.191637±0.000193 | 0.002394±0.000110 | 0.040606±0.000069 | 0.005025±0.000017 | 0.058457±0.000071 | 0.000597±0.000003 | 0.017675±0.000077 | 0.013467±0.000036 | 0.092643±0.000086 |
| FredFormer | 0.032467±0.001674 | 0.147525±0.003916 | 0.035877±0.002281 | 0.158858±0.004587 | 0.034810±0.001981 | 0.156403±0.003958 | 0.035184±0.001633 | 0.156715±0.003151 | 0.031599±0.000873 | 0.155111±0.001986 |
| DAF (Ours) | 0.014248±0.001202 | **0.084121±0.004787** | **0.000738±0.000049** | **0.020709±0.000266** | **0.001042±0.000035** | 0.024979±0.000098 | **0.000302±0.000004** | **0.011767±0.000210** | **0.002860±0.000024** | **0.037055±0.000051** |
| Improvements | -0.15±18.34 | 5.64±10.14 | 34.63±11.98 | 16.28±5.90 | 7.54±13.36 | -1.33±6.41 | 32.59±1.49 | 11.97±2.01 | 59.81±0.93 | 39.08±0.54 |

| Dataset | ICEYE | | KINEIS | | KUIPER | | LEMUR | | SKYSAT | |
|---|---|---|---|---|---|---|---|---|---|---|
| Metric | MSE | MAE | MSE | MAE | MSE | MAE | MSE | MAE | MSE | MAE |
| Time-MoE | 2.251507±0.024534 | 1.195950±0.008941 | 2.384254±0.057027 | 1.242035±0.020190 | 2.213490±0.020493 | 1.246838±0.004917 | 2.259241±0.026369 | 1.187898±0.009867 | 2.237188±0.024447 | 1.191708±0.008350 |
| Timer | 0.285239±0.003321 | 0.397307±0.003440 | 0.941015±0.010247 | 0.808324±0.004747 | 0.109496±0.003219 | 0.272394±0.004455 | 0.295066±0.003655 | 0.417342±0.003438 | 0.159977±0.003275 | 0.303088±0.004015 |
| Timer-XL | 0.071666±0.041104 | 0.192768±0.072018 | 0.177071±0.106651 | 0.321643±0.130304 | 0.044190±0.023449 | 0.159048±0.060318 | 0.075421±0.043647 | 0.198898±0.076439 | 0.051863±0.028689 | 0.168050±0.061334 |
| Time-LLM | 1.016547±0.108007 | 0.830448±0.059991 | 1.148139±0.090069 | 0.883287±0.023071 | 0.963177±0.167258 | 0.851150±0.091467 | 1.023576±0.099477 | 0.826542±0.054822 | 0.990677±0.149426 | 0.821108±0.078290 |
| TTM | 0.430893±0.322651 | 0.476646±0.252310 | 0.951737±0.704434 | 0.752327±0.389504 | 0.260675±0.191555 | 0.404975±0.213164 | 0.454790±0.341278 | 0.493409±0.263567 | 0.321774±0.238945 | 0.428811±0.225242 |
| TimesFM | 0.002702±0.000098 | 0.036138±0.000858 | 0.024786±0.001319 | 0.124216±0.003607 | 0.004377±0.000356 | 0.031227±0.001891 | 0.002977±0.000093 | 0.033964±0.000949 | 0.002506±0.000141 | 0.030870±0.001267 |
| MOMENT | 0.011772±0.000750 | 0.084199±0.002938 | 0.016214±0.002147 | 0.093539±0.006595 | 0.016863±0.000450 | 0.093891±0.002498 | 0.012093±0.000768 | 0.083998±0.003017 | 0.012131±0.000609 | 0.086694±0.002571 |
| MOIRAI | 0.025507±0.015314 | 0.103988±0.039031 | 0.078260±0.038068 | 0.200425±0.063082 | 0.014616±0.007930 | 0.080221±0.031354 | 0.026792±0.016018 | 0.107371±0.040504 | 0.017238±0.010722 | 0.087685±0.034610 |
| GRU | 0.003493±0.001476 | 0.042391±0.004099 | 0.010210±0.002803 | 0.074563±0.010229 | 0.001242±0.000351 | 0.022845±0.003531 | 0.003085±0.000633 | 0.038363±0.005254 | 0.001888±0.000329 | 0.032268±0.003412 |
| LSTM | 0.004873±0.000863 | 0.049932±0.006160 | 0.016452±0.001341 | 0.096129±0.004177 | 0.001765±0.000563 | 0.028110±0.005645 | 0.004761±0.000681 | 0.049704±0.004797 | 0.002855±0.000673 | 0.038525±0.005457 |
| DLinear | 0.002692±0.000037 | 0.031739±0.000220 | 0.018088±0.000297 | 0.099608±0.000820 | 0.000520±0.000003 | **0.010687±0.000052** | 0.002533±0.000036 | 0.030934±0.000222 | 0.001126±0.000015 | 0.016291±0.000102 |
| SparseTSF | 0.004657±0.000012 | 0.046133±0.000041 | 0.033884±0.000093 | 0.152013±0.000139 | 0.000553±0.000001 | 0.012730±0.000080 | 0.004201±0.000012 | 0.043359±0.000043 | 0.001798±0.000004 | 0.020233±0.000026 |
| FredFormer | 0.034384±0.001476 | 0.155599±0.002844 | 0.030640±0.000907 | 0.144051±0.002667 | 0.035653±0.001776 | 0.016794±0.004120 | 0.034760±0.001669 | 0.155006±0.003154 | 0.036094±0.001763 | 0.159925±0.003471 |
| DAF (Ours) | **0.001004±0.000025** | **0.021340±0.000110** | **0.009251±0.000427** | **0.066928±0.002425** | **0.000474±0.000001** | 0.012720±0.000071 | **0.000942±0.000026** | **0.020974±0.000122** | **0.000628±0.000010** | **0.015339±0.000090** |
| Improvements | 62.70±1.44 | 32.76±0.81 | 9.40±29.06 | 10.22±15.57 | 8.85±0.72 | -19.00±1.24 | 62.81±1.55 | 32.20±0.88 | 44.25±1.63 | 5.84±1.14 |

Table 12: Zero-shot evaluation on individual satellites of three cross-domain datasets (ASTROCAST, CAPELLA, ICEYE) with mean and standard deviation in MSE and MAE. "Improvements" presents the relative performance of our model compared to the best baseline for each dataset and metric. The best result is highlighted in **bold**, and the second-best is highlighted with underline.

| Dataset | ASTROCAST-1 | | ASTROCAST-2 | | ASTROCAST-3 | | ASTROCAST-4 | | ASTROCAST-5 | | ASTROCAST-6 | |
|---|---|---|---|---|---|---|---|---|---|---|---|---|
| Metric | MSE | MAE | MSE | MAE | MSE | MAE | MSE | MAE | MSE | MAE | MSE | MAE |
| Time-MoE | 2.241446±0.020478 | 1.193310±0.007028 | 2.204455±0.019279 | 1.183778±0.005738 | 2.198212±0.019937 | 1.180867±0.005610 | 2.204810±0.017097 | 1.184818±0.005282 | 2.208653±0.018960 | 1.185501±0.006265 | 2.189656±0.017570 | 1.162630±0.003545 |
| Timer | 1.672082±0.002808 | 1.086631±0.001148 | 2.027767±0.001186 | 1.204328±0.000237 | 2.069629±0.000832 | 1.216346±0.000408 | 2.036096±0.001940 | 1.207396±0.000714 | 2.033434±0.001378 | 1.206320±0.000215 | 2.221036±0.001321 | 1.245760±0.000320 |
| Timer-XL | 1.924443±0.131249 | 1.168909±0.041676 | 2.072158±0.084493 | 1.220263±0.027673 | 2.093018±0.084380 | 1.225395±0.026210 | 2.079138±0.087018 | 1.222548±0.026941 | 2.077779±0.091096 | 1.221575±0.028206 | 2.194647±0.025171 | 1.240280±0.008297 |
| Time-LLM | 1.055431±0.060210 | 0.832568±0.038114 | 1.057526±0.058066 | 0.833602±0.037009 | 1.061183±0.060236 | 0.834053±0.037356 | 1.062231±0.062277 | 0.834002±0.038042 | 0.960147±0.151859 | 0.841751±0.087854 | 0.935907±0.165856 | 0.832085±0.096158 |
| TTM | 0.355219±0.021316 | 0.457843±0.190561 | 0.205257±0.118620 | 0.349112±0.138286 | 0.189802±0.109545 | 0.335102±0.132313 | 0.201313±0.116612 | 0.345791±0.137437 | 0.202913±0.117594 | 0.346987±0.138004 | 0.155867±0.088402 | 0.297724±0.113550 |
| TimesFM | 0.001325±0.000118 | 0.027095±0.001172 | 0.001382±0.000113 | 0.027696±0.001136 | 0.001385±0.000104 | 0.027716±0.001030 | 0.001339±0.000093 | 0.027343±0.000974 | 0.001346±0.000101 | 0.027383±0.001017 | 0.001571±0.000117 | 0.028831±0.000666 |
| MOMENT | 0.116201±0.000514 | 0.171346±0.000726 | 0.114428±0.000571 | 0.165687±0.001265 | 0.114196±0.001572 | 0.164901±0.002513 | 0.113372±0.000497 | 0.165073±0.001855 | 0.114106±0.001315 | 0.165327±0.002359 | 0.111784±0.001165 | 0.159114±0.002100 |
| MOIRAI | 0.024159±0.010571 | 0.113308±0.028198 | 0.016075±0.006324 | 0.093663±0.020684 | 0.015312±0.005896 | 0.091420±0.019638 | 0.015968±0.006238 | 0.093459±0.020493 | 0.015877±0.006227 | 0.093132±0.020462 | 0.012633±0.004547 | 0.082242±0.016264 |
| GRU | 0.001298±0.000534 | 0.027855±0.006129 | 0.002568±0.001045 | 0.039118±0.007288 | 0.002798±0.001141 | 0.040870±0.007648 | 0.002647±0.001071 | 0.039751±0.007377 | 0.002631±0.001060 | 0.039645±0.007303 | 0.002610±0.000289 | 0.039894±0.001892 |
| LSTM | 0.002939±0.000877 | 0.040628±0.007100 | 0.001669±0.000602 | 0.030714±0.006443 | 0.001623±0.000584 | 0.030234±0.006479 | 0.001626±0.000604 | 0.030648±0.006419 | 0.001661±0.000604 | 0.030668±0.006419 | 0.001877±0.001027 | 0.032943±0.010761 |
| DLinear | 0.000518±0.000006 | 0.014878±0.000113 | 0.000223±0.000001 | 0.008694±0.000034 | 0.000203±0.000001 | 0.008553±0.000037 | 0.000184±0.000001 | 0.008017±0.000032 | 0.000196±0.000001 | 0.008242±0.000033 | 0.000246±0.000001 | 0.010143±0.000047 |
| SparseTSF | 0.000453±0.000004 | 0.015699±0.000108 | 0.000307±0.000003 | 0.012181±0.000155 | 0.000388±0.000006 | 0.013043±0.000165 | 0.000288±0.000005 | 0.011558±0.000131 | 0.000266±0.000008 | 0.011375±0.000168 | 0.000445±0.000007 | 0.015854±0.000163 |
| FedFormer | 0.036831±0.002427 | 0.162168±0.004921 | 0.035604±0.002017 | 0.159620±0.004008 | 0.035376±0.001958 | 0.159045±0.003877 | 0.035541±0.001985 | 0.159444±0.003933 | 0.035555±0.002013 | 0.159488±0.004001 | 0.034506±0.001664 | 0.154938±0.003181 |
| DAF (Ours) | **0.000422±0.000017** | **0.014594±0.000273** | **0.000227±0.000005** | **0.010423±0.000406** | **0.000227±0.000005** | **0.010423±0.000406** | **0.000205±0.000000** | **0.009944±0.000406** | **0.000211±0.000008** | **0.010019±0.000437** | **0.000164±0.000007** | **0.008518±0.000611** |
| Improvements | 6.84±4.58 | 1.91±2.58 | -1.79±2.70 | -19.89±5.14 | 0.49±3.94 | -14.98±6.48 | -11.41±3.87 | -24.04±5.56 | -7.65±4.63 | -21.56±5.79 | 33.33±3.12 | 16.02±6.41 |

| Dataset | ASTROCAST-7 | | ASTROCAST-8 | | ASTROCAST-9 | | ASTROCAST-10 | | ASTROCAST-11 | | ASTROCAST-12 | |
|---|---|---|---|---|---|---|---|---|---|---|---|---|
| Metric | MSE | MAE | MSE | MAE | MSE | MAE | MSE | MAE | MSE | MAE | MSE | MAE |
| Time-MoE | 2.173325±0.011445 | 1.159749±0.003943 | 2.166314±0.011226 | 1.157171±0.005049 | 2.139732±0.018210 | 1.136947±0.008637 | 2.160046±0.015011 | 1.148438±0.005583 | 2.163388±0.015419 | 1.150390±0.004707 | 2.172768±0.018157 | 1.154260±0.006444 |
| Timer | 2.222032±0.001971 | 1.246040±0.000312 | 2.223626±0.003286 | 1.245626±0.000686 | 2.551549±0.004137 | 1.321783±0.000826 | 2.419238±0.004368 | 1.296029±0.000864 | 2.338733±0.003627 | 1.274210±0.001024 | 2.310403±0.001245 | 1.266656±0.000192 |
| Timer-XL | 2.192762±0.026974 | 1.240012±0.008506 | 2.188664±0.025750 | 1.238471±0.008012 | 2.344538±0.008653 | 1.270269±0.000557 | 2.256323±0.035164 | 1.253407±0.010973 | 2.215175±0.038391 | 1.241795±0.011859 | 2.200336±0.041625 | 1.237822±0.012834 |
| Time-LLM | 1.162740±0.084262 | 0.932590±0.020114 | 1.100531±0.009328 | 0.908783±0.009591 | 1.13790±0.051332 | 0.923016±0.008001 | 1.144697±0.039559 | 0.873645±0.003043 | 1.091056±0.022454 | 0.863103±0.022014 | 1.04295±0.094596 | 0.822933±0.048957 |
| TTM | 0.155804±0.088383 | 0.297790±0.113575 | 0.155766±0.088576 | 0.297525±0.113851 | 0.082656±0.044072 | 0.208024±0.071755 | 0.091863±0.049598 | 0.222497±0.078148 | 0.112880±0.062137 | 0.250213±0.090979 | 0.260266±0.095282 | |
| TimesFM | 0.001485±0.000122 | 0.028154±0.000899 | 0.001498±0.000122 | 0.028214±0.000834 | 0.001896±0.000042 | 0.030311±0.000037 | 0.001698±0.000133 | 0.029823±0.000978 | 0.001757±0.000132 | 0.030308±0.000850 | 0.001618±0.000155 | 0.029233±0.001129 |
| MOMENT | 0.112683±0.000708 | 0.159540±0.001965 | 0.112471±0.001115 | 0.159446±0.002014 | 0.112329±0.001250 | 0.156141±0.000838 | 0.110596±0.003232 | 0.155321±0.001625 | 0.112288±0.001851 | 0.157529±0.002511 | 0.111112±0.000689 | 0.156890±0.001688 |
| MOIRAI | 0.012658±0.004546 | 0.082308±0.016228 | 0.012636±0.012583 | 0.082213±0.016049 | 0.010165±0.002894 | 0.073469±0.011252 | 0.010699±0.003232 | 0.075507±0.012259 | 0.011601±0.003717 | 0.078543±0.013639 | 0.011787±0.003860 | 0.079291±0.014125 |
| GRU | 0.002567±0.000306 | 0.039442±0.002078 | 0.002574±0.000316 | 0.039418±0.002110 | 0.006594±0.001023 | 0.060651±0.005093 | 0.004734±0.001332 | 0.052743±0.007153 | 0.004037±0.001188 | 0.048856±0.006827 | 0.003763±0.001188 | 0.047381±0.007037 |
| LSTM | 0.001878±0.001002 | 0.032777±0.010416 | 0.001843±0.000986 | 0.032487±0.010265 | 0.004198±0.002290 | 0.046818±0.015499 | 0.002438±0.001280 | 0.036842±0.011307 | 0.002162±0.001123 | 0.034674±0.010877 | 0.002056±0.000982 | 0.034470±0.009752 |
| DLinear | 0.000225±0.000001 | 0.009855±0.000049 | 0.000228±0.000001 | 0.009916±0.000046 | 0.000305±0.000002 | 0.011696±0.000046 | 0.000414±0.000000 | 0.013497±0.000051 | 0.000436±0.000002 | 0.013391±0.000056 | 0.000377±0.000001 | 0.013191±0.000043 |
| SparseTSF | 0.000535±0.000008 | 0.015992±0.000168 | 0.000440±0.000007 | 0.015749±0.000165 | 0.000493±0.000006 | 0.016642±0.000128 | 0.000687±0.000008 | 0.019520±0.000131 | 0.000743±0.000008 | 0.020143±0.000132 | 0.000667±0.000008 | 0.019489±0.000139 |
| FedFormer | 0.034499±0.001759 | 0.155148±0.003368 | 0.034470±0.001807 | 0.155070±0.003430 | 0.031546±0.001508 | 0.146756±0.003212 | 0.033220±0.001429 | 0.150831±0.002631 | 0.033914±0.001556 | 0.152399±0.002847 | 0.033987±0.001597 | 0.152597±0.002970 |
| DAF (Ours) | **0.000159±0.000009** | **0.008533±0.000659** | **0.000152±0.000011** | **0.008467±0.000705** | **0.000153±0.000008** | **0.008379±0.000414** | **0.000229±0.000004** | **0.009735±0.000459** | **0.000258±0.000005** | **0.010152±0.000559** | **0.000212±0.000005** | **0.009697±0.000605** |
| Improvements | 29.33±4.31 | 13.41±7.12 | 33.33±5.12 | 14.61±7.51 | 49.84±3.28 | 28.36±3.82 | 44.69±1.23 | 27.87±3.67 | 40.88±1.19 | 24.19±4.49 | 43.77±1.48 | 26.49±4.83 |

| Dataset | ASTROCAST-13 | | ASTROCAST-14 | | ASTROCAST-15 | | ASTROCAST-16 | | CAPELLA-1 | | CAPELLA-2 | |
|---|---|---|---|---|---|---|---|---|---|---|---|---|
| Metric | MSE | MAE | MSE | MAE | MSE | MAE | MSE | MAE | MSE | MAE | MSE | MAE |
| Time-MoE | 2.256373±0.024297 | 1.182822±0.008516 | 2.256871±0.025732 | 1.182716±0.008277 | 2.257724±0.027336 | 1.182581±0.009033 | 2.255854±0.025273 | 1.182297±0.008599 | 2.126943±0.006795 | 1.213386±0.000686 | 2.089090±0.012122 | 1.199660±0.005162 |
| Timer | 1.518578±0.003904 | 1.020033±0.001508 | 1.520604±0.003678 | 1.020806±0.001187 | 1.520063±0.004179 | 1.020447±0.001539 | 1.521514±0.003058 | 1.020953±0.001347 | 2.324498±0.000552 | 1.351008±0.000527 | 2.555783±0.008077 | 1.418488±0.009351 |
| Timer-XL | 1.855909±0.132515 | 1.131265±0.044336 | 1.859865±0.134854 | 1.132171±0.043062 | 1.859588±0.135401 | 1.132171±0.042985 | 1.859162±0.134618 | 1.131928±0.042933 | 2.214777±0.034698 | 1.322637±0.013063 | 2.298650±0.026122 | 1.348131±0.009351 |
| Time-LLM | 1.039033±0.082565 | 0.836599±0.048336 | 1.002102±0.127512 | 0.823030±0.070337 | 0.999418±0.132103 | 0.820967±0.072654 | 1.001837±0.125862 | 0.823099±0.069773 | 1.001980±0.127705 | 0.822624±0.070487 | 0.990530±0.146404 | 0.805722±0.088207 |
| TTM | 0.432783±0.259315 | 0.498052±0.209684 | 0.431410±0.258726 | 0.497076±0.209727 | 0.433109±0.259644 | 0.498030±0.209691 | 0.431482±0.258786 | 0.496928±0.209601 | 0.109979±0.062254 | 0.250135±0.093606 | 0.061734±0.033312 | 0.189366±0.065694 |
| TimesFM | 0.001285±0.000117 | 0.026315±0.001152 | 0.001279±0.000111 | 0.026238±0.001080 | 0.001308±0.000119 | 0.026466±0.001148 | 0.001291±0.000117 | 0.026330±0.001101 | 0.001625±0.000100 | 0.030560±0.000763 | 0.001828±0.000048 | 0.032265±0.000426 |
| MOMENT | 0.117137±0.001961 | 0.171941±0.002706 | 0.117234±0.000702 | 0.171574±0.002112 | 0.117803±0.000286 | 0.172633±0.001748 | 0.117557±0.001030 | 0.171973±0.002102 | 0.110406±0.001035 | 0.164465±0.000796 | 0.109311±0.001412 | 0.165696±0.001768 |
| MOIRAI | 0.028261±0.012787 | 0.120282±0.031102 | 0.028184±0.012583 | 0.120119±0.030747 | 0.028300±0.012656 | 0.120421±0.030732 | 0.028137±0.012656 | 0.120167±0.030934 | 0.011250±0.003633 | 0.082227±0.014352 | 0.009422±0.002541 | 0.075676±0.011608 |
| GRU | 0.001277±0.000933 | 0.026863±0.009836 | 0.001295±0.000943 | 0.026946±0.009885 | 0.001313±0.000951 | 0.027065±0.009987 | 0.001323±0.000944 | 0.027176±0.009782 | 0.000889±0.000378 | 0.020485±0.004398 | 0.000313±0.000103 | 0.012797±0.002746 |
| LSTM | 0.003591±0.000877 | 0.045087±0.006856 | 0.003644±0.000861 | 0.045260±0.006823 | 0.003691±0.000884 | 0.045492±0.006899 | 0.003692±0.000876 | 0.045580±0.006876 | 0.001149±0.000477 | 0.023554±0.005442 | 0.000382±0.000098 | 0.011249±0.002352 |
| DLinear | 0.000857±0.000014 | 0.020617±0.000186 | 0.000873±0.000013 | 0.020670±0.000179 | 0.000909±0.000014 | 0.020926±0.000192 | 0.000902±0.000013 | 0.020887±0.000176 | 0.000296±0.000000 | 0.012088±0.000037 | 0.000297±0.000003 | 0.012732±0.000064 |
| SparseTSF | 0.001096±0.000008 | 0.026186±0.000088 | 0.001184±0.000008 | 0.026890±0.000083 | 0.001174±0.000008 | 0.026813±0.000085 | 0.001125±0.000000 | 0.026513±0.000085 | 0.000625±0.000004 | 0.018303±0.000107 | 0.000477±0.000005 | 0.017239±0.000098 |
| FedFormer | 0.036558±0.002660 | 0.159316±0.005228 | 0.036553±0.002672 | 0.159293±0.005258 | 0.036588±0.002667 | 0.159340±0.005258 | 0.036591±0.001507 | 0.160082±0.003646 | 0.032343±0.001507 | 0.160082±0.003646 | 0.031073±0.001212 | 0.157079±0.002986 |
| DAF (Ours) | **0.000507±0.000032** | **0.016716±0.000384** | **0.000522±0.000029** | **0.016712±0.000310** | **0.000554±0.000030** | **0.016812±0.000331** | **0.000544±0.000032** | **0.017062±0.000343** | **0.000185±0.000009** | **0.009630±0.000484** | **0.000144±0.000008** | **0.008768±0.000421** |
| Improvements | 40.84±4.70 | 18.92±2.59 | 40.21±4.21 | 18.66±2.31 | 39.05±4.24 | 18.21±2.27 | 39.69±4.42 | 18.31±2.33 | 37.50±3.04 | 20.33±4.25 | 51.52±3.18 | 31.13±3.65 |

| Dataset | CAPELLA-3 | | CAPELLA-4 | | CAPELLA-5 | | CAPELLA-6 | | ICEYE-1 | | ICEYE-2 | |
|---|---|---|---|---|---|---|---|---|---|---|---|---|
| Metric | MSE | MAE | MSE | MAE | MSE | MAE | MSE | MAE | MSE | MAE | MSE | MAE |
| Time-MoE | 2.376299±0.045500 | 1.314458±0.016757 | 2.276702±0.024165 | 1.280691±0.008930 | 2.346579±0.035438 | 1.304352±0.013292 | 2.318884±0.024808 | 1.218230±0.009814 | 2.281445±0.026833 | 1.216442±0.009581 | 2.236685±0.021438 | 1.174393±0.006612 |
| Timer | 0.645828±0.006160 | 0.693960±0.003324 | 1.048516±0.006220 | 0.900312±0.002577 | 0.810849±0.006899 | 0.783948±0.003446 | 0.889407±0.005939 | 0.777115±0.002784 | 1.273369±0.005647 | 0.949323±0.002201 | 1.701693±0.004052 | 1.080896±0.001335 |
| Timer-XL | 1.504105±0.248249 | 1.080201±0.091892 | 1.679405±0.197315 | 1.147549±0.069924 | 1.584145±0.217069 | 1.110950±0.078551 | 1.601587±0.196985 | 1.053986±0.067645 | 1.763714±0.160422 | 1.122947±0.053605 | 1.940112±0.126161 | 1.157056±0.039063 |
| Time-LLM | 0.984796±0.146182 | 0.805368±0.078771 | 0.989263±0.146869 | 0.806693±0.078568 | 0.959528±0.168847 | 0.787248±0.089775 | 0.973929±0.164091 | 0.795775±0.087081 | 0.976159±0.155804 | 0.797818±0.083483 | 0.977492±0.155091 | 0.798456±0.084381 |
| TTM | 0.876587±0.528224 | 0.759065±0.321922 | 0.637011±0.387573 | 0.643854±0.279059 | 0.774221±0.469947 | 0.711048±0.306284 | 0.754945±0.455866 | 0.664274±0.283171 | 0.540457±0.326245 | 0.568295±0.242609 | 0.329672±0.195919 | 0.435504±0.180693 |
| TimesFM | 0.015463±0.001144 | 0.104420±0.003492 | 0.001970±0.000072 | 0.034437±0.000585 | 0.007593±0.000585 | 0.072240±0.002422 | 0.005732±0.000075 | 0.058433±0.000537 | 0.001381±0.000109 | 0.027852±0.001132 | 0.001623±0.000160 | 0.029371±0.001355 |
| MOMENT | 0.132049±0.002486 | 0.221093±0.048992 | 0.120740±0.003303 | 0.195055±0.001889 | 0.125401±0.001872 | 0.126596±0.003937 | 0.126962±0.000783 | 0.194098±0.002450 | 0.037351±0.016831 | 0.182179±0.001930 | 0.115557±0.001363 | 0.168598±0.001768 |
| MOIRAI | 0.069941±0.028171 | 0.200503±0.048467 | 0.042986±0.019236 | 0.157170±0.040714 | 0.057676±0.024676 | 0.181741±0.046055 | 0.056495±0.024090 | 0.170152±0.042949 | 0.037351±0.016831 | 0.140339±0.036578 | 0.023052±0.010020 | 0.109003±0.026908 |
| GRU | 0.006289±0.001889 | 0.061575±0.004031 | 0.003028±0.000186 | 0.041873±0.000577 | 0.004584±0.000160 | 0.052451±0.006014 | 0.004932±0.002991 | 0.051309±0.016014 | 0.001986±0.001470 | 0.031983±0.011891 | 0.001667±0.000573 | 0.031841±0.005824 |
| LSTM | 0.009806±0.001889 | 0.077749±0.005473 | 0.005072±0.001864 | 0.054937±0.010153 | 0.007384±0.001725 | 0.067223±0.007896 | 0.009934±0.000905 | 0.074932±0.004984 | 0.005764±0.000907 | 0.056466±0.006036 | 0.002300±0.000907 | 0.034987±0.007362 |
| DLinear | 0.013247±0.000245 | 0.090682±0.000867 | 0.004253±0.000070 | 0.051088±0.000452 | 0.008599±0.000154 | 0.072691±0.000684 | 0.007244±0.000136 | 0.062960±0.000610 | 0.002490±0.000039 | 0.036062±0.000317 | 0.000607±0.000006 | **0.015094±0.000125** |
| SparseTSF | 0.026034±0.000083 | 0.141335±0.000161 | 0.007528±0.000030 | 0.075524±0.000115 | 0.016364±0.000054 | 0.111924±0.000134 | 0.013034±0.000100 | 0.094353±0.000134 | 0.003861±0.000016 | 0.050793±0.000091 | 0.000824±0.000003 | 0.018864±0.000083 |
| FedFormer | 0.029663±0.000752 | 0.150887±0.002434 | 0.033092±0.001974 | 0.161317±0.004727 | 0.031056±0.000941 | 0.155143±0.004727 | 0.032543±0.001173 | 0.150227±0.002198 | 0.035562±0.002684 | 0.159777±0.005567 | 0.036970±0.002388 | 0.159283±0.004561 |
| DAF (Ours) | **0.006162±0.000052** | **0.057393±0.001019** | **0.001348±0.000040** | **0.029814±0.000550** | **0.003329±0.000080** | **0.043147±0.000252** | **0.002431±0.000060** | **0.036817±0.000583** | **0.001154±0.000060** | **0.024915±0.000346** | **0.000533±0.000015** | 0.015468±0.000122 |
| Improvements | 2.02±8.76 | 6.79±7.76 | 27.38±4.47 | 27.15±4.47 | 27.58±4.47 | 17.74±2.19 | 50.71±1.15 | 28.24±23.53 | 16.44±10.94 | 10.55±4.88 | 12.19±3.34 | -2.48±1.66 |

Table 13: Zero-shot evaluation on individual satellites of ICEYE dataset with mean and standard deviation in MSE and MAE. "Improvements" presents the relative performance of our model compared to the best baseline for each dataset and metric. A dash ('-') indicates that our model fails to achieve either the best or second-best performance. The best result is highlighted in **bold**, and the second-best is highlighted with underline.

| Dataset | ICEYE-3 | | ICEYE-4 | | ICEYE-5 | | ICEYE-6 | | ICEYE-7 | | ICEYE-8 | |
| --- | --- | --- | --- | --- | --- | --- | --- | --- | --- | --- | --- | --- |
| Metric | MSE | MAE | MSE | MAE | MSE | MAE | MSE | MAE | MSE | MAE | MSE | MAE |
| Time-MoE | 2.265252±0.024884 | 1.183607±0.009877 | 2.266881±0.018425 | 1.201742±0.006342 | 2.313715±0.021415 | 1.229923±0.008306 | 2.129433±0.013724 | 1.141823±0.004852 | 2.168727±0.017825 | 1.167562±0.000691 | 2.174873±0.014950 | 1.163928±0.005529 |
| Timer | 1.465843±0.003550 | 0.999385±0.001367 | 1.496195±0.003360 | 1.025114±0.001471 | 1.020003±0.005781 | 0.844656±0.002606 | 2.640502±0.003107 | 1.355810±0.000381 | 2.328085±0.001519 | 1.288137±0.000597 | 2.279073±0.001034 | 1.266808±0.000210 |
| Timer-XL | 1.848283±0.154663 | 1.126651±0.049040 | 1.852675±0.155699 | 1.145133±0.049458 | 1.675061±0.208590 | 1.091918±0.069854 | 2.395291±0.010250 | 1.295178±0.001017 | 2.205982±0.052687 | 1.256599±0.016548 | 2.213715±0.021395 | 1.250611±0.007078 |
| Time-LLM | 1.063730±0.051688 | 0.834002±0.033935 | 1.060315±0.058650 | 0.844883±0.037616 | 1.122333±0.020378 | 0.875501±0.004928 | 0.965255±0.173064 | 0.793549±0.091673 | 0.974073±0.155858 | 0.807701±0.085109 | 0.982754±0.152822 | 0.806595±0.082019 |
| TTM | 0.540457±0.326245 | 0.568295±0.242609 | 0.329672±0.195919 | 0.435504±0.180693 | 0.435386±0.261986 | 0.499931±0.211879 | 0.415278±0.251327 | 0.494733±0.211736 | 0.656160±0.397797 | 0.626736±0.269274 | 0.056792±0.029241 | 0.171118±0.055549 |
| TimesFM | 0.001365±0.000121 | 0.026930±0.001117 | 0.001394±0.000151 | 0.027594±0.001356 | 0.003150±0.000156 | 0.041131±0.000642 | 0.001945±0.000156 | 0.031187±0.000642 | 0.001647±0.000113 | 0.030024±0.000795 | 0.001663±0.000150 | 0.029855±0.001077 |
| MOMENT | 0.118179±0.001659 | 0.173059±0.002869 | 0.118205±0.000971 | 0.175122±0.001826 | 0.124096±0.002080 | 0.190532±0.002891 | 0.112057±0.000744 | 0.157629±0.000676 | 0.110730±0.000846 | 0.158723±0.001930 | 0.112898±0.000837 | 0.159822±0.001449 |
| MOIRAI | 0.037351±0.016831 | 0.140339±0.036378 | 0.023052±0.010020 | 0.109003±0.026908 | 0.029412±0.013152 | 0.122656±0.031325 | 0.027926±0.012425 | 0.121604±0.030722 | 0.048290±0.020781 | 0.159471±0.039918 | 0.009475±0.002452 | 0.071659±0.009986 |
| GRU | 0.001207±0.000557 | 0.026659±0.006543 | 0.001680±0.000324 | 0.030077±0.003607 | 0.003180±0.000145 | 0.039935±0.003965 | 0.010238±0.002831 | 0.039935±0.009881 | 0.004810±0.001808 | 0.053073±0.009347 | 0.003053±0.000335 | 0.042867±0.003344 |
| LSTM | 0.003298±0.001269 | 0.041736±0.008834 | 0.002561±0.001398 | 0.036705±0.011253 | 0.006822±0.002614 | 0.060077±0.011577 | 0.002554±0.001188 | 0.038330±0.010446 | 0.001720±0.000656 | 0.031558±0.007401 | 0.002130±0.001117 | 0.034852±0.011098 |
| DLinear | 0.001218±0.000018 | 0.024126±0.000210 | 0.001040±0.000015 | 0.022581±0.000194 | 0.000416±0.000048 | 0.055127±0.000522 | 0.000416±0.000002 | 0.012563±0.000048 | 0.000416±0.000002 | 0.013693±0.000063 | 0.000426±0.000002 | 0.013662±0.000054 |
| SparseTSF | 0.001656±0.000010 | 0.031586±0.000088 | 0.001348±0.000092 | 0.029007±0.000092 | 0.000514±0.000091 | 0.079111±0.000098 | 0.000754±0.000000 | 0.016722±0.000097 | 0.000754±0.000288 | 0.021138±0.000132 | 0.000692±0.000008 | 0.019861±0.000139 |
| FedFormer | 0.036469±0.002701 | 0.158302±0.005301 | 0.036515±0.002694 | 0.160880±0.005432 | 0.033384±0.001816 | 0.155578±0.003716 | 0.031235±0.001151 | 0.147829±0.002356 | 0.033880±0.001550 | 0.150070±0.002991 | 0.034201±0.001592 | 0.155240±0.003111 |
| DAF (Ours) | **0.000696±0.000047** | **0.018802±0.000446** | **0.000592±0.000038** | **0.017919±0.000406** | **0.000271±0.000400** | **0.032772±0.000400** | **0.000247±0.000033** | **0.009303±0.000313** | **0.000239±0.000002** | **0.009969±0.000357** | **0.000255±0.000003** | **0.010555±0.000589** |
| Improvements | 42.86±4.70 | 22.07±2.53 | 43.08±4.47 | 20.65±2.48 | 37.59±4.94 | 17.94±7.09 | 34.86±2.24 | 25.95±2.77 | 40.62±1.25 | 24.20±4.99 | 63.15±0.86 | 46.86±3.34 |

| Dataset | ICEYE-9 | | ICEYE-10 | | ICEYE-11 | | ICEYE-12 | | ICEYE-13 | | ICEYE-14 | |
| --- | --- | --- | --- | --- | --- | --- | --- | --- | --- | --- | --- | --- |
| Metric | MSE | MAE | MSE | MAE | MSE | MAE | MSE | MAE | MSE | MAE | MSE | MAE |
| Time-MoE | 2.126243±0.015284 | 1.132395±0.005494 | 2.205098±0.015793 | 1.173248±0.005623 | 2.208241±0.015925 | 1.177897±0.005214 | 2.066945±0.015141 | 1.134657±0.004451 | 2.036338±0.011608 | 1.128640±0.000787 | 2.152797±0.014122 | 1.159395±0.004804 |
| Timer | 2.676910±0.005728 | 1.360002±0.001083 | 2.019034±0.001229 | 1.191371±0.000662 | 1.948373±0.000149 | 1.171870±0.000436 | 2.976292±0.010633 | 1.461619±0.002599 | 3.109202±0.009104 | 1.493967±0.001718 | 2.508564±0.004428 | 1.336683±0.000960 |
| Timer-XL | 2.423434±0.033092 | 1.296297±0.008269 | 2.097258±0.045925 | 1.216916±0.014830 | 2.588064±0.005999 | 1.199964±0.030428 | 2.580864±0.005909 | 1.363824±0.001777 | 2.671538±0.051646 | 1.388339±0.012297 | 2.325150±0.002668 | 1.288739±0.001524 |
| Time-LLM | 0.959319±0.175389 | 0.787199±0.092981 | 0.998758±0.124095 | 0.814997±0.068804 | 1.010388±0.116386 | 0.821425±0.064523 | 0.954401±0.167826 | 0.802383±0.091275 | 0.957362±0.151667 | 0.805723±0.082211 | 0.963546±0.169614 | 0.801624±0.001555 |
| TTM | 0.105886±0.058200 | 0.246776±0.089904 | 0.136679±0.076549 | 0.279080±0.104376 | 0.230811±0.134293 | 0.164378±0.052971 | 0.230811±0.134293 | 0.365860±0.146014 | 0.240194±0.140052 | 0.374666±0.150699 | 0.079630±0.043034 | 0.199144±0.070849 |
| TimesFM | 0.001870±0.000145 | 0.030933±0.000745 | 0.001590±0.000143 | 0.029164±0.001195 | 0.001524±0.000117 | 0.028850±0.001128 | 0.003642±0.000950 | 0.042467±0.004439 | 0.001774±0.000141 | 0.030733±0.000886 | 0.001756±0.000133 | 0.030630±0.000783 |
| MOMENT | 0.109623±0.000849 | 0.151564±0.000785 | 0.113317±0.000528 | 0.163914±0.002021 | 0.113776±0.000494 | 0.165948±0.001347 | 0.117103±0.000595 | 0.174950±0.005948 | 0.139380±0.005238 | 0.221762±0.009370 | 0.111055±0.001176 | 0.156488±0.000951 |
| MOIRAI | 0.011685±0.003781 | 0.080107±0.014080 | 0.012072±0.004105 | 0.080775±0.014945 | 0.008722±0.002158 | 0.068514±0.009021 | 0.015983±0.006299 | 0.092407±0.020290 | 0.017811±0.007181 | 0.097564±0.022091 | 0.011115±0.002648 | 0.078028±0.009978 |
| GRU | 0.006833±0.000778 | 0.063149±0.003365 | 0.001672±0.000288 | 0.031270±0.002818 | 0.001979±0.000744 | 0.034609±0.005776 | 0.020897±0.004746 | 0.109979±0.013178 | 0.005357±0.000670 | 0.056406±0.003617 | 0.005218±0.000776 | 0.054821±0.004755 |
| LSTM | 0.004365±0.002034 | 0.049669±0.013508 | 0.001890±0.000699 | 0.033025±0.007623 | 0.002076±0.000703 | 0.039658±0.010246 | 0.008928±0.002330 | 0.072469±0.010246 | 0.003035±0.001630 | 0.041287±0.012921 | 0.002972±0.001591 | 0.040224±0.012652 |
| DLinear | 0.000426±0.000002 | 0.013365±0.000046 | 0.000323±0.000001 | 0.010615±0.000052 | 0.000361±0.000002 | 0.010774±0.000058 | 0.000747±0.000009 | 0.016911±0.000127 | 0.000440±0.000002 | 0.014386±0.000051 | 0.000406±0.000001 | 0.013329±0.000043 |
| SparseTSF | 0.000642±0.000003 | 0.018134±0.000050 | 0.000508±0.000050 | 0.015480±0.000112 | 0.000318±0.000080 | 0.014517±0.000080 | 0.000728±0.000007 | 0.020435±0.000114 | 0.000400±0.000002 | 0.020435±0.000114 | 0.000686±0.000006 | 0.018837±0.000104 |
| FedFormer | 0.030772±0.001195 | 0.145263±0.002470 | 0.035739±0.002026 | 0.158615±0.003889 | 0.036141±0.004051 | 0.159903±0.004151 | 0.025935±0.001715 | 0.136004±0.004317 | 0.023402±0.001241 | 0.151735±0.002530 | 0.032782±0.001389 | 0.152736±0.002818 |
| DAF (Ours) | **0.000277±0.000007** | **0.010592±0.000223** | **0.000307±0.000008** | **0.012603±0.000451** | **0.000361±0.000002** | **0.012859±0.000289** | **0.000365±0.000010** | **0.013512±0.000289** | **0.000239±0.000002** | **0.009969±0.000357** | **0.000248±0.000004** | **0.010160±0.000446** |
| Improvements | 34.98±1.95 | 20.75±1.94 | 4.95±1.22 | -13.63±4.81 | 0.00±2.77 | -19.35±2.55 | 51.14±1.93 | 20.10±2.73 | 45.68±0.70 | 30.70±2.73 | 38.92±1.14 | 23.78±3.59 |

| Dataset | ICEYE-15 | | ICEYE-16 | | ICEYE-17 | | ICEYE-18 | | ICEYE-19 | | ICEYE-20 | |
| --- | --- | --- | --- | --- | --- | --- | --- | --- | --- | --- | --- | --- |
| Metric | MSE | MAE | MSE | MAE | MSE | MAE | MSE | MAE | MSE | MAE | MSE | MAE |
| Time-MoE | 2.170412±0.016723 | 1.166078±0.003391 | 2.210584±0.016392 | 1.182021±0.005372 | 2.190305±0.019450 | 1.173149±0.005498 | 2.215058±0.011250 | 1.182011±0.005172 | 2.206868±0.012731 | 1.178902±0.004797 | 2.202965±0.012435 | 1.177810±0.003705 |
| Timer | 2.453704±0.003156 | 1.324353±0.000780 | 1.934075±0.001768 | 1.171024±0.000886 | 2.167147±0.000886 | 1.240462±0.000731 | 1.956754±0.001353 | 1.176866±0.000717 | 2.020425±0.002318 | 1.196705±0.001096 | 2.075047±0.002034 | 1.213528±0.000888 |
| Timer-XL | 2.290651±0.102519 | 1.284496±0.002828 | 2.029638±0.097147 | 1.232648±0.030512 | 2.194744±0.068646 | 1.232519±0.021455 | 2.054385±0.078293 | 1.207477±0.024325 | 2.079119±0.067715 | 1.215410±0.020934 | 2.105747±0.062607 | 1.224171±0.019445 |
| Time-LLM | 0.974636±0.162804 | 0.807558±0.087680 | 1.001795±0.15375 | 0.824192±0.064309 | 0.986772±0.140097 | 0.812357±0.076937 | 1.012261±0.118234 | 0.823337±0.065286 | 1.003226±0.125856 | 0.819712±0.069228 | 1.000967±0.131074 | 0.818705±0.071981 |
| TTM | 0.145555±0.077044 | 0.286606±0.106661 | 0.073012±0.038346 | 0.209080±0.067661 | 0.091699±0.049671 | 0.222598±0.078262 | 0.243399±0.142444 | 0.378398±0.153171 | 0.157237±0.089320 | 0.302201±0.116104 | 0.250887±0.146642 | 0.383551±0.154282 |
| TimesFM | 0.001472±0.000123 | 0.028442±0.001220 | 0.001449±0.000114 | 0.028168±0.000931 | 0.001375±0.000142 | 0.027288±0.001187 | 0.001634±0.000145 | 0.029676±0.001128 | 0.001468±0.000133 | 0.028093±0.001006 | 0.001623±0.000169 | 0.029521±0.001348 |
| MOMENT | 0.111058±0.001513 | 0.157433±0.001913 | 0.114777±0.001529 | 0.166809±0.002539 | 0.112366±0.001691 | 0.161362±0.002581 | 0.113490±0.000560 | 0.165061±0.001727 | 0.113591±0.000560 | 0.164672±0.001628 | 0.113219±0.001324 | 0.163492±0.002162 |
| MOIRAI | 0.014595±0.003294 | 0.090475±0.010817 | 0.009708±0.002708 | 0.073052±0.010856 | 0.010561±0.003218 | 0.076043±0.012401 | 0.017947±0.007292 | 0.098200±0.022454 | 0.013798±0.005024 | 0.086641±0.017396 | 0.017300±0.007060 | 0.096366±0.022074 |
| GRU | 0.001983±0.000765 | 0.034591±0.005921 | 0.003343±0.001376 | 0.044699±0.008558 | 0.001531±0.000285 | 0.030822±0.002704 | 0.001807±0.000353 | 0.033877±0.002837 | 0.002040±0.000430 | 0.035442±0.003528 | 0.001284±0.000585 | 0.027537±0.006035 |
| LSTM | 0.002026±0.000734 | 0.033572±0.006899 | 0.001631±0.000551 | 0.030427±0.006344 | 0.001834±0.000727 | 0.034427±0.006344 | 0.001907±0.000775 | 0.032993±0.008340 | 0.001788±0.000836 | 0.031915±0.009068 | 0.002801±0.000693 | 0.039058±0.006572 |
| DLinear | 0.000316±0.000001 | 0.010160±0.000053 | 0.000236±0.000001 | 0.009713±0.000042 | 0.000171±0.000001 | 0.007621±0.000038 | 0.000337±0.000000 | 0.010810±0.000058 | 0.000209±0.000001 | 0.008496±0.000033 | 0.000665±0.000005 | 0.015417±0.000090 |
| SparseTSF | 0.000480±0.000003 | 0.013956±0.000081 | 0.000400±0.000006 | 0.010258±0.000152 | 0.000288±0.000004 | 0.010258±0.000152 | 0.000400±0.000006 | 0.015986±0.000109 | 0.000400±0.000006 | 0.013832±0.000167 | 0.000686±0.000003 | 0.016985±0.000075 |
| FedFormer | 0.036171±0.002151 | 0.160425±0.004306 | 0.034838±0.001823 | 0.157240±0.003583 | 0.035976±0.002062 | 0.159411±0.004013 | 0.035808±0.001994 | 0.158851±0.003861 | 0.035367±0.001922 | 0.158064±0.003719 | 0.036929±0.002359 | 0.160979±0.004668 |
| DAF (Ours) | **0.000336±0.000007** | **0.012557±0.000244** | **0.000180±0.000009** | **0.008180±0.000636** | **0.000212±0.000005** | **0.010291±0.000248** | **0.000333±0.000004** | **0.012457±0.000471** | **0.000204±0.000007** | **0.009643±0.000509** | **0.000573±0.000022** | **0.015879±0.000186** |
| Improvements | -6.33±2.55 | -23.59±3.05 | 23.73±4.14 | -13.63±6.95 | -23.98±3.65 | -35.03±3.93 | 1.19±1.77 | -15.24±4.98 | 2.39±3.82 | -13.50±6.43 | 13.83±3.96 | -3.00±1.81 |

| Dataset | ICEYE-21 | | ICEYE-22 | | ICEYE-23 | | ICEYE-24 | | ICEYE-25 | | ICEYE-26 | |
| --- | --- | --- | --- | --- | --- | --- | --- | --- | --- | --- | --- | --- |
| Metric | MSE | MAE | MSE | MAE | MSE | MAE | MSE | MAE | MSE | MAE | MSE | MAE |
| Time-MoE | 2.243077±0.015293 | 1.188747±0.005270 | 2.258806±0.024969 | 1.193511±0.008474 | 2.109775±0.011507 | 1.132824±0.005181 | 2.230866±0.020333 | 1.177627±0.006581 | 2.243892±0.026630 | 1.185669±0.008724 | 2.301636±0.022377 | 1.218742±0.009942 |
| Timer | 1.717463±0.003598 | 1.097497±0.001216 | 1.505151±0.001216 | 1.023903±0.001139 | 2.764314±0.007889 | 1.384566±0.002118 | 1.772952±0.005446 | 1.108447±0.001683 | 1.644469±0.003161 | 1.069312±0.001587 | 1.179911±0.006414 | 0.908897±0.002514 |
| Timer-XL | 1.949125±0.102519 | 1.171473±0.032393 | 1.851008±0.141249 | 1.139208±0.045496 | 2.474616±0.055749 | 1.313563±0.012630 | 1.958739±0.110312 | 1.167915±0.034941 | 1.907127±0.125535 | 1.154574±0.039922 | 1.738073±0.152967 | 1.108821±0.050618 |
| Time-LLM | 1.037016±0.087233 | 0.831692±0.050484 | 1.063860±0.059530 | 0.842156±0.037050 | 0.960183±0.174245 | 0.789691±0.037050 | 1.027050±0.095222 | 0.823061±0.053999 | 1.041559±0.078525 | 0.831144±0.046406 | 1.103040±0.099535 | 0.864586±0.016465 |
| TTM | 0.224879±0.130689 | 0.362661±0.144548 | 0.203797±0.117594 | 0.345349±0.135835 | 0.352979±0.209606 | 0.453647±0.187922 | 0.434738±0.260616 | 0.503387±0.212407 | 0.049251±0.025150 | 0.153004±0.047866 | 0.315399±0.186709 | 0.426407±0.175951 |
| TimesFM | 0.001391±0.000124 | 0.027597±0.001282 | 0.000279±0.000315 | 0.033284±0.001351 | 0.001335±0.000125 | 0.026866±0.001332 | 0.001546±0.000141 | 0.029092±0.001326 | 0.001365±0.000120 | 0.027146±0.001205 | 0.001826±0.000143 | 0.031255±0.001184 |
| MOMENT | 0.116578±0.001117 | 0.170609±0.002210 | 0.117202±0.001218 | 0.173990±0.001959 | 0.114370±0.001218 | 0.160874±0.002236 | 0.116183±0.001110 | 0.168207±0.001809 | 0.121411±0.001110 | 0.177201±0.002215 | 0.121411±0.001110 | 0.183421±0.025888 |
| MOIRAI | 0.016328±0.006399 | 0.093732±0.020481 | 0.015246±0.005910 | 0.090609±0.019509 | 0.029079±0.009915 | 0.109938±0.026950 | 0.028972±0.012896 | 0.122904±0.031336 | 0.009832±0.002264 | 0.071970±0.009009 | 0.021485±0.009270 | 0.096366±0.022074 |
| GRU | 0.001397±0.000935 | 0.027873±0.009366 | 0.001645±0.000245 | 0.028202±0.010685 | 0.001339±0.000432 | 0.028483±0.004758 | 0.001355±0.000618 | 0.028262±0.006423 | 0.002900±0.001991 | 0.039053±0.013913 | 0.003460±0.002273 | 0.043363±0.014385 |
| LSTM | 0.003995±0.000960 | 0.047245±0.007063 | 0.002900±0.000806 | 0.041122±0.010386 | 0.002474±0.000806 | 0.037242±0.007182 | 0.003265±0.000806 | 0.042394±0.007160 | 0.006058±0.001207 | 0.058345±0.005532 | 0.006897±0.001207 | 0.062454±0.005972 |
| DLinear | 0.001081±0.000015 | 0.022767±0.000190 | 0.000440±0.000000 | 0.013660±0.000035 | 0.000324±0.000000 | 0.011000±0.000083 | 0.000711±0.000008 | 0.019790±0.000147 | 0.000350±0.000055 | 0.042025±0.000381 | 0.004338±0.000077 | 0.048276±0.000087 |
| SparseTSF | 0.001332±0.000008 | 0.028663±0.000091 | 0.000755±0.000003 | 0.020209±0.000096 | 0.000313±0.000000 | 0.010919±0.000000 | 0.000870±0.000096 | 0.020806±0.000096 | 0.000313±0.000000 | 0.058864±0.000095 | 0.007139±0.000026 | 0.069796±0.000087 |
| FedFormer | 0.036575±0.002717 | 0.160788±0.005492 | 0.030428±0.001002 | 0.146604±0.002097 | 0.036674±0.002322 | 0.159972±0.004533 | 0.036952±0.002403 | 0.157461±0.004902 | 0.034950±0.002403 | 0.157461±0.004902 | 0.034139±0.004902 | 0.155143±0.004069 |
| DAF (Ours) | **0.000658±0.000043** | **0.018463±0.000477** | **0.000308±0.000001** | **0.011238±0.000218** | **0.000586±0.000027** | **0.016468±0.000287** | **0.001241±0.000050** | **0.026291±0.000195** | **0.001384±0.000045** | **0.028418±0.000150** | **0.001384±0.000195** | **0.028418±0.000150** |
| Improvements | 39.13±4.82 | 18.90±2.77 | 30.00±0.23 | 17.73±1.81 | 17.58±4.72 | 4.20±2.49 | 9.08±11.66 | 3.15±5.02 | 24.21±8.40 | 9.08±8.92 | - | - |

Table 14: Zero-shot evaluation on individual satellites of two cross-domain datasets (ICEYE and KINEIS) with mean and standard deviation in MSE and MAE. "Improvements" presents the relative performance of our model compared to the best baseline for each dataset and metric. The best result is highlighted in **bold**, and the second-best is highlighted with underline.

| Dataset | ICEYE-27 MSE | ICEYE-27 MAE | ICEYE-28 MSE | ICEYE-28 MAE | ICEYE-29 MSE | ICEYE-29 MAE | ICEYE-30 MSE | ICEYE-30 MAE | ICEYE-31 MSE | ICEYE-31 MAE | ICEYE-32 MSE | ICEYE-32 MAE |
|---|---|---|---|---|---|---|---|---|---|---|---|---|
| Time-MoE | 2.281445±0.026833 | 1.216442±0.009581 | 2.236685±0.021438 | 1.174393±0.006612 | 2.265252±0.024884 | 1.183607±0.009877 | 2.266881±0.018425 | 1.201742±0.006342 | 2.313715±0.021415 | 1.229923±0.008306 | 2.129433±0.013724 | 1.141823±0.004852 |
| Timer | 1.083233±0.004866 | 0.867530±0.001856 | 1.096562±0.005748 | 0.873200±0.002287 | 0.971613±0.005983 | 0.813652±0.002286 | 0.970245±0.005983 | 0.813420±0.002286 | 1.005704±0.005329 | 0.827769±0.002200 | 0.968820±0.004839 | 0.812713±0.001912 |
| Timer-XL | 1.698194±0.164263 | 1.093036±0.005433 | 1.704350±0.161783 | 1.095575±0.054032 | 1.640497±0.193354 | 1.066549±0.065102 | 1.641426±0.193903 | 1.067482±0.065329 | 1.655575±0.191987 | 1.070160±0.064070 | 1.638314±0.192212 | 1.066042±0.064768 |
| Time-LLM | 1.117601±0.007101 | 0.868939±0.009590 | 1.111870±0.006993 | 0.867074±0.010890 | 1.132746±0.023145 | 0.868784±0.003063 | 1.132746±0.023145 | 0.866072±0.003063 | 1.122509±0.022752 | 0.865828±0.004246 | 1.30113±0.027881 | 0.867762±0.002115 |
| TTM | 0.671192±0.404603 | 0.629693±0.267395 | 0.664440±0.401077 | 0.624431±0.266567 | 0.709502±0.428716 | 0.643160±0.274623 | 0.709502±0.428869 | 0.643365±0.274796 | 0.690943±0.417242 | 0.633852±0.270376 | 0.711184±0.430095 | 0.643811±0.275324 |
| TimesFM | 0.001596±0.000143 | 0.029505±0.001256 | 0.003717±0.000098 | 0.046218±0.000544 | 0.002982±0.000164 | 0.046261±0.000223 | 0.002982±0.000164 | 0.041135±0.001158 | 0.003857±0.000066 | 0.047010±0.000396 | 0.001199±0.000153 | 0.026556±0.001250 |
| MOMENT | 0.123222±0.000835 | 0.186750±0.002066 | 0.121199±0.002644 | 0.185002±0.003252 | 0.124308±0.001280 | 0.189198±0.002439 | 0.125204±0.000935 | 0.189257±0.002581 | 0.124678±0.001082 | 0.188425±0.002180 | 0.12361±0.001188 | 0.188762±0.002526 |
| MOIRAI | 0.045849±0.020281 | 0.154485±0.039790 | 0.045061±0.019924 | 0.153264±0.039554 | 0.051179±0.022247 | 0.162277±0.041224 | 0.051179±0.022247 | 0.161604±0.041197 | 0.049662±0.021589 | 0.159261±0.040630 | 0.051647±0.022363 | 0.162423±0.041380 |
| GRU | 0.003273±0.002223 | 0.042270±0.014416 | 0.004080±0.002680 | 0.046524±0.015703 | 0.004153±0.002493 | 0.046967±0.015427 | 0.003683±0.002493 | 0.044162±0.015427 | 0.004234±0.002732 | 0.047284±0.015846 | 0.002241±0.000429 | 0.035264±0.001432 |
| LSTM | 0.000619±0.001169 | 0.061376±0.005890 | 0.008750±0.000916 | 0.070403±0.005282 | 0.008180±0.000939 | 0.068164±0.005384 | 0.008180±0.000939 | 0.068164±0.005384 | 0.008953±0.000939 | 0.071087±0.005183 | 0.003752±0.001614 | 0.044663±0.010520 |
| DLinear | 0.004090±0.000074 | 0.046933±0.000440 | 0.005788±0.000106 | 0.056198±0.000538 | 0.005178±0.000096 | 0.056255±0.000520 | 0.005178±0.000096 | 0.053141±0.000520 | 0.010206±0.000034 | 0.056695±0.000525 | 0.002237±0.000035 | 0.036923±0.000319 |
| SparseTSF | 0.006675±0.003044 | 0.067693±0.000090 | 0.010236±0.000095 | 0.083346±0.000995 | 0.009219±0.000035 | 0.079041±0.000097 | 0.009219±0.000033 | 0.079041±0.000092 | 0.012722±0.000044 | 0.083408±0.000095 | 0.003153±0.000017 | 0.052379±0.000101 |
| FedFormer | 0.034206±0.002088 | 0.155324±0.004156 | 0.033162±0.001567 | 0.152044±0.002981 | 0.033204±0.001479 | 0.152117±0.002671 | 0.033204±0.001479 | 0.152133±0.002671 | 0.033288±0.001518 | 0.092647±0.000101 | 0.034527±0.002595 | 0.165378±0.006139 |
| DAF (Ours) | **0.001293±0.000043** | **0.027624±0.000111** | **0.001683±0.000111** | **0.031546±0.000502** | **0.001496±0.000062** | **0.029878±0.000343** | **0.001856±0.000057** | **0.031963±0.000364** | **0.001730±0.000070** | **0.031841±0.000505** | **0.000918±0.000048** | **0.024337±0.000184** |
| Improvements | 18.98±9.95 | 6.38±4.36 | 54.72±3.08 | 31.75±1.89 | 49.83±4.84 | 27.37±2.88 | 47.50±2.73 | 25.99±1.90 | 55.15±2.58 | 32.27±1.64 | 23.44±13.77 | 8.36±5.01 |

| Dataset | ICEYE-33 MSE | ICEYE-33 MAE | ICEYE-34 MSE | ICEYE-34 MAE | ICEYE-35 MSE | ICEYE-35 MAE | ICEYE-36 MSE | ICEYE-36 MAE | ICEYE-37 MSE | ICEYE-37 MAE | ICEYE-38 MSE | ICEYE-38 MAE |
|---|---|---|---|---|---|---|---|---|---|---|---|---|
| Time-MoE | 2.168727±0.017825 | 1.167562±0.006191 | 2.174873±0.014950 | 1.163928±0.005529 | 2.126243±0.015284 | 1.132395±0.005494 | 2.205098±0.015793 | 1.173248±0.005623 | 2.208241±0.015925 | 1.177897±0.005214 | 2.066945±0.015141 | 1.134657±0.004451 |
| Timer | 1.238134±0.004995 | 0.981590±0.002112 | 1.240364±0.004754 | 0.982068±0.001913 | 0.895118±0.006878 | 0.777032±0.002933 | 0.908195±0.005926 | 0.782678±0.002552 | 0.998259±0.006001 | 0.822422±0.002486 | 0.902733±0.006625 | 0.780220±0.002750 |
| Timer-XL | 1.753132±0.173155 | 1.173540±0.006052 | 1.752112±0.175838 | 1.173253±0.061224 | 1.608863±0.199337 | 1.052308±0.067596 | 1.613022±0.199672 | 1.053685±0.065567 | 1.645918±0.186557 | 1.065373±0.062983 | 1.606835±0.198325 | 1.051396±0.067379 |
| Time-LLM | 1.073296±0.021932 | 0.897347±0.022495 | 1.074522±0.021620 | 0.897163±0.023133 | 1.142534±0.042624 | 0.870262±0.003687 | 1.138701±0.036312 | 0.868574±0.002101 | 1.127139±0.022259 | 0.863563±0.003867 | 1.139562±0.040498 | 0.869147±0.002922 |
| TTM | 0.538489±0.327184 | 0.591029±0.255668 | 0.539167±0.327589 | 0.591101±0.255951 | 0.747603±0.454791 | 0.661449±0.281630 | 0.747603±0.451202 | 0.658657±0.280280 | 0.697068±0.421266 | 0.634970±0.271007 | 0.751129±0.453179 | 0.660265±0.280947 |
| TimesFM | 0.001211±0.000138 | 0.026654±0.001130 | 0.005514±0.000064 | 0.057219±0.000471 | 0.005304±0.000047 | 0.055755±0.000356 | 0.003535±0.000075 | 0.043188±0.000617 | 0.005458±0.000044 | 0.056678±0.000310 | 0.005594±0.000070 | 0.057403±0.000458 |
| MOMENT | 0.117694±0.001198 | 0.189009±0.002611 | 0.118158±0.001495 | 0.189088±0.001495 | 0.126448±0.001408 | 0.193039±0.002829 | 0.121176±0.000741 | 0.193429±0.002529 | 0.123753±0.001525 | 0.188045±0.002950 | 0.125430±0.000740 | 0.191976±0.002226 |
| MOIRAI | 0.034966±0.015830 | 0.142004±0.036841 | 0.035017±0.015894 | 0.142144±0.036949 | 0.056225±0.024027 | 0.169132±0.042766 | 0.055385±0.023613 | 0.167864±0.042266 | 0.050198±0.021825 | 0.159485±0.040649 | 0.055820±0.023941 | 0.168450±0.042769 |
| GRU | 0.002243±0.000409 | 0.035382±0.001450 | 0.004970±0.000036 | 0.051649±0.016225 | 0.004913±0.002988 | 0.051165±0.016146 | 0.004052±0.002638 | 0.046128±0.015874 | 0.004926±0.003021 | 0.051267±0.016211 | 0.005924±0.001040 | 0.056571±0.003479 |
| LSTM | 0.003773±0.001630 | 0.046793±0.001519 | 0.009869±0.000972 | 0.074718±0.005136 | 0.009808±0.000983 | 0.074292±0.005202 | 0.008681±0.000886 | 0.069507±0.005415 | 0.009838±0.000976 | 0.074358±0.005128 | 0.007531±0.002873 | 0.063488±0.013395 |
| DLinear | 0.002258±0.000035 | 0.037028±0.000315 | 0.007183±0.000134 | 0.062151±0.000598 | 0.007016±0.000099 | 0.061625±0.000590 | 0.005644±0.000099 | 0.054739±0.000503 | 0.007097±0.000131 | 0.061992±0.000592 | 0.007232±0.000136 | 0.062319±0.000601 |
| SparseTSF | 0.003583±0.000018 | 0.052081±0.000098 | 0.012841±0.000043 | 0.093358±0.000104 | 0.012318±0.000040 | 0.091368±0.000098 | 0.012722±0.000034 | 0.079982±0.000093 | 0.012722±0.000092 | 0.092647±0.000101 | 0.012970±0.000043 | 0.093363±0.000100 |
| FedFormer | 0.034575±0.002606 | 0.165458±0.006167 | 0.032599±0.001173 | 0.149842±0.002142 | 0.032788±0.001214 | 0.150342±0.002201 | 0.033541±0.001647 | 0.152354±0.003024 | 0.032673±0.001182 | 0.150146±0.002124 | 0.032628±0.001212 | 0.149601±0.002085 |
| DAF (Ours) | **0.000942±0.000053** | **0.024508±0.000230** | **0.002351±0.000069** | **0.036216±0.000595** | **0.002271±0.000078** | **0.035643±0.000651** | **0.001856±0.000057** | **0.031963±0.000364** | **0.002324±0.000077** | **0.035979±0.000672** | **0.002429±0.000071** | **0.036500±0.000617** |
| Improvements | 22.21±13.24 | 8.05±4.76 | 52.70±30.28 | 29.88±23.18 | 53.78±29.70 | 30.34±23.26 | 49.83±4.84 | 27.37±2.88 | 52.82±30.50 | 29.82±23.50 | 56.35±1.81 | 35.48±5.06 |

| Dataset | ICEYE-39 MSE | ICEYE-39 MAE | ICEYE-40 MSE | ICEYE-40 MAE | ICEYE-41 MSE | ICEYE-41 MAE | KINEIS-1 MSE | KINEIS-1 MAE | KINEIS-2 MSE | KINEIS-2 MAE | KINEIS-3 MSE | KINEIS-3 MAE |
|---|---|---|---|---|---|---|---|---|---|---|---|---|
| Time-MoE | 2.036338±0.011608 | 1.128640±0.009787 | 2.152797±0.014122 | 1.159395±0.004804 | 2.170412±0.016723 | 1.160078±0.003291 | 2.363187±0.041345 | 1.224442±0.014970 | 2.363263±0.041345 | 1.224171±0.014647 | 2.376758±0.046048 | 1.226933±0.016189 |
| Timer | 0.892927±0.006257 | 0.773756±0.002822 | 0.900127±0.005468 | 0.777262±0.002466 | 0.897257±0.005865 | 0.776149±0.002511 | 0.617692±0.006507 | 0.634506±0.003405 | 0.617692±0.006507 | 0.634506±0.003709 | 0.558958±0.006467 | 0.600136±0.003709 |
| Timer-XL | 1.603360±0.225096 | 1.048846±0.074895 | 1.605339±0.228775 | 1.049273±0.075883 | 1.605339±0.228775 | 1.049273±0.075883 | 1.499409±0.237765 | 0.108339±0.082969 | 1.499409±0.237765 | 1.246098±0.086669 | 1.468746±0.246098 | 0.999139±0.086669 |
| Time-LLM | 1.140685±0.039510 | 0.867286±0.003136 | 1.137600±0.040905 | 0.866678±0.003167 | 1.140175±0.040160 | 0.866678±0.002875 | 1.182535±0.098073 | 0.882127±0.023239 | 1.190690±0.108080 | 0.884309±0.027325 | 1.190690±0.108080 | 0.884309±0.027325 |
| TTM | 0.736340±0.449496 | 0.650705±0.282417 | 0.734647±0.447638 | 0.649968±0.281263 | 0.886387±0.531896 | 0.719143±0.301471 | 0.891698±0.534906 | 0.721309±0.302433 | 0.929992±0.556120 | 0.736873±0.307017 | 0.929992±0.556120 | 0.736873±0.307017 |
| TimesFM | 0.005501±0.000066 | 0.056706±0.000444 | 0.005478±0.000058 | 0.056530±0.000413 | 0.018023±0.000765 | 0.056402±0.000478 | 0.018422±0.000802 | 0.106594±0.002472 | 0.023281±0.000980 | 0.120267±0.002710 | 0.023281±0.000980 | 0.120267±0.002710 |
| MOMENT | 0.126669±0.001006 | 0.193328±0.002534 | 0.126641±0.000896 | 0.193226±0.001735 | 0.136160±0.000835 | 0.193340±0.002426 | 0.137692±0.002038 | 0.215548±0.004369 | 0.139449±0.002820 | 0.218217±0.005320 | 0.139449±0.002820 | 0.218217±0.005320 |
| MOIRAI | 0.055262±0.023313 | 0.167852±0.041369 | 0.054821±0.022992 | 0.167068±0.040949 | 0.054729±0.029774 | 0.166862±0.041034 | 0.076829±0.029943 | 0.198004±0.046653 | 0.082597±0.031603 | 0.205605±0.047696 | 0.082597±0.031603 | 0.205605±0.047696 |
| GRU | 0.005868±0.001051 | 0.056344±0.003530 | 0.005883±0.001055 | 0.056328±0.003533 | 0.005913±0.001043 | 0.056401±0.003512 | 0.006768±0.002192 | 0.060622±0.009360 | **0.007647±0.002268** | **0.064690±0.008980** | 0.007647±0.002268 | 0.064690±0.008980 |
| LSTM | 0.007470±0.002870 | 0.063264±0.013355 | 0.007141±0.002842 | 0.063229±0.013330 | 0.007488±0.002843 | 0.063240±0.013314 | 0.014040±0.001291 | 0.087571±0.004828 | 0.015331±0.001371 | 0.091596±0.004748 | 0.015331±0.001371 | 0.091596±0.004748 |
| DLinear | 0.007129±0.000133 | 0.061898±0.000593 | 0.007141±0.000132 | 0.061927±0.000596 | 0.007156±0.000133 | 0.061987±0.000593 | 0.015074±0.000297 | 0.090091±0.000899 | 0.017424±0.000352 | 0.096905±0.000985 | 0.017424±0.000352 | 0.096905±0.000985 |
| SparseTSF | 0.012776±0.000041 | 0.092580±0.000097 | 0.012915±0.000043 | 0.092933±0.000097 | 0.012685±0.000043 | 0.092273±0.000101 | 0.028128±0.000092 | 0.137133±0.000145 | 0.032855±0.000108 | 0.148182±0.000159 | 0.032855±0.000108 | 0.148182±0.000159 |
| FedFormer | 0.032692±0.001317 | 0.149704±0.002327 | 0.032641±0.001262 | 0.149562±0.002210 | 0.032584±0.001226 | 0.149516±0.002098 | 0.030593±0.001127 | 0.142837±0.003354 | 0.030608±0.001184 | 0.142552±0.003498 | 0.030608±0.001184 | 0.142552±0.003498 |
| DAF (Ours) | **0.002361±0.000080** | **0.036115±0.000669** | **0.002390±0.000072** | **0.036241±0.000616** | **0.002388±0.000066** | **0.036252±0.000594** | **0.002324±0.000188** | **0.035339±0.001594** | 0.008712±0.000451 | 0.064257±0.002681 | 0.008712±0.000451 | 0.064257±0.002681 |
| Improvements | 57.08±1.97 | 35.90±5.20 | 56.37±1.78 | 35.66±5.13 | 56.10±1.79 | 35.73±1.60 | -8.08±37.78 | 3.44±17.46 | -13.93±39.69 | 0.67±7.793 | -13.93±39.69 | 0.67±7.793 |

| Dataset | KINEIS-4 MSE | KINEIS-4 MAE | KINEIS-5 MSE | KINEIS-5 MAE | KINEIS-6 MSE | KINEIS-6 MAE | KINEIS-7 MSE | KINEIS-7 MAE | KINEIS-8 MSE | KINEIS-8 MAE | KINEIS-9 MSE | KINEIS-9 MAE |
|---|---|---|---|---|---|---|---|---|---|---|---|---|
| Time-MoE | 2.361364±0.041595 | 1.223209±0.015173 | 2.361737±0.043308 | 1.223451±0.015651 | 2.386155±0.051714 | 1.227699±0.017343 | 2.383921±0.050179 | 1.227205±0.016736 | 2.382516±0.049233 | 1.226647±0.016859 | 2.38321±0.051543 | 1.227084±0.017669 |
| Timer | 0.622934±0.006323 | 0.637512±0.003420 | 0.625528±0.007494 | 0.638950±0.003991 | 0.520691±0.006415 | 0.576150±0.003563 | 0.520691±0.006415 | 0.576150±0.004192 | 0.522496±0.007031 | 0.577307±0.004255 | 0.521269±0.007899 | 0.576866±0.004950 |
| Timer-XL | 1.498262±0.238044 | 1.010649±0.082740 | 1.501027±0.235684 | 1.011752±0.081892 | 1.432471±0.231545 | 0.983878±0.082399 | 1.433108±0.230479 | 0.983878±0.082852 | 1.434620±0.230479 | 0.984300±0.081816 | 1.430744±0.229485 | 0.982959±0.081708 |
| Time-LLM | 1.179688±0.097073 | 0.881293±0.023196 | 1.181560±0.097229 | 0.881864±0.023128 | 1.197577±0.116193 | 0.885709±0.029775 | 1.201667±0.115983 | 0.886618±0.029804 | 1.204724±0.113030 | 0.886618±0.029284 | 1.204724±0.113030 | 0.887446±0.029284 |
| TTM | 0.889346±0.533037 | 0.720377±0.321114 | 0.886778±0.532114 | 0.719248±0.301686 | 0.987953±0.587565 | 0.757722±0.312858 | 0.988334±0.587182 | 0.757720±0.312262 | 0.989790±0.587918 | 0.758535±0.312553 | 0.989790±0.587918 | 0.758535±0.312553 |
| TimesFM | 0.018090±0.000764 | 0.105631±0.002388 | 0.017905±0.000730 | 0.104984±0.002295 | 0.026816±0.001126 | 0.129242±0.002917 | 0.026646±0.001117 | 0.128850±0.002877 | 0.026646±0.001117 | 0.128850±0.002877 | 0.026754±0.001113 | 0.129151±0.002875 |
| MOMENT | 0.136619±0.029695 | 0.212311±0.004479 | 0.136279±0.003094 | 0.212457±0.005495 | 0.143147±0.003047 | 0.224229±0.005882 | 0.141781±0.003402 | 0.223041±0.005949 | 0.144703±0.005949 | 0.223235±0.005988 | 0.144703±0.005988 | 0.223235±0.005988 |
| MOIRAI | 0.076196±0.029695 | 0.197343±0.046428 | 0.075689±0.029622 | 0.196562±0.046434 | 0.087049±0.032925 | 0.211156±0.048542 | 0.087079±0.033009 | 0.210630±0.048861 | 0.086073±0.032916 | 0.210662±0.048714 | 0.086073±0.032916 | 0.210662±0.048714 |
| GRU | **0.006599±0.002189** | **0.059962±0.009395** | **0.006596±0.002180** | **0.059819±0.009360** | 0.011635±0.004732 | 0.079690±0.016681 | 0.011550±0.004725 | 0.079432±0.016690 | 0.017830±0.001624 | 0.099785±0.016683 | 0.017511±0.004725 | 0.079285±0.016683 |
| LSTM | 0.013831±0.001275 | 0.087016±0.004817 | 0.013816±0.001275 | 0.086817±0.004817 | 0.018000±0.001598 | 0.100224±0.005026 | 0.017907±0.001672 | 0.099888±0.005037 | 0.019136±0.000389 | 0.099073±0.005070 | 0.017830±0.001624 | 0.099713±0.005070 |
| DLinear | 0.014756±0.000293 | 0.089171±0.000901 | 0.014714±0.000289 | 0.088977±0.000885 | 0.019298±0.000385 | 0.102274±0.001026 | 0.019114±0.000387 | 0.101762±0.001033 | 0.019114±0.001033 | 0.101762±0.001039 | 0.019136±0.000389 | 0.101827±0.001039 |
| SparseTSF | 0.027281±0.000091 | 0.135211±0.000149 | 0.027383±0.000089 | 0.135399±0.000145 | 0.036077±0.000124 | 0.156095±0.000178 | 0.036363±0.000122 | 0.156028±0.000175 | 0.036309±0.000145 | 0.155997±0.000176 | 0.036309±0.000145 | 0.155997±0.000176 |
| FedFormer | 0.030782±0.001094 | 0.143322±0.003216 | 0.030684±0.001119 | 0.143094±0.003357 | 0.030684±0.001139 | 0.142755±0.003473 | 0.030679±0.001123 | 0.142709±0.003441 | 0.030712±0.001143 | 0.142795±0.003410 | 0.030712±0.001143 | 0.142795±0.003410 |
| DAF (Ours) | 0.007172±0.000170 | 0.057996±0.001480 | 0.007123±0.000169 | 0.057708±0.001468 | **0.010055±0.000695** | **0.069617±0.003679** | **0.009943±0.000686** | **0.069217±0.003646** | **0.009977±0.000685** | **0.069439±0.003623** | **0.009977±0.000685** | **0.069439±0.003623** |
| Improvements | -8.68±38.63 | 3.28±17.62 | -7.99±38.25 | 3.53±17.55 | 13.58±41.12 | 12.64±22.90 | 13.91±41.16 | 12.86±22.90 | 13.33±41.53 | 12.42±23.00 | 13.33±41.53 | 12.42±23.00 |

Table 15: Zero-shot evaluation on individual satellites of two cross-domain datasets (KINEIS and KUIPER) with mean and standard deviation in MSE and MAE. A dash ('-') indicates that our model fails to achieve either the best or second-best performance. "Improvements" presents the relative performance of our model compared to the best baseline for each dataset and metric. The best result is highlighted in **bold**, and the second-best is highlighted with underline.

| Dataset | KINEIS-10 | | KINEIS-11 | | KINEIS-12 | | KINEIS-13 | | KINEIS-14 | | KINEIS-15 | |
| --- | --- | --- | --- | --- | --- | --- | --- | --- | --- | --- | --- | --- |
| Metric | MSE | MAE | MSE | MAE | MSE | MAE | MSE | MAE | MSE | MAE | MSE | MAE |
| Time-MoE | 2.382871±0.049719 | 1.227161±0.0016934 | 2.404022±0.051520 | 1.252228±0.019322 | 2.398584±0.046914 | 1.248923±0.015333 | 2.393839±0.045102 | 1.248783±0.015797 | 2.408809±0.050374 | 1.249923±0.017126 | 2.384278±0.044790 | 1.244286±0.015813 |
| Timer | 0.520499±0.006960 | 0.576353±0.004401 | 0.507268±0.007192 | 0.575802±0.004483 | 0.511439±0.007760 | 0.578434±0.004728 | 0.518093±0.007153 | 0.582329±0.004567 | 0.515099±0.007476 | 0.580355±0.004729 | 0.576264±0.007134 | 0.617360±0.004008 |
| Timer-XL | 1.432824±0.231832 | 0.983685±0.082454 | 1.465110±0.220066 | 1.009243±0.077844 | 1.466423±0.219927 | 1.009776±0.078148 | 1.469882±0.218521 | 1.011020±0.077675 | 1.469751±0.219124 | 1.011315±0.078004 | 1.500608±0.209796 | 1.021754±0.037738 |
| Time-LLM | 1.198519±0.116661 | 0.885791±0.029842 | 1.204470±0.116720 | 0.899725±0.030662 | 1.200216±0.115830 | 0.898409±0.030477 | 1.200006±0.119653 | 0.898305±0.031382 | 1.198762±0.115959 | 0.897590±0.029769 | 1.187704±0.103212 | 0.893249±0.025304 |
| TTM | 0.987953±0.587329 | 0.757562±0.312576 | 1.033511±0.611466 | 0.786205±0.321283 | 1.031540±0.610265 | 0.785226±0.320885 | 1.026967±0.607078 | 0.783699±0.320858 | 1.028940±0.609274 | 0.784154±0.320955 | 0.985259±0.587134 | 0.765550±0.317056 |
| TimesFM | 0.026662±0.001111 | 0.128910±0.002865 | 0.027792±0.001287 | 0.133479±0.003113 | 0.027516±0.001326 | 0.132893±0.003015 | 0.027058±0.001209 | 0.131633±0.003015 | 0.027391±0.001357 | 0.132314±0.003485 | 0.021313±0.001081 | 0.116386±0.003094 |
| MOMENT | 0.142335±0.003341 | 0.223566±0.005950 | 0.143195±0.001760 | 0.228102±0.005332 | 0.143362±0.003322 | 0.227638±0.006444 | 0.142462±0.002880 | 0.227027±0.006047 | 0.143793±0.004301 | 0.227963±0.007292 | 0.138728±0.004446 | 0.219108±0.006792 |
| MOIRAI | 0.086991±0.032911 | 0.210684±0.048709 | 0.090281±0.033285 | 0.218363±0.049076 | 0.090045±0.032985 | 0.218009±0.048649 | 0.089973±0.033270 | 0.217817±0.049160 | 0.090063±0.032787 | 0.217973±0.048491 | 0.082757±0.031281 | 0.208473±0.047825 |
| GRU | 0.011480±0.004708 | 0.079231±0.016689 | 0.011902±0.002685 | 0.081520±0.009341 | 0.011864±0.002707 | 0.081311±0.009533 | 0.011977±0.002672 | 0.081611±0.009392 | 0.011942±0.002681 | 0.081399±0.009346 | 0.010215±0.002502 | 0.075392±0.009428 |
| LSTM | 0.017800±0.001611 | 0.099683±0.005060 | 0.016960±0.002845 | 0.098762±0.007823 | 0.016958±0.002818 | 0.098362±0.007812 | 0.016981±0.002818 | 0.098673±0.007812 | 0.016967±0.002879 | 0.098463±0.008003 | 0.014809±0.002648 | 0.092146±0.007883 |
| DLinear | 0.019096±0.000389 | 0.101723±0.001039 | 0.020158±0.000412 | 0.105742±0.001077 | 0.020047±0.000407 | 0.105422±0.001062 | 0.019955±0.000403 | 0.105131±0.001062 | 0.020078±0.000408 | 0.105321±0.001073 | 0.016781±0.000340 | 0.096280±0.000974 |
| SparseTSF | 0.036365±0.000123 | 0.156093±0.000127 | 0.037809±0.000197 | 0.161556±0.000185 | 0.037355±0.000125 | 0.159612±0.000182 | 0.036454±0.000125 | 0.158611±0.000181 | 0.037166±0.000123 | 0.159948±0.000182 | 0.031240±0.000103 | 0.146646±0.000161 |
| FedFormer | 0.030666±0.001180 | 0.142679±0.003541 | 0.030673±0.001393 | 0.144602±0.003972 | 0.030622±0.001041 | 0.144370±0.002998 | 0.030930±0.001210 | 0.145063±0.003453 | 0.030768±0.001120 | 0.144749±0.003182 | 0.030484±0.001126 | 0.144318±0.003240 |
| **DAF (Ours)** | 0.009940±0.000688 | 0.069225±0.003645 | 0.010663±0.000751 | 0.072877±0.004001 | 0.010569±0.000675 | 0.072200±0.003453 | 0.010569±0.000675 | 0.072251±0.003633 | 0.010624±0.000709 | 0.072348±0.003741 | 0.008162±0.000326 | 0.062966±0.002211 |
| Improvements | 13.41±41.50 | 12.63±23.00 | 10.41±26.52 | 10.60±15.15 | 11.27±26.31 | 11.08±13.13 | 11.76±25.32 | 11.47±14.64 | 11.04±25.91 | 11.12±14.80 | 20.10±22.76 | 16.56±13.37 |

| Dataset | KINEIS-16 | | KINEIS-17 | | KINEIS-18 | | KINEIS-19 | | KINEIS-20 | | KINEIS-21 | |
| --- | --- | --- | --- | --- | --- | --- | --- | --- | --- | --- | --- | --- |
| Metric | MSE | MAE | MSE | MAE | MSE | MAE | MSE | MAE | MSE | MAE | MSE | MAE |
| Time-MoE | 2.387730±0.052483 | 1.258183±0.018524 | 2.379878±0.050885 | 1.255464±0.018133 | 2.378396±0.048325 | 1.255229±0.017597 | 2.382850±0.046198 | 1.256892±0.015803 | 2.372318±0.047256 | 1.253062±0.016951 | 2.405618±0.051482 | 1.258459±0.016947 |
| Timer | 0.531787±0.007579 | 0.596496±0.004286 | 0.545044±0.006635 | 0.605053±0.004262 | 0.568956±0.006435 | 0.618873±0.004306 | 0.539971±0.006799 | 0.601000±0.004306 | 0.581137±0.006394 | 0.625905±0.003865 | 0.522920±0.006018 | 0.589108±0.004007 |
| Timer-XL | 1.445647±0.239502 | 1.011559±0.086811 | 1.451451±0.236548 | 1.013682±0.085757 | 1.446432±0.235506 | 1.017698±0.085303 | 1.470487±0.236557 | 1.01914±0.085431 | 1.470487±0.236557 | 1.020677±0.085214 | 1.455367±0.222246 | 1.010943±0.079878 |
| Time-LLM | 1.196834±0.115026 | 0.905283±0.029973 | 1.195276±0.112909 | 0.904786±0.029241 | 1.192712±0.105655 | 0.903653±0.026982 | 1.197824±0.114214 | 0.905523±0.030017 | 1.187813±0.100653 | 0.901656±0.025325 | 1.202747±0.113533 | 0.903883±0.029581 |
| TTM | 0.960597±0.573263 | 0.764356±0.317300 | 0.949785±0.567021 | 0.759748±0.315918 | 0.935472±0.558760 | 0.754067±0.313746 | 0.954550±0.569180 | 0.762076±0.315998 | 0.927351±0.555230 | 0.750294±0.313646 | 1.014682±0.601253 | 0.782446±0.320897 |
| TimesFM | 0.026047±0.001071 | 0.129805±0.002998 | 0.024626±0.000923 | 0.126081±0.002714 | 0.022594±0.000882 | 0.120894±0.002620 | 0.025477±0.000997 | 0.128157±0.002829 | 0.021556±0.000854 | 0.117968±0.002654 | 0.026050±0.001492 | 0.129956±0.003770 |
| MOMENT | 0.142370±0.003197 | 0.228052±0.005981 | 0.141470±0.003216 | 0.225494±0.005802 | 0.137945±0.001924 | 0.220985±0.004786 | 0.141093±0.003517 | 0.226670±0.006252 | 0.138591±0.003160 | 0.221060±0.004620 | 0.142105±0.003160 | 0.227017±0.005908 |
| MOIRAI | 0.085563±0.032121 | 0.213621±0.048856 | 0.084417±0.048856 | 0.212000±0.049152 | 0.086500±0.032506 | 0.209514±0.048694 | 0.085336±0.032342 | 0.213290±0.049494 | 0.081236±0.030951 | 0.208038±0.048243 | 0.087631±0.032708 | 0.215668±0.049261 |
| GRU | 0.009953±0.003845 | 0.073794±0.013791 | 0.009650±0.003774 | 0.072686±0.013802 | 0.009143±0.003684 | 0.070841±0.013856 | 0.009918±0.003826 | 0.073585±0.013825 | 0.008828±0.003607 | 0.069827±0.013831 | 0.011956±0.004462 | 0.081472±0.016288 |
| LSTM | 0.017312±0.001153 | 0.099052±0.003711 | 0.016958±0.001121 | 0.098082±0.003727 | 0.016354±0.001121 | 0.096321±0.003808 | 0.017292±0.001097 | 0.098972±0.003653 | 0.015981±0.001097 | 0.095286±0.003809 | 0.017277±0.002488 | 0.099031±0.006800 |
| DLinear | 0.018501±0.000372 | 0.102200±0.001034 | 0.017924±0.000356 | 0.100625±0.001006 | 0.016868±0.000338 | 0.097638±0.000984 | 0.018281±0.000365 | 0.101513±0.001027 | 0.016396±0.000324 | 0.096258±0.000960 | 0.019496±0.000398 | 0.103591±0.001057 |
| SparseTSF | 0.035563±0.000119 | 0.157805±0.000178 | 0.034267±0.000176 | 0.154784±0.000176 | 0.032001±0.000106 | 0.149755±0.001006 | 0.034605±0.000106 | 0.155543±0.000165 | 0.031036±0.000104 | 0.147460±0.000165 | 0.035834±0.000123 | 0.157087±0.000184 |
| FedFormer | 0.030398±0.001008 | 0.145157±0.003134 | 0.030381±0.001058 | 0.145145±0.003276 | 0.030450±0.001081 | 0.145396±0.003305 | 0.030641±0.001106 | 0.145730±0.003297 | 0.030438±0.001081 | 0.145463±0.003310 | 0.030600±0.001018 | 0.144494±0.002912 |
| **DAF (Ours)** | 0.009594±0.000618 | 0.069045±0.003398 | 0.009112±0.000521 | 0.067133±0.003003 | 0.008421±0.000374 | 0.064473±0.002445 | 0.009510±0.000545 | 0.068670±0.003158 | 0.008097±0.000308 | 0.063101±0.002181 | 0.010108±0.000671 | 0.070072±0.003475 |
| Improvements | 3.61±43.45 | 6.44±22.09 | 5.58±42.33 | 7.64±21.67 | 7.90±41.20 | 8.99±21.25 | 4.11±42.48 | 6.68±21.82 | 8.28±40.96 | 9.62±21.03 | 16.21±36.88 | 13.99±20.40 |

| Dataset | KINEIS-22 | | KINEIS-23 | | KINEIS-24 | | KINEIS-25 | | KUIPER-1 | | KUIPER-2 | |
| --- | --- | --- | --- | --- | --- | --- | --- | --- | --- | --- | --- | --- |
| Metric | MSE | MAE | MSE | MAE | MSE | MAE | MSE | MAE | MSE | MAE | MSE | MAE |
| Time-MoE | 2.400384±0.050179 | 1.256636±0.016727 | 2.399712±0.042603 | 1.257537±0.015455 | 2.402853±0.048731 | 1.258518±0.016417 | 2.403575±0.043778 | 1.258316±0.015160 | 2.142842±0.029724 | 1.199049±0.010721 | 2.227838±0.014225 | 1.255281±0.003126 |
| Timer | 0.530670±0.006512 | 0.593988±0.004196 | 0.530918±0.007581 | 0.593847±0.004373 | 0.528343±0.006860 | 0.592479±0.004239 | 0.528785±0.007454 | 0.592510±0.004544 | 2.622731±0.004970 | 1.406423±0.001293 | 1.642149±0.001950 | 1.131892±0.001138 |
| Timer-XL | 1.458226±0.218606 | 1.012321±0.079540 | 1.458638±0.221922 | 1.012072±0.079580 | 1.456067±0.217128 | 1.011660±0.078006 | 1.456556±0.219015 | 1.011375±0.078621 | 2.356695±0.014509 | 1.335873±0.002308 | 1.912373±0.124859 | 1.227730±0.042284 |
| Time-LLM | 1.196454±0.114122 | 0.902268±0.029540 | 1.199938±0.113719 | 0.903027±0.030011 | 1.197815±0.115282 | 0.902457±0.030108 | 1.198718±0.114169 | 0.902853±0.029476 | 0.933171±0.169258 | 0.810843±0.095453 | 1.023213±0.078012 | 0.874512±0.049779 |
| TTM | 1.008119±0.597415 | 0.780446±0.320086 | 1.010021±0.598374 | 0.781041±0.320358 | 1.011014±0.598952 | 0.781041±0.320418 | 1.010798±0.598326 | 0.781150±0.320418 | 0.051123±0.024376 | 0.157849±0.051791 | 0.350918±0.210680 | 0.473782±0.320313 |
| TimesFM | 0.025273±0.001460 | 0.127951±0.003802 | 0.025492±0.001315 | 0.128509±0.003359 | 0.025278±0.001414 | 0.127990±0.003355 | 0.025679±0.001398 | 0.128919±0.003541 | 0.007371±0.000082 | 0.035875±0.000374 | 0.001361±0.000201 | 0.028372±0.001777 |
| MOMENT | 0.141916±0.002947 | 0.226308±0.005678 | 0.141600±0.003034 | 0.226105±0.005936 | 0.140845±0.005283 | 0.225577±0.005889 | 0.141874±0.003205 | 0.226316±0.006027 | 0.136160±0.002582 | 0.221988±0.004470 | 0.137692±0.002038 | 0.213548±0.004369 |
| MOIRAI | 0.086650±0.032336 | 0.214767±0.048905 | 0.086725±0.032511 | 0.214620±0.049085 | 0.086590±0.032506 | 0.214529±0.049149 | 0.087473±0.032935 | 0.215506±0.049494 | 0.014549±0.002137 | 0.075572±0.009406 | 0.022951±0.009969 | 0.115668±0.028427 |
| GRU | 0.011786±0.004398 | 0.080897±0.015226 | 0.011818±0.004401 | 0.080938±0.015197 | 0.011631±0.004402 | 0.080480±0.015197 | 0.011970±0.004423 | 0.081333±0.015266 | 0.000903±0.000223 | 0.021640±0.002496 | 0.001234±0.000402 | 0.025053±0.002831 |
| LSTM | 0.017048±0.002430 | 0.098449±0.006780 | 0.017095±0.002402 | 0.098573±0.006671 | 0.016634±0.001121 | 0.098050±0.006659 | 0.017216±0.002441 | 0.098898±0.006705 | 0.000779±0.000263 | 0.019827±0.003981 | 0.001952±0.000924 | 0.031011±0.008114 |
| DLinear | 0.019107±0.000388 | 0.102565±0.001036 | 0.019173±0.000388 | 0.102738±0.001035 | 0.019060±0.000391 | 0.102450±0.001042 | 0.019328±0.000390 | 0.103118±0.001035 | 0.000254±0.000001 | 0.010912±0.000039 | 0.000612±0.000002 | 0.016035±0.000096 |
| SparseTSF | 0.035031±0.000119 | 0.155533±0.000178 | 0.035126±0.000118 | 0.155597±0.000176 | 0.035300±0.000119 | 0.156100±0.000172 | 0.035767±0.001019 | 0.155897±0.000179 | 0.000510±0.000008 | 0.016054±0.000022 | 0.000636±0.000022 | 0.020646±0.000022 |
| FedFormer | 0.030618±0.000986 | 0.144599±0.002888 | 0.030822±0.001067 | 0.145014±0.003160 | 0.030438±0.003160 | 0.144157±0.002841 | 0.030767±0.001296 | 0.144853±0.003008 | 0.029767±0.001296 | 0.149746±0.003017 | 0.036021±0.002380 | 0.169230±0.005500 |
| **DAF (Ours)** | 0.009738±0.000600 | 0.069058±0.003268 | 0.009799±0.000566 | 0.069304±0.003215 | 0.008864±0.000374 | 0.064473±0.002306 | 0.009946±0.000608 | 0.069811±0.003277 | 0.000150±0.000006 | 0.008216±0.000294 | 0.000409±0.000020 | 0.014900±0.000283 |
| Improvements | 17.38±35.92 | 14.63±20.11 | 17.08±35.67 | 14.37±20.05 | 16.92±36.67 | 14.43±20.31 | 16.49±36.17 | 14.17±20.14 | 40.94±4.59 | 24.71±2.96 | 33.17±3.49 | 7.08±2.32 |

| Dataset | KUIPER-3 | | KUIPER-4 | | KUIPER-5 | | KUIPER-6 | | KUIPER-7 | | KUIPER-8 | |
| --- | --- | --- | --- | --- | --- | --- | --- | --- | --- | --- | --- | --- |
| Metric | MSE | MAE | MSE | MAE | MSE | MAE | MSE | MAE | MSE | MAE | MSE | MAE |
| Time-MoE | 2.210636±0.015550 | 1.246138±0.003782 | 2.206669±0.013596 | 1.245057±0.002588 | 2.208527±0.018218 | 1.245318±0.004233 | 2.207466±0.013596 | 1.245967±0.002774 | 2.205820±0.009962 | 1.244857±0.001507 | 2.221887±0.019065 | 1.251593±0.005987 |
| Timer | 1.850399±0.002359 | 1.204729±0.000719 | 1.856490±0.001730 | 1.207421±0.000067 | 1.845935±0.001730 | 1.203276±0.001106 | 1.843852±0.001580 | 1.202623±0.000539 | 1.829886±0.001992 | 1.198126±0.000707 | 1.797214±0.002805 | 1.186963±0.000922 |
| Timer-XL | 1.993152±0.097530 | 1.253866±0.003951 | 1.992043±0.095337 | 1.253700±0.032197 | 1.992977±0.100170 | 1.254088±0.033734 | 1.990244±0.095455 | 1.253015±0.032395 | 1.988657±0.101411 | 1.252436±0.033918 | 1.971763±0.105120 | 1.247032±0.035522 |
| Time-LLM | 1.008927±0.107878 | 0.865988±0.063296 | 0.999116±0.106281 | 0.862710±0.062841 | 1.002369±0.106404 | 0.864083±0.063537 | 0.999022±0.104094 | 0.862810±0.062866 | 1.005802±0.105243 | 0.865362±0.062541 | 1.010348±0.100684 | 0.867392±0.060072 |
| TTM | 0.260658±0.154606 | 0.409602±0.161732 | 0.258716±0.153077 | 0.408361±0.168583 | 0.261634±0.155306 | 0.410359±0.170199 | 0.262221±0.155355 | 0.411124±0.169945 | 0.269513±0.160063 | 0.416419±0.172947 | 0.283335±0.168601 | 0.427391±0.178143 |
| TimesFM | 0.001311±0.000167 | 0.028032±0.001459 | 0.001414±0.000172 | 0.028326±0.001502 | 0.001319±0.000191 | 0.028006±0.001616 | 0.001359±0.000171 | 0.028390±0.001504 | 0.001350±0.000188 | 0.028310±0.001623 | 0.001408±0.000158 | 0.028784±0.001374 |
| MOMENT | 0.139449±0.002820 | 0.218217±0.005320 | 0.136619±0.003204 | 0.212311±0.004479 | 0.136279±0.003094 | 0.212457±0.005495 | 0.143147±0.003047 | 0.224229±0.005882 | 0.142569±0.003129 | 0.224388±0.005875 | 0.144781±0.003402 | 0.223041±0.005949 |
| MOIRAI | 0.017866±0.007240 | 0.103060±0.023547 | 0.018012±0.007391 | 0.103271±0.023826 | 0.017905±0.007371 | 0.103151±0.023826 | 0.018061±0.007416 | 0.103517±0.023963 | 0.018385±0.007579 | 0.104388±0.024288 | 0.019425±0.008114 | 0.107130±0.025211 |
| GRU | 0.000940±0.000335 | 0.021127±0.003237 | 0.001089±0.000348 | 0.022279±0.003324 | 0.000955±0.000334 | 0.021324±0.003247 | 0.001033±0.000325 | 0.021982±0.003239 | 0.001011±0.000328 | 0.021749±0.003266 | 0.001207±0.000362 | 0.023318±0.003195 |
| LSTM | 0.001458±0.000704 | 0.026534±0.007459 | 0.001624±0.000509 | 0.027317±0.007509 | 0.001624±0.000692 | 0.026633±0.007517 | 0.001545±0.000729 | 0.027233±0.007509 | 0.001545±0.000729 | 0.027262±0.007599 | 0.001775±0.000777 | 0.028868±0.007718 |
| DLinear | 0.000176±0.000001 | 0.007897±0.000056 | 0.000276±0.000002 | 0.008953±0.000055 | 0.000169±0.000001 | 0.007978±0.000049 | 0.000210±0.000001 | 0.008561±0.000049 | 0.000221±0.000001 | 0.008811±0.000056 | 0.000325±0.000002 | 0.010449±0.000067 |
| SparseTSF | 0.000215±0.000002 | 0.008465±0.000123 | 0.000283±0.000000 | 0.009257±0.000112 | 0.000261±0.000000 | 0.008863±0.000119 | 0.000278±0.000000 | 0.008643±0.000119 | 0.000278±0.000001 | 0.009510±0.000108 | 0.000318±0.000001 | 0.010351±0.000054 |
| FedFormer | 0.035759±0.002210 | 0.168619±0.005129 | 0.035793±0.002204 | 0.168564±0.005183 | 0.035767±0.002227 | 0.168622±0.005183 | 0.035846±0.002227 | 0.168691±0.005077 | 0.035846±0.002227 | 0.168832±0.005180 | 0.035982±0.002279 | 0.169144±0.005288 |
| **DAF (Ours)** | 0.000237±0.000003 | 0.011098±0.000134 | 0.000317±0.000003 | 0.012031±0.000126 | 0.000260±0.000001 | 0.011163±0.000127 | 0.000269±0.000001 | 0.011591±0.000137 | 0.000269±0.000001 | 0.011731±0.000151 | 0.000356±0.000005 | 0.012960±0.000162 |
| Improvements | -35.50±5.17 | - | - | - | - | - | - | - | -21.72±1.00 | - | - | - |

Table 16: Zero-shot evaluation on individual satellites of KUIPER dataset with mean and standard deviation in MSE and MAE. "Improvements" presents the relative performance of our model compared to the best baseline for each dataset and metric. A dash ('-') indicates that our model fails to achieve either the best or second-best performance. The best result is highlighted in **bold**, and the second-best is highlighted with underline.

### KUIPER-9 – KUIPER-14

| Dataset / Metric | KUIPER-9 MSE | KUIPER-9 MAE | KUIPER-10 MSE | KUIPER-10 MAE | KUIPER-11 MSE | KUIPER-11 MAE | KUIPER-12 MSE | KUIPER-12 MAE | KUIPER-13 MSE | KUIPER-13 MAE | KUIPER-14 MSE | KUIPER-14 MAE |
|---|---|---|---|---|---|---|---|---|---|---|---|---|
| Time-MoE | 2.245668±0.021669 | 1.262751±0.008367 | 2.217046±0.018026 | 1.249779±0.004242 | 2.208631±0.014748 | 1.245839±0.003894 | 2.209209±0.014298 | 1.246638±0.003565 | 2.206898±0.013469 | 1.246479±0.003161 | 2.209318±0.013440 | 1.246078±0.003596 |
| Timer | 1.504165±0.004615 | 1.080581±0.001926 | 1.758607±0.001654 | 1.172980±0.001014 | 1.877573±0.002270 | 1.213867±0.001231 | 1.811507±0.000597 | 1.214978±0.000571 | 1.861819±0.001603 | 1.208409±0.000696 | 1.852729±0.002040 | 1.205294±0.000859 |
| Timer-XL | 1.855342±0.142344 | 1.208363±0.048576 | 1.958157±0.108473 | 1.242575±0.036762 | 2.007977±0.098592 | 1.258374±0.032952 | 2.007352±0.009625 | 1.258405±0.032534 | 2.001750±0.101694 | 1.256518±0.033918 | 1.994316±0.095629 | 1.254105±0.032450 |
| Time-LLM | 1.044472±0.060315 | 0.883323±0.041063 | 1.013080±0.095993 | 0.869002±0.058308 | 1.004978±0.110235 | 0.864196±0.064778 | 1.002428±0.110485 | 0.863147±0.064963 | 1.001494±0.109394 | 0.863138±0.065023 | 1.001337±0.10831 | 0.863296±0.064443 |
| TTM | 0.416009±0.250759 | 0.515545±0.219370 | 0.298784±0.178557 | 0.437555±0.183616 | 0.250008±0.148044 | 0.400749±0.165706 | 0.248718±0.146651 | 0.400423±0.164173 | 0.256064±0.151578 | 0.406151±0.167577 | 0.259188±0.153555 | 0.408684±0.169041 |
| TimesFM | 0.001410±0.000158 | 0.028814±0.001546 | 0.001326±0.000180 | 0.028135±0.001525 | 0.001384±0.000182 | 0.028583±0.001517 | 0.001373±0.000185 | 0.028470±0.001558 | 0.001343±0.000175 | 0.028226±0.001462 | 0.001391±0.000184 | 0.028663±0.001663 |
| MOMENT | 0.115508±0.001528 | 0.182418±0.002638 | 0.113885±0.000923 | 0.176899±0.002009 | 0.112749±0.000433 | 0.174497±0.001891 | 0.112840±0.000307 | 0.174127±0.002307 | 0.113189±0.001335 | 0.174899±0.002453 | 0.112274±0.000797 | 0.174718±0.001406 |
| MOIRAI | 0.027480±0.012148 | 0.125646±0.031647 | 0.020042±0.008491 | 0.108539±0.025857 | 0.017568±0.007127 | 0.102174±0.023308 | 0.017575±0.007114 | 0.102119±0.023248 | 0.017758±0.007231 | 0.102595±0.023511 | 0.018053±0.007444 | 0.103440±0.024009 |
| GRU | 0.001751±0.000422 | 0.030019±0.002422 | 0.001054±0.000349 | 0.022572±0.003053 | 0.001037±0.000341 | 0.021835±0.003426 | 0.001004±0.000311 | 0.021469±0.003470 | 0.000991±0.000351 | 0.021487±0.003389 | 0.001030±0.000332 | 0.022020±0.003293 |
| LSTM | 0.002610±0.001038 | 0.036192±0.008114 | 0.001480±0.000676 | 0.026335±0.007557 | 0.001526±0.000680 | 0.027114±0.007343 | 0.001472±0.000660 | 0.026703±0.007278 | 0.001530±0.000700 | 0.026804±0.007404 | 0.001551±0.000694 | 0.027411±0.007457 |
| DLinear | 0.001277±0.000014 | 0.024139±0.000178 | 0.000296±0.000003 | 0.010843±0.000080 | 0.000240±0.000001 | **0.008558±0.000051** | **0.000213±0.000001** | **0.008156±0.000051** | 0.000209±0.000001 | **0.008409±0.000054** | 0.000259±0.000001 | **0.008989±0.000051** |
| SparseTSF | 0.001812±0.000029 | 0.031151±0.000306 | 0.000430±0.000002 | 0.012977±0.000049 | 0.000310±0.000002 | 0.009764±0.000150 | 0.000273±0.000003 | 0.009421±0.000115 | **0.000251±0.000002** | 0.008519±0.000049 | 0.000391±0.000002 | 0.009391±0.000109 |
| FedFormer | 0.035730±0.002514 | 0.168405±0.005838 | 0.035965±0.002312 | 0.169078±0.005354 | 0.035635±0.002174 | 0.168330±0.005058 | 0.035718±0.002184 | 0.168258±0.004970 | 0.035727±0.002184 | 0.168555±0.005082 | 0.035738±0.002177 | 0.168576±0.005054 |
| DAF (Ours) | **0.000683±0.000039** | **0.019013±0.000378** | **0.000291±0.000003** | **0.012535±0.000208** | **0.000264±0.000003** | 0.011649±0.000063 | 0.000256±0.000002 | 0.011306±0.000078 | 0.000256±0.000002 | 0.011428±0.000123 | 0.000288±0.000003 | 0.011877±0.000121 |
| Improvements | 46.52±3.64 | 21.24±2.15 | 1.69±2.01 | -15.60±2.77 | -18.33±1.74 | - | -23.47±1.05 | - | -22.49±1.54 | - | - | - |

### KUIPER-15 – KUIPER-20

| Dataset / Metric | KUIPER-15 MSE | KUIPER-15 MAE | KUIPER-16 MSE | KUIPER-16 MAE | KUIPER-17 MSE | KUIPER-17 MAE | KUIPER-18 MSE | KUIPER-18 MAE | KUIPER-19 MSE | KUIPER-19 MAE | KUIPER-20 MSE | KUIPER-20 MAE |
|---|---|---|---|---|---|---|---|---|---|---|---|---|
| Time-MoE | 2.206366±0.013321 | 1.244876±0.002756 | 2.198285±0.008841 | 1.241777±0.001319 | 2.207904±0.015643 | 1.245192±0.003516 | 2.203300±0.017864 | 1.243767±0.002417 | 2.201731±0.012435 | 1.243767±0.002417 | 2.206026±0.014219 | 1.244998±0.002693 |
| Timer | 1.842509±0.002412 | 1.201817±0.001146 | 1.907691±0.002429 | 1.223502±0.000844 | 1.874222±0.001915 | 1.212815±0.000914 | 1.898146±0.001924 | 1.220278±0.000843 | 1.895577±0.000812 | 1.219732±0.000843 | 1.877975±0.002127 | 1.214216±0.001079 |
| Timer-XL | 1.991412±0.100731 | 1.254429±0.033774 | 2.020642±0.098785 | 1.263046±0.033104 | 2.008239±0.102437 | 1.258420±0.034219 | 2.017665±0.096528 | 1.261870±0.034007 | 2.012786±0.101257 | 1.261703±0.034007 | 2.003494±0.094956 | 1.257118±0.032001 |
| Time-LLM | 1.003338±0.106084 | 0.864225±0.063347 | 0.997125±0.114040 | 0.861177±0.067101 | 1.002028±0.110813 | 0.863277±0.065065 | 0.996941±0.111745 | 0.861163±0.067033 | 0.996941±0.111745 | 0.861163±0.067033 | 0.998622±0.110699 | 0.861608±0.065730 |
| TTM | 0.264644±0.157101 | 0.412749±0.171141 | 0.238366±0.140507 | 0.391605±0.160599 | 0.246074±0.145262 | 0.397987±0.163509 | 0.241520±0.142497 | 0.394063±0.161765 | 0.241520±0.142497 | 0.394063±0.162397 | 0.249601±0.147543 | 0.400871±0.165070 |
| TimesFM | 0.001336±0.000171 | 0.028191±0.001470 | 0.001379±0.000158 | 0.028638±0.001254 | 0.001481±0.000207 | 0.029121±0.001556 | 0.001392±0.000182 | 0.028692±0.001561 | 0.001392±0.000182 | 0.028692±0.001561 | 0.001408±0.000179 | 0.028779±0.001600 |
| MOMENT | 0.113542±0.001503 | 0.175466±0.002275 | 0.112386±0.001035 | 0.173843±0.002029 | 0.114558±0.001303 | 0.175501±0.002453 | 0.112970±0.000721 | 0.174665±0.001900 | 0.112692±0.000863 | 0.174778±0.002266 | 0.112752±0.001422 | 0.174602±0.002620 |
| MOIRAI | 0.018124±0.008060 | 0.103722±0.023837 | 0.017119±0.006894 | 0.100947±0.022670 | 0.017716±0.007139 | 0.101402±0.023246 | 0.017295±0.006900 | 0.101355±0.022804 | 0.017064±0.006897 | 0.100778±0.022729 | 0.017522±0.007079 | 0.101973±0.023158 |
| GRU | 0.001008±0.000334 | 0.021745±0.003260 | 0.001000±0.000333 | 0.021182±0.003486 | 0.001080±0.000345 | 0.022106±0.003361 | 0.001044±0.000348 | 0.021709±0.003476 | 0.000932±0.000347 | 0.021032±0.003505 | 0.001048±0.000335 | 0.021878±0.003420 |
| LSTM | 0.001539±0.000717 | 0.027134±0.007555 | 0.001419±0.000658 | 0.026335±0.007230 | 0.001617±0.000716 | 0.027612±0.007392 | 0.001530±0.000663 | 0.026853±0.007180 | 0.001420±0.000663 | 0.026230±0.007210 | 0.001540±0.000670 | 0.027112±0.007304 |
| DLinear | **0.000220±0.000001** | **0.008557±0.000054** | **0.000202±0.000001** | **0.007892±0.000049** | 0.000297±0.000000 | **0.009013±0.000041** | 0.000251±0.000001 | **0.008519±0.000049** | **0.000174±0.000001** | **0.007734±0.000053** | 0.000242±0.000001 | **0.008578±0.000054** |
| SparseTSF | 0.000271±0.000002 | 0.009271±0.000103 | 0.000317±0.000003 | 0.010310±0.000171 | 0.000271±0.000002 | 0.009187±0.000144 | 0.000287±0.000003 | 0.009893±0.000163 | 0.000251±0.000001 | 0.009070±0.000163 | **0.000221±0.000002** | 0.008813±0.000138 |
| FedFormer | 0.035824±0.002220 | 0.168753±0.005155 | 0.035500±0.002094 | 0.167975±0.004888 | 0.035723±0.002202 | 0.168505±0.005109 | 0.035533±0.002130 | 0.168056±0.004966 | 0.035533±0.002130 | 0.168049±0.005000 | 0.035636±0.002163 | 0.168327±0.005035 |
| DAF (Ours) | 0.000267±0.000003 | 0.011626±0.000139 | 0.000254±0.000002 | 0.011036±0.000159 | 0.000386±0.000015 | 0.012389±0.000135 | 0.000292±0.000003 | 0.011421±0.000097 | 0.000292±0.000002 | 0.011421±0.000097 | 0.000292±0.000003 | 0.011686±0.000076 |
| Improvements | -21.36±1.92 | - | -25.74±1.61 | - | -18.33±1.74 | - | -22.41±1.85 | - | -11.16±1.24 | - | -22.49±1.54 | - |

### KUIPER-21 – KUIPER-26

| Dataset / Metric | KUIPER-21 MSE | KUIPER-21 MAE | KUIPER-22 MSE | KUIPER-22 MAE | KUIPER-23 MSE | KUIPER-23 MAE | KUIPER-24 MSE | KUIPER-24 MAE | KUIPER-25 MSE | KUIPER-25 MAE | KUIPER-26 MSE | KUIPER-26 MAE |
|---|---|---|---|---|---|---|---|---|---|---|---|---|
| Time-MoE | 2.213601±0.013999 | 1.247784±0.002490 | 2.197686±0.011044 | 1.240770±0.002270 | 2.203901±0.012016 | 1.244178±0.002169 | 2.265656±0.024964 | 1.269801±0.008270 | 2.199393±0.012373 | 1.241355±0.002506 | 2.221019±0.018380 | 1.251185±0.003791 |
| Timer | 1.799109±0.003992 | 1.187042±0.001472 | 1.895177±0.001442 | 1.219431±0.000662 | 1.884105±0.002008 | 1.215763±0.000981 | 1.363698±0.005957 | 1.018203±0.002609 | 1.896109±0.001634 | 1.219988±0.000536 | 1.708200±0.002980 | 1.155259±0.001097 |
| Timer-XL | 1.977905±0.113488 | 1.248772±0.037813 | 2.011879±0.094494 | 1.260123±0.031677 | 2.011226±0.098757 | 1.259357±0.033180 | 1.797324±0.156836 | 1.188209±0.054933 | 2.012398±0.093625 | 1.260102±0.031408 | 1.940941±0.117685 | 1.237029±0.039555 |
| Time-LLM | 1.005698±0.099780 | 0.866097±0.060173 | 1.003090±0.112717 | 0.863078±0.066128 | 1.003865±0.113232 | 0.863631±0.066123 | 1.065304±0.036352 | 0.891668±0.030366 | 1.002718±0.114447 | 0.862819±0.066725 | 1.017023±0.088187 | 0.871209±0.054518 |
| TTM | 0.284042±0.169101 | 0.426962±0.177941 | 0.242936±0.143476 | 0.395265±0.162505 | 0.250074±0.145262 | 0.399987±0.163509 | 0.491432±0.296891 | 0.554884±0.236570 | 0.242044±0.142822 | 0.394688±0.161998 | 0.321897±0.192889 | 0.453351±0.191073 |
| TimesFM | 0.001381±0.000184 | 0.028596±0.001497 | 0.001322±0.000176 | 0.028099±0.001515 | 0.001357±0.000185 | 0.028383±0.001595 | 0.002858±0.000041 | 0.038299±0.000697 | 0.001344±0.000176 | 0.028285±0.001494 | 0.001375±0.000206 | 0.028523±0.001712 |
| MOMENT | 0.113563±0.001567 | 0.176364±0.003071 | 0.113042±0.000297 | 0.173871±0.001484 | 0.112512±0.001960 | 0.174069±0.002761 | 0.117949±0.001255 | 0.188114±0.002516 | 0.113862±0.001665 | 0.174733±0.002594 | 0.114181±0.000296 | 0.177886±0.000961 |
| MOIRAI | 0.019224±0.008060 | 0.106447±0.025184 | 0.017092±0.006894 | 0.108590±0.022877 | 0.017296±0.007001 | 0.101402±0.023004 | 0.040488±0.014886 | 0.137762±0.034526 | 0.017228±0.007038 | 0.101157±0.023248 | 0.021339±0.009089 | 0.113713±0.026796 |
| GRU | 0.001049±0.000326 | 0.022443±0.003121 | 0.000931±0.000328 | 0.021103±0.003406 | 0.001027±0.000365 | 0.021862±0.003494 | 0.002232±0.000383 | 0.034135±0.001732 | 0.000987±0.000353 | 0.021342±0.003515 | 0.001178±0.000391 | 0.023970±0.003138 |
| LSTM | 0.001623±0.000748 | 0.028230±0.007794 | 0.001419±0.000658 | 0.026384±0.007230 | 0.001545±0.000696 | 0.027123±0.007274 | 0.003427±0.001136 | 0.041218±0.007418 | 0.001485±0.000670 | 0.026506±0.007190 | 0.001858±0.000866 | 0.029959±0.007935 |
| DLinear | 0.000286±0.000003 | 0.010093±0.000073 | **0.000168±0.000001** | **0.007554±0.000042** | 0.000213±0.000001 | **0.008332±0.000046** | 0.002589±0.000271 | 0.032382±0.000271 | 0.000211±0.000001 | **0.008125±0.000046** | 0.000454±0.000005 | **0.013528±0.000099** |
| SparseTSF | **0.000291±0.000001** | 0.010813±0.000071 | 0.000324±0.000003 | 0.010208±0.000151 | 0.000324±0.000002 | 0.009592±0.000142 | 0.004607±0.000016 | 0.046755±0.000058 | **0.000250±0.000002** | 0.009067±0.000146 | 0.000707±0.000002 | 0.017619±0.000036 |
| FedFormer | 0.035918±0.002229 | 0.168980±0.005175 | 0.035570±0.002140 | 0.168143±0.004990 | 0.035626±0.002139 | 0.168293±0.004995 | 0.034598±0.002105 | 0.165216±0.004923 | 0.035541±0.002140 | 0.168089±0.004991 | 0.035939±0.002333 | 0.169040±0.005406 |
| DAF (Ours) | 0.000297±0.000000 | 0.012339±0.000177 | 0.000240±0.000003 | 0.010819±0.000101 | 0.000386±0.000003 | 0.011398±0.000070 | **0.001090±0.000038** | **0.023001±0.000302** | 0.000250±0.000000 | 0.011146±0.000098 | **0.000371±0.000009** | 0.013897±0.000192 |
| Improvements | - | - | -33.33±2.58 | - | -22.07±1.51 | - | 57.90±2.15 | 29.92±1.50 | -18.48±0.56 | - | 18.28±2.88 | -2.73±2.17 |

### KUIPER-27 – KUIPER-32

| Dataset / Metric | KUIPER-27 MSE | KUIPER-27 MAE | KUIPER-28 MSE | KUIPER-28 MAE | KUIPER-29 MSE | KUIPER-29 MAE | KUIPER-30 MSE | KUIPER-30 MAE | KUIPER-31 MSE | KUIPER-31 MAE | KUIPER-32 MSE | KUIPER-32 MAE |
|---|---|---|---|---|---|---|---|---|---|---|---|---|
| Time-MoE | 2.222980±0.017613 | 1.252823±0.004577 | 2.203181±0.012325 | 1.243558±0.004362 | 2.208760±0.020284 | 1.243421±0.002236 | 2.211896±0.023421 | 1.244062±0.006686 | 2.203848±0.015914 | 1.241553±0.004362 | 2.213799±0.015811 | 1.244761±0.004345 |
| Timer | 1.722361±0.001796 | 1.160339±0.000444 | 1.890464±0.003094 | 1.218163±0.001010 | 1.839956±0.005248 | 1.200057±0.001197 | 1.821617±0.003061 | 1.194339±0.000749 | 1.818480±0.002753 | 1.193549±0.001010 | 1.837748±0.003452 | 1.199751±0.000853 |
| Timer-XL | 1.941068±0.113762 | 1.237168±0.038461 | 2.007989±0.089654 | 1.258605±0.035833 | 1.936202±0.108159 | 1.235223±0.036976 | 1.925794±0.100157 | 1.232256±0.034438 | 1.930270±0.105779 | 1.233665±0.035833 | 1.932497±0.096856 | 1.234935±0.033453 |
| Time-LLM | 1.022363±0.088697 | 0.872414±0.054010 | 0.991314±0.111203 | 0.859461±0.066596 | 1.002305±0.108240 | 0.862756±0.063931 | 1.007430±0.100335 | 0.865674±0.061017 | 1.006141±0.096094 | 0.864838±0.058980 | 1.014524±0.107734 | 0.867063±0.063219 |
| TTM | 0.316552±0.189395 | 0.449657±0.188818 | 0.244926±0.145352 | 0.397113±0.162663 | 0.269590±0.159249 | 0.415231±0.169093 | 0.278248±0.165010 | 0.421922±0.172829 | 0.276659±0.163748 | 0.420458±0.171544 | 0.272239±0.161506 | 0.416743±0.171051 |
| TimesFM | 0.001442±0.000185 | 0.029162±0.001563 | 0.001371±0.000165 | 0.028553±0.001398 | 0.001379±0.000200 | 0.028258±0.001503 | 0.001370±0.000226 | 0.028308±0.001758 | 0.001411±0.000236 | 0.028569±0.001775 | 0.001374±0.000237 | 0.028243±0.001917 |
| MOMENT | 0.114201±0.001075 | 0.177658±0.001926 | 0.113325±0.001571 | 0.175387±0.001961 | 0.113677±0.002914 | 0.175387±0.001961 | 0.112744±0.001970 | 0.174729±0.003713 | 0.114881±0.002734 | 0.175877±0.004095 | 0.111243±0.001936 | 0.173642±0.002314 |
| MOIRAI | 0.021227±0.009085 | 0.111350±0.026800 | 0.017279±0.007026 | 0.101330±0.023097 | 0.018539±0.007584 | 0.105011±0.024166 | 0.019065±0.008043 | 0.106255±0.025198 | 0.018739±0.007632 | 0.105392±0.024137 | 0.018860±0.007781 | 0.105392±0.024601 |
| GRU | 0.001226±0.000378 | 0.024244±0.003072 | 0.000970±0.000344 | 0.021372±0.003427 | 0.000981±0.000344 | 0.022034±0.003427 | 0.000995±0.000824 | 0.022034±0.009723 | 0.001038±0.000844 | 0.022240±0.009741 | 0.000942±0.000812 | 0.021701±0.009702 |
| LSTM | 0.001904±0.000873 | 0.030096±0.007896 | 0.001460±0.000423 | 0.026624±0.004294 | 0.001236±0.000423 | 0.026624±0.004294 | 0.001303±0.000417 | 0.026497±0.004294 | 0.001303±0.000417 | 0.026497±0.004406 | 0.001210±0.000405 | 0.026180±0.004348 |
| DLinear | 0.000465±0.000004 | **0.013307±0.000084** | 0.000210±0.000000 | **0.008074±0.000038** | **0.000192±0.000000** | **0.008728±0.000047** | 0.000236±0.000000 | **0.008731±0.000064** | 0.000236±0.000001 | **0.008944±0.000054** | **0.000154±0.000001** | **0.007749±0.000046** |
| SparseTSF | 0.000567±0.000002 | 0.015867±0.000035 | 0.000231±0.000003 | 0.008591±0.000149 | 0.000366±0.000000 | 0.009955±0.000124 | 0.000480±0.000000 | 0.011204±0.000096 | 0.000480±0.000002 | 0.011204±0.000096 | 0.000441±0.000002 | 0.010962±0.000118 |
| FedFormer | 0.035927±0.002337 | 0.169004±0.005395 | 0.035609±0.002141 | 0.168247±0.004983 | 0.035713±0.002163 | 0.167968±0.004982 | 0.035834±0.002164 | 0.168284±0.004853 | 0.035834±0.002164 | 0.168128±0.004991 | 0.035766±0.002213 | 0.168106±0.005098 |
| DAF (Ours) | **0.000391±0.000012** | 0.014017±0.000194 | **0.000240±0.000003** | 0.011143±0.000065 | 0.000371±0.000001 | 0.011558±0.000041 | **0.000298±0.000006** | 0.011638±0.000094 | **0.000298±0.000000** | 0.012045±0.000032 | 0.000227±0.000005 | 0.011283±0.000007 |
| Improvements | 15.91±3.30 | -5.34±2.12 | -33.33±2.58 | - | -22.07±2.75 | - | -24.09±1.47 | - | -26.27±3.08 | - | -45.61±0.95 | - |

Table 17: Zero-shot evaluation on individual satellites of two cross-domain datasets (KUIPER and LEMUR) with mean and standard deviation in MSE and MAE. "Improvements" presents the relative performance of our model compared to the best baseline for each dataset and metric. A dash ('-') indicates that our model fails to achieve either the best or second-best performance. The best result is highlighted in **bold**, and the second-best is highlighted with underline.

| Dataset | KUIPER-33 | | KUIPER-34 | | KUIPER-35 | | KUIPER-36 | | KUIPER-37 | | KUIPER-38 | |
| Metric | MSE | MAE | MSE | MAE | MSE | MAE | MSE | MAE | MSE | MAE | MSE | MAE |
|---|---|---|---|---|---|---|---|---|---|---|---|---|
| Time-MoE | 2.142842±0.029724 | 1.199049±0.010721 | 2.227838±0.014225 | 1.255281±0.003126 | 2.210636±0.015550 | 1.246138±0.003782 | 2.206669±0.013576 | 1.245057±0.002588 | 2.208527±0.018218 | 1.245318±0.004298 | 2.207466±0.013596 | 1.245967±0.002774 |
| Timer | 2.627231±0.004970 | 1.406423±0.001293 | 1.642149±0.001950 | 1.131892±0.001138 | 1.850399±0.002359 | 1.204729±0.000719 | 1.856490±0.001730 | 1.20742±0.000067 | 1.845945±0.002907 | 1.20376±0.001106 | 1.843852±0.001580 | 1.202623±0.000539 |
| Timer-XL | 2.356695±0.014509 | 1.335873±0.002308 | 1.912337±0.124859 | 1.227730±0.042284 | 1.993152±0.097530 | 1.253865±0.032951 | 1.992043±0.095337 | 1.253700±0.032197 | 1.992977±0.100170 | 1.254088±0.033734 | 1.990244±0.095455 | 1.253015±0.032395 |
| Time-LLM | 0.933317±0.169258 | 0.810843±0.095453 | 1.023713±0.078012 | 0.874512±0.049779 | 1.008927±0.107878 | 0.865988±0.063296 | 0.999116±0.106281 | 0.862714±0.063991 | 1.002369±0.106404 | 0.864083±0.063537 | 0.999022±0.104094 | 0.862810±0.062866 |
| TTM | 0.051123±0.024376 | 0.157849±0.051791 | 0.350918±0.210680 | 0.473782±0.200313 | 0.260658±0.154606 | 0.409602±0.169732 | 0.258716±0.153077 | 0.408361±0.168583 | 0.261634±0.155306 | 0.410359±0.170199 | 0.262221±0.155355 | 0.411124±0.169945 |
| TimesFM | 0.001429±0.000241 | 0.028685±0.001774 | 0.001485±0.000214 | 0.029266±0.001658 | 0.001411±0.000215 | 0.028529±0.001650 | 0.001476±0.000231 | 0.02907±0.001799 | 0.001433±0.000226 | 0.02812±0.001572 | 0.001384±0.000256 | 0.028110±0.001860 |
| MOMENT | 0.112034±0.001162 | 0.160461±0.000997 | 0.114813±0.000646 | 0.179405±0.002260 | 0.113014±0.000809 | 0.174541±0.001754 | 0.113433±0.000512 | 0.174949±0.001385 | 0.112434±0.001148 | 0.174544±0.002428 | 0.113360±0.000750 | 0.174964±0.002121 |
| MOIRAI | 0.014549±0.002137 | 0.075572±0.009406 | 0.022951±0.009969 | 0.115668±0.028427 | 0.017866±0.007240 | 0.103060±0.023524 | 0.01801±0.007391 | 0.10327±0.023826 | 0.017905±0.007371 | 0.10315±0.023826 | 0.018061±0.007416 | 0.103517±0.023963 |
| GRU | 0.001034±0.000832 | 0.022239±0.009723 | 0.001225±0.000893 | 0.025534±0.009765 | 0.001029±0.000837 | 0.022476±0.009744 | 0.001171±0.000868 | 0.022372±0.009698 | 0.000933±0.000774 | 0.021502±0.009329 | 0.000879±0.000783 | 0.021315±0.009638 |
| LSTM | 0.001276±0.000423 | 0.026143±0.004505 | 0.001510±0.000438 | 0.027686±0.004436 | 0.001310±0.000438 | 0.026899±0.004350 | 0.001467±0.000424 | 0.027704±0.004350 | 0.001184±0.000424 | 0.025443±0.004592 | 0.001157±0.000410 | 0.025875±0.004467 |
| DLinear | 0.000249±0.000001 | 0.009063±0.000048 | 0.000366±0.000001 | 0.010621±0.000053 | 0.000213±0.000001 | 0.008730±0.000057 | 0.000308±0.000002 | 0.009476±0.000057 | **0.000179±0.000001** | **0.008212±0.000051** | **0.000125±0.000001** | **0.007282±0.000052** |
| SparseTSF | 0.000554±0.000002 | 0.012935±0.000004 | 0.000510±0.000004 | 0.012125±0.000066 | 0.000364±0.000002 | 0.008561±0.000066 | 0.000417±0.000088 | 0.010738±0.000086 | 0.000373±0.000001 | 0.010328±0.000116 | 0.010396±0.000002 | 0.008396±0.000088 |
| FedFormer | 0.035782±0.002205 | 0.168152±0.005079 | 0.035887±0.002171 | 0.168375±0.004989 | 0.035806±0.002144 | 0.168162±0.004945 | 0.035898±0.002181 | 0.16837±0.005017 | 0.035662±0.002157 | 0.167891±0.004979 | 0.035782±0.002178 | 0.168164±0.004998 |
| DAF (Ours) | 0.000284±0.000005 | 0.011785±0.000129 | 0.000381±0.000006 | 0.013195±0.000066 | 0.000210±0.000003 | 0.011809±0.000091 | 0.000239±0.000008 | 0.012551±0.000140 | 0.000231±0.000008 | 0.011315±0.000134 | 0.000208±0.000010 | 0.011016±0.000154 |
| Improvements | -14.06±0.025 | -30.03±0.37 | -4.10±0.07 | - | -23.00±0.28 | - | -10.06±0.25 | - | -29.05±1.02 | - | -66.40±3.24 | - |

| Dataset | KUIPER-39 | | KUIPER-40 | | KUIPER-41 | | KUIPER-42 | | KUIPER-43 | | KUIPER-44 | |
| Metric | MSE | MAE | MSE | MAE | MSE | MAE | MSE | MAE | MSE | MAE | MSE | MAE |
|---|---|---|---|---|---|---|---|---|---|---|---|---|
| Time-MoE | 2.205820±0.009962 | 1.244857±0.001507 | 2.221887±0.019065 | 1.251593±0.005987 | 2.245668±0.021669 | 1.262751±0.008367 | 2.217046±0.018026 | 1.249779±0.004242 | 2.208631±0.014748 | 1.245839±0.003894 | 2.209090±0.014298 | 1.246638±0.003565 |
| Timer | 1.829886±0.001992 | 1.198126±0.000707 | 1.797214±0.002805 | 1.189663±0.000922 | 1.504165±0.004615 | 1.080581±0.001926 | 1.758607±0.001654 | 1.172980±0.001014 | 1.877573±0.002270 | 1.213867±0.001231 | 1.881507±0.000597 | 1.214978±0.000571 |
| Timer-XL | 1.988652±0.101411 | 1.252436±0.033918 | 1.971763±0.105120 | 1.247032±0.035522 | 1.958157±0.142344 | 1.208363±0.048576 | 1.942575±0.036762 | 1.242575±0.036762 | 2.007977±0.098592 | 1.258374±0.032952 | 2.017286±0.101257 | 1.258405±0.032554 |
| Time-LLM | 1.005802±0.105243 | 0.865362±0.062541 | 1.010348±0.100684 | 0.867392±0.060072 | 1.044472±0.060315 | 0.883323±0.041063 | 1.013080±0.095993 | 0.869002±0.058308 | 1.004978±0.110235 | 0.864196±0.064778 | 1.002428±0.110485 | 0.863147±0.064963 |
| TTM | 0.269513±0.160063 | 0.416419±0.172947 | 0.283335±0.168601 | 0.427391±0.178143 | 0.416009±0.250759 | 0.515545±0.219370 | 0.298784±0.178557 | 0.437555±0.183616 | 0.250008±0.148044 | 0.400749±0.165706 | 0.248718±0.146651 | 0.400423±0.164173 |
| TimesFM | 0.001428±0.000242 | 0.028544±0.001784 | 0.001392±0.000213 | 0.028447±0.001536 | 0.001433±0.000231 | 0.028737±0.001588 | 0.001447±0.000221 | 0.028805±0.001685 | 0.001509±0.000239 | 0.029331±0.001770 | 0.001428±0.000213 | 0.028762±0.001714 |
| MOMENT | 0.114586±0.000665 | 0.176354±0.002016 | 0.113203±0.001076 | 0.176289±0.001981 | 0.115508±0.001528 | 0.182418±0.002638 | 0.113885±0.000923 | 0.176899±0.002009 | 0.112749±0.000433 | 0.174497±0.001891 | 0.112840±0.001028 | 0.174127±0.002307 |
| MOIRAI | 0.018385±0.007579 | 0.104388±0.024288 | 0.019425±0.008114 | 0.107130±0.025211 | 0.027480±0.012148 | 0.125046±0.031647 | 0.020042±0.008491 | 0.108539±0.025857 | 0.017568±0.007127 | 0.102497±0.023308 | 0.017575±0.007114 | 0.100778±0.023248 |
| GRU | 0.000945±0.000806 | 0.021779±0.009647 | 0.000863±0.000766 | 0.020886±0.009594 | 0.000876±0.000766 | 0.021175±0.009509 | 0.000819±0.000725 | 0.020499±0.009231 | 0.000917±0.000764 | 0.021453±0.009309 | 0.001105±0.000409 | 0.021045±0.008990 |
| LSTM | 0.001212±0.000422 | 0.026180±0.004494 | 0.001115±0.000403 | 0.026258±0.004480 | 0.001136±0.000397 | 0.025454±0.004527 | 0.001046±0.000459 | 0.024325±0.004462 | 0.001159±0.000409 | 0.025368±0.004549 | 0.001105±0.000412 | 0.024799±0.004509 |
| DLinear | **0.000159±0.000001** | **0.008162±0.000044** | **0.000142±0.000001** | **0.007786±0.000054** | **0.000126±0.000001** | **0.007067±0.000061** | **0.000135±0.000001** | **0.007438±0.000039** | **0.000166±0.000001** | **0.008309±0.000050** | **0.000145±0.000001** | **0.007627±0.000045** |
| SparseTSF | 0.000561±0.000003 | 0.012513±0.000037 | 0.000371±0.000002 | 0.010296±0.000089 | 0.000349±0.000001 | 0.009804±0.000105 | 0.000562±0.000002 | 0.012916±0.000088 | 0.000397±0.000004 | 0.010577±0.000141 | 0.000352±0.000004 | 0.010383±0.000186 |
| FedFormer | 0.035734±0.002106 | 0.168029±0.004846 | 0.035710±0.002136 | 0.168057±0.005098 | 0.035710±0.002210 | 0.168057±0.005998 | 0.035639±0.002160 | 0.167863±0.004981 | 0.035503±0.002117 | 0.167743±0.004890 | 0.035537±0.002180 | 0.167569±0.005040 |
| DAF (Ours) | 0.000229±0.000006 | 0.011326±0.000042 | 0.000209±0.000008 | 0.010972±0.000111 | 0.000210±0.000001 | 0.010839±0.000138 | 0.000190±0.000013 | 0.010458±0.000240 | 0.000220±0.000009 | 0.011175±0.000132 | 0.000194±0.000008 | 0.010418±0.000128 |
| Improvements | -44.03±3.88 | -38.77±0.91 | -47.18±5.73 | - | -66.67±5.71 | - | -40.74±9.69 | -40.60±3.31 | -32.53±5.48 | - | -33.79±5.59 | - |

| Dataset | KUIPER-45 | | KUIPER-46 | | KUIPER-47 | | KUIPER-48 | | KUIPER-49 | | KUIPER-50 | |
| Metric | MSE | MAE | MSE | MAE | MSE | MAE | MSE | MAE | MSE | MAE | MSE | MAE |
|---|---|---|---|---|---|---|---|---|---|---|---|---|
| Time-MoE | 2.206898±0.013161 | 1.244679±0.003161 | 2.209318±0.014637 | 1.246078±0.003596 | 2.206366±0.013321 | 1.244876±0.002756 | 2.198285±0.008841 | 1.241777±0.001319 | 2.207904±0.015643 | 1.245192±0.003516 | 2.201731±0.012435 | 1.243767±0.002417 |
| Timer | 1.861819±0.001603 | 1.208409±0.000696 | 1.852729±0.002040 | 1.205204±0.000859 | 1.842509±0.002412 | 1.201817±0.001146 | 1.907691±0.002429 | 1.223502±0.000844 | 1.874222±0.001915 | 1.212815±0.000949 | 1.895577±0.000949 | 1.219732±0.000812 |
| Timer-XL | 2.001750±0.101694 | 1.256518±0.033918 | 1.994316±0.095629 | 1.254105±0.032450 | 1.991412±0.100731 | 1.253429±0.033747 | 2.020642±0.098785 | 1.262496±0.033747 | 2.008239±0.102437 | 1.258420±0.034219 | 2.017286±0.101257 | 1.261703±0.034007 |
| Time-LLM | 1.001494±0.109394 | 0.863138±0.065023 | 1.001337±0.108031 | 0.863296±0.064443 | 1.003338±0.106084 | 0.864225±0.063347 | 0.997125±0.114040 | 0.861177±0.067101 | 1.002028±0.110813 | 0.863277±0.065065 | 0.996941±0.111745 | 0.861245±0.066474 |
| TTM | 0.256064±0.151578 | 0.406151±0.167577 | 0.264644±0.169041 | 0.408684±0.169041 | 0.001459±0.000231 | 0.029109±0.001703 | 0.238366±0.140507 | 0.391605±0.160599 | 0.250782±0.148132 | 0.401709±0.165322 | 0.242744±0.143402 | 0.242744±0.143402 |
| TimesFM | 0.001453±0.000230 | 0.028949±0.001549 | 0.001472±0.000237 | 0.029011±0.001805 | 0.113542±0.001503 | 0.175466±0.002275 | 0.001423±0.000199 | 0.028622±0.001689 | 0.001523±0.000233 | 0.029316±0.001754 | 0.001399±0.000200 | 0.028507±0.001489 |
| MOMENT | 0.113189±0.001335 | 0.174899±0.002453 | 0.112274±0.001406 | 0.174718±0.001406 | 0.018124±0.007420 | 0.107322±0.023837 | 0.112386±0.001035 | 0.173843±0.002029 | 0.114558±0.001303 | 0.175501±0.002453 | 0.112692±0.000863 | 0.173929±0.002266 |
| MOIRAI | 0.017758±0.007231 | 0.102595±0.023511 | 0.018053±0.007444 | 0.103440±0.024009 | 0.001039±0.000795 | 0.022001±0.009499 | 0.017119±0.006881 | 0.100947±0.022670 | 0.017716±0.007139 | 0.102497±0.023246 | 0.017064±0.006897 | 0.100778±0.022729 |
| GRU | 0.000849±0.000751 | 0.020866±0.009384 | 0.000945±0.000794 | 0.021718±0.009547 | 0.001307±0.000406 | 0.026301±0.004417 | 0.000879±0.000731 | 0.021033±0.009111 | 0.000817±0.000722 | 0.024487±0.009206 | 0.000894±0.000774 | 0.021220±0.009511 |
| LSTM | 0.001104±0.000398 | 0.025112±0.004435 | 0.001226±0.000417 | 0.026258±0.004480 | 0.001307±0.000406 | 0.026301±0.004417 | 0.001120±0.000406 | 0.024948±0.004462 | 0.001040±0.000395 | 0.024260±0.004534 | 0.001105±0.000412 | 0.025378±0.004603 |
| DLinear | **0.000113±0.000001** | **0.006949±0.000048** | **0.000161±0.000001** | **0.007937±0.000059** | **0.000242±0.000001** | **0.008550±0.000054** | **0.000144±0.000001** | **0.007419±0.000053** | **0.000132±0.000001** | **0.007328±0.000052** | **0.000162±0.000001** | **0.008052±0.000052** |
| SparseTSF | 0.000390±0.000002 | 0.010575±0.000119 | 0.000550±0.000001 | 0.011882±0.000116 | 0.000550±0.000002 | 0.009972±0.000115 | 0.000349±0.000001 | 0.009625±0.000156 | 0.000441±0.000004 | 0.011251±0.000151 | 0.000510±0.000002 | 0.011645±0.000107 |
| FedFormer | 0.035709±0.002232 | 0.168027±0.005167 | 0.035729±0.002084 | 0.168032±0.005190 | 0.035720±0.002084 | 0.168082±0.005190 | 0.035547±0.002165 | 0.167652±0.004998 | 0.035586±0.002117 | 0.167698±0.004869 | 0.035701±0.002181 | 0.167996±0.005017 |
| DAF (Ours) | 0.000201±0.000012 | 0.010781±0.000222 | 0.000242±0.000007 | 0.011886±0.000159 | 0.000312±0.000007 | 0.011886±0.000064 | 0.000186±0.000009 | 0.010654±0.000156 | 0.000186±0.000012 | 0.011175±0.000162 | 0.000221±0.000102 | 0.011109±0.000112 |
| Improvements | -77.88±6.00 | - | -50.00±2.98 | -44.36±1.59 | -22.44±0.65 | - | -43.06±4.27 | -40.60±3.31 | -32.53±5.48 | -40.81±1.63 | -36.11±2.82 | -38.00±1.33 |

| Dataset | KUIPER-51 | | KUIPER-52 | | KUIPER-53 | | KUIPER-54 | | KUIPER-55 | | LEMUR-1 | |
| Metric | MSE | MAE | MSE | MAE | MSE | MAE | MSE | MAE | MSE | MAE | MSE | MAE |
|---|---|---|---|---|---|---|---|---|---|---|---|---|
| Time-MoE | 2.203300±0.017864 | 1.243840±0.004271 | 2.206026±0.014219 | 1.244998±0.002693 | 2.213601±0.013999 | 1.247784±0.002490 | 2.197686±0.011044 | 1.240770±0.002270 | 2.203901±0.012016 | 1.244178±0.002169 | 2.262186±0.021645 | 1.177649±0.007740 |
| Timer | 1.898146±0.001924 | 1.220278±0.000843 | 1.877975±0.002127 | 1.214216±0.001079 | 1.799109±0.013992 | 1.187042±0.001472 | 1.895177±0.001442 | 1.219431±0.000662 | 1.884105±0.002008 | 1.215763±0.000981 | 1.456705±0.005265 | 0.993270±0.001740 |
| Timer-XL | 2.017665±0.096528 | 1.261840±0.032173 | 2.003494±0.094956 | 1.257118±0.032001 | 1.977905±0.113488 | 1.248772±0.037813 | 2.011879±0.094494 | 1.260123±0.031677 | 2.011226±0.098757 | 1.259357±0.033180 | 1.874711±0.103555 | 1.131665±0.032705 |
| Time-LLM | 0.966900±0.113090 | 0.861163±0.067033 | 0.998622±0.110699 | 0.861608±0.065730 | 1.005698±0.099780 | 0.866097±0.060173 | 1.003090±0.112717 | 0.863078±0.066128 | 1.003865±0.113232 | 0.863631±0.066123 | 1.066624±0.051435 | 0.832592±0.033500 |
| TTM | 0.241520±0.142497 | 0.394060±0.161765 | 0.249601±0.147543 | 0.400871±0.165070 | 0.284042±0.169101 | 0.400871±0.165070 | 0.242936±0.145476 | 0.395265±0.162505 | 0.246074±0.145262 | 0.397987±0.163509 | 0.484452±0.291521 | 0.524423±0.222016 |
| TimesFM | 0.001428±0.000245 | 0.028720±0.001773 | 0.001549±0.000254 | 0.029667±0.001935 | 0.001458±0.000254 | 0.028893±0.001803 | 0.001427±0.000254 | 0.028673±0.001786 | 0.001430±0.000235 | 0.028518±0.001786 | 0.001292±0.000139 | 0.026020±0.001180 |
| MOMENT | 0.112970±0.000721 | 0.174665±0.001900 | 0.112752±0.001422 | 0.174602±0.002620 | 0.113563±0.001567 | 0.176364±0.003071 | 0.113042±0.000297 | 0.173887±0.001484 | 0.112512±0.001960 | 0.174069±0.002761 | 0.118409±0.000509 | 0.179251±0.001420 |
| MOIRAI | 0.017295±0.006931 | 0.101355±0.022804 | 0.017522±0.007079 | 0.101973±0.023158 | 0.019224±0.008060 | 0.106447±0.025184 | 0.017290±0.006894 | 0.100890±0.022877 | 0.017296±0.007001 | 0.101402±0.023004 | 0.030350±0.013680 | 0.124132±0.031924 |
| GRU | 0.000936±0.000786 | 0.021441±0.009593 | 0.000909±0.000760 | 0.021054±0.009330 | 0.001039±0.000795 | 0.021140±0.009109 | 0.001079±0.000735 | 0.022564±0.009692 | 0.000876±0.000729 | 0.020799±0.009131 | 0.001449±0.000583 | 0.027845±0.006515 |
| LSTM | 0.001168±0.000398 | 0.025248±0.004478 | 0.001151±0.000402 | 0.024929±0.004547 | 0.001132±0.000403 | 0.026564±0.004407 | 0.001332±0.000423 | 0.024568±0.004407 | 0.001104±0.000729 | 0.024568±0.004407 | 0.003255±0.000728 | 0.043577±0.000087 |
| DLinear | **0.000177±0.000001** | **0.008058±0.000046** | **0.000198±0.000001** | **0.008145±0.000051** | **0.000164±0.000001** | **0.007888±0.000046** | **0.000236±0.000001** | **0.008784±0.000046** | **0.000155±0.000001** | **0.007399±0.000050** | 0.001199±0.000018 | 0.024256±0.000202 |
| SparseTSF | 0.000369±0.000002 | 0.010612±0.000107 | 0.000583±0.000001 | 0.013552±0.000119 | 0.000430±0.000003 | 0.009926±0.000148 | 0.000430±0.000003 | 0.011039±0.000151 | 0.000236±0.000001 | 0.011896±0.000145 | 0.001548±0.000010 | 0.031174±0.000087 |
| FedFormer | 0.035665±0.002174 | 0.167927±0.005006 | 0.035660±0.002153 | 0.167856±0.004965 | 0.035539±0.002139 | 0.167631±0.004965 | 0.035547±0.002165 | 0.167886±0.005150 | 0.035607±0.002197 | 0.167786±0.005060 | 0.036429±0.002872 | 0.158102±0.005633 |
| DAF (Ours) | 0.000246±0.000000 | 0.011289±0.000172 | 0.000253±0.000011 | 0.011343±0.000265 | 0.000301±0.000009 | 0.010678±0.000159 | 0.000301±0.000000 | 0.012045±0.000019 | 0.000226±0.000001 | 0.010792±0.000213 | **0.000635±0.000044** | **0.018427±0.000338** |
| Improvements | -39.04±2.98 | - | -21.89±4.57 | -39.31±3.37 | -27.46±3.03 | - | -22.44±0.65 | - | -27.54±1.04 | -45.93±2.86 | -45.81±5.05 | 23.99±1.48 |

Table 18: Zero-shot evaluation on individual satellites of LEMUR dataset with mean and standard deviation in MSE and MAE. "Improvements" presents the relative performance of our model compared to the best baseline for each dataset and metric. A dash ('-') indicates that our model fails to achieve either the best or second-best performance. The best result is highlighted in **bold**, and the second-best is highlighted with underline.

### Block 1 (LEMUR-1 … LEMUR-7)

| Dataset / Metric | LEMUR-1 MSE | LEMUR-1 MAE | LEMUR-2 MSE | LEMUR-2 MAE | LEMUR-3 MSE | LEMUR-3 MAE | LEMUR-4 MSE | LEMUR-4 MAE | LEMUR-5 MSE | LEMUR-5 MAE | LEMUR-6 MSE | LEMUR-6 MAE | LEMUR-7 MSE | LEMUR-7 MAE |
|---|---|---|---|---|---|---|---|---|---|---|---|---|---|---|
| Time-MoE | 2.270557±0.021837 | 1.181452±0.008054 | 2.270776±0.024840 | 1.181442±0.008499 | 2.263674±0.026764 | 1.177746±0.008657 | 2.269115±0.028305 | 1.180724±0.010799 | 2.245731±0.024363 | 1.170082±0.008491 | 2.226021±0.023306 | 1.167437±0.007895 | | |
| Timer | 1.390327±0.004634 | 0.969750±0.001738 | 1.409772±0.004637 | 0.976550±0.001670 | 1.449399±0.003715 | 0.988995±0.001483 | 1.418423±0.006092 | 0.979079±0.002248 | 1.596563±0.003556 | 1.039972±0.001411 | 1.823798±0.002460 | 1.116722±0.000926 | | |
| Timer-XL | 1.852936±0.115626 | 1.124566±0.034176 | 1.858761±0.108848 | 1.126664±0.034476 | 1.872772±0.109992 | 1.130132±0.034541 | 1.864888±0.112155 | 1.127624±0.035398 | 1.932366±0.090335 | 1.148481±0.027956 | 1.987119±0.104828 | 1.167001±0.031754 | | |
| Time-LLM | 1.072625±0.043632 | 0.835038±0.030418 | 1.073021±0.044481 | 0.835159±0.030646 | 1.067641±0.049129 | 0.832380±0.032874 | 1.068855±0.043839 | 0.833304±0.030766 | 1.050314±0.069734 | 0.824684±0.041809 | 1.020443±0.100739 | 0.814670±0.056371 | | |
| TTM | 0.517698±0.312095 | 0.542187±0.230070 | 0.509558±0.307310 | 0.537896±0.228308 | 0.488668±0.294042 | 0.526401±0.222597 | 0.503989±0.303854 | 0.534558±0.226687 | 0.416500±0.249964 | 0.485291±0.204076 | 0.281597±0.168388 | 0.399196±0.166686 | | |
| TimesFM | 0.001289±0.000143 | 0.025929±0.001290 | 0.001280±0.000124 | 0.026023±0.001182 | 0.001296±0.000116 | 0.026019±0.001237 | 0.001265±0.000116 | 0.025804±0.001083 | 0.001298±0.000127 | 0.026142±0.001140 | 0.001343±0.000135 | 0.026627±0.001322 | | |
| MOMENT | 0.118135±0.000867 | 0.173576±0.000744 | 0.118875±0.001909 | 0.173934±0.002824 | 0.119036±0.001382 | 0.173310±0.002531 | 0.118428±0.000849 | 0.173338±0.002284 | 0.116032±0.001845 | 0.168794±0.002920 | 0.114297±0.000434 | 0.169470±0.001543 | | |
| MOIRAI | 0.032605±0.014574 | 0.128383±0.032879 | 0.031861±0.014306 | 0.127006±0.032615 | 0.030633±0.013839 | 0.124429±0.032192 | 0.031675±0.014315 | 0.126554±0.032681 | 0.025866±0.011580 | 0.114681±0.029235 | 0.019855±0.008316 | 0.101452±0.023908 | | |
| GRU | 0.001669±0.000605 | 0.029787±0.006399 | 0.001591±0.000647 | 0.029289±0.006755 | 0.001511±0.000548 | 0.028386±0.006074 | 0.001576±0.000569 | 0.028905±0.006053 | 0.001163±0.000446 | 0.025332±0.005565 | 0.001530±0.000337 | 0.029370±0.003758 | | |
| LSTM | 0.003557±0.000772 | 0.045564±0.004711 | 0.003466±0.000669 | 0.029289±0.006755 | 0.003212±0.000669 | 0.043451±0.004353 | 0.003387±0.000741 | 0.044451±0.004461 | 0.002562±0.000537 | 0.038899±0.003898 | 0.001764±0.000592 | 0.032056±0.006550 | | |
| DLinear | 0.001531±0.000025 | 0.027507±0.000239 | 0.001410±0.000023 | 0.026495±0.000240 | 0.001222±0.000018 | 0.024442±0.000206 | 0.001363±0.000022 | 0.025960±0.000230 | 0.000674±0.000009 | 0.017467±0.000142 | **0.000267±0.000002** | **0.009436±0.000065** | | |
| SparseTSF | 0.002169±0.000012 | 0.037118±0.000088 | 0.001906±0.000012 | 0.034966±0.000088 | 0.001675±0.000010 | 0.032342±0.000084 | 0.001864±0.000011 | 0.034458±0.000084 | 0.000763±0.000006 | 0.020900±0.000076 | 0.000275±0.000001 | 0.009652±0.005000 | | |
| FedFormer | 0.036228±0.002888 | 0.157705±0.005664 | 0.036284±0.002861 | 0.157897±0.005637 | 0.036450±0.002804 | 0.158004±0.005500 | 0.036321±0.002867 | 0.157791±0.005605 | 0.036786±0.002654 | 0.158421±0.005090 | 0.035626±0.002300 | 0.158052±0.004272 | | |
| DAF (Ours) | **0.000765±0.000051** | **0.020009±0.000284** | **0.000697±0.000049** | **0.019388±0.000322** | **0.000210±0.000009** | **0.008509±0.000293** | **0.000682±0.000046** | **0.018509±0.000297** | **0.000497±0.000026** | **0.015317±0.000314** | 0.000304±0.000005 | 0.012024±0.000058 | | |
| Improvements | 40.65±6.21 | 22.84±3.22 | 45.55±5.59 | 25.50±2.99 | 47.12±2.17 | 24.29±1.42 | 23.91±2.58 | 6.60±5.52 | 33.67±3.90 | 12.30±1.80 | 46.09±5.21 | 25.89±2.86 | 44.58±2.00 | 23.14±3.52 |

### Block 2 (LEMUR-8 … LEMUR-13)

| Dataset / Metric | LEMUR-8 MSE | LEMUR-8 MAE | LEMUR-9 MSE | LEMUR-9 MAE | LEMUR-10 MSE | LEMUR-10 MAE | LEMUR-11 MSE | LEMUR-11 MAE | LEMUR-12 MSE | LEMUR-12 MAE | LEMUR-13 MSE | LEMUR-13 MAE |
|---|---|---|---|---|---|---|---|---|---|---|---|---|
| Time-MoE | 2.133689±0.017159 | 1.134209±0.005950 | 2.226061±0.016911 | 1.167599±0.006040 | 2.185401±0.020653 | 1.152031±0.007701 | 2.120949±0.015184 | 1.107181±0.004279 | 2.124512±0.022187 | 1.123994±0.007318 | 2.136297±0.015111 | 1.142362±0.006551 |
| Timer | 2.602057±0.005035 | 1.337182±0.000739 | 1.806608±0.001634 | 1.111795±0.000793 | 2.170905±0.000705 | 1.219771±0.000162 | 2.628155±0.006602 | 1.316072±0.001955 | 2.608792±0.005341 | 1.330441±0.001177 | 2.600266±0.004903 | 1.348083±0.000715 |
| Timer-XL | 2.355040±0.003900 | 1.275336±0.003900 | 1.975695±0.108719 | 1.165562±0.032977 | 2.133608±0.064191 | 1.211971±0.019221 | 2.394136±0.014862 | 1.259023±0.000931 | 2.368578±0.009649 | 1.270157±0.001812 | 2.363068±0.007367 | 1.287237±0.002673 |
| Time-LLM | 0.967904±0.166656 | 0.789511±0.088232 | 1.022458±0.098862 | 0.816201±0.055487 | 0.991568±0.141623 | 0.800104±0.075521 | 0.968342±0.168026 | 0.773080±0.085769 | 0.966455±0.169121 | 0.782371±0.089321 | 0.966254±0.167289 | 0.793769±0.089284 |
| TTM | 0.075748±0.042529 | 0.189550±0.069417 | 0.289132±0.172934 | 0.404910±0.169286 | 0.155156±0.088904 | 0.288336±0.113416 | 0.070941±0.039399 | 0.182674±0.065924 | 0.068029±0.037627 | 0.181330±0.065030 | 0.071625±0.040015 | 0.187107±0.067934 |
| TimesFM | 0.001936±0.000135 | 0.030595±0.000625 | 0.001355±0.000133 | 0.026795±0.001301 | 0.001485±0.000111 | 0.027807±0.000720 | 0.001835±0.000198 | 0.030007±0.000762 | 0.001797±0.000141 | 0.030174±0.000535 | 0.001920±0.000194 | 0.030985±0.000941 |
| MOMENT | 0.111667±0.000682 | 0.156092±0.000221 | 0.115577±0.000465 | 0.169987±0.002465 | 0.112348±0.001316 | 0.158975±0.002066 | 0.112153±0.000262 | 0.154634±0.000113 | 0.113387±0.001505 | 0.156844±0.000776 | 0.111939±0.000867 | 0.158019±0.000471 |
| MOIRAI | 0.010085±0.002856 | 0.072949±0.011058 | 0.020182±0.008391 | 0.102338±0.023988 | 0.013798±0.004937 | 0.084949±0.016666 | 0.009600±0.002470 | 0.070019±0.002659 | 0.009676±0.002957 | 0.071177±0.010537 | 0.009893±0.002762 | 0.110679±0.010937 |
| GRU | 0.005916±0.002771 | 0.052158±0.015740 | 0.001495±0.000316 | 0.029083±0.003677 | 0.002766±0.001029 | 0.037685±0.008130 | 0.003768±0.003103 | 0.045101±0.019312 | 0.004014±0.003111 | 0.045777±0.019545 | 0.003886±0.002858 | 0.044615±0.019092 |
| LSTM | 0.004992±0.002192 | 0.050026±0.010222 | 0.001783±0.000584 | 0.032166±0.006574 | 0.002010±0.001120 | 0.034288±0.009047 | 0.003662±0.000952 | 0.046426±0.007330 | 0.003909±0.000981 | 0.047019±0.007057 | 0.004000±0.001292 | 0.046977±0.018177 |
| DLinear | 0.000259±0.000002 | 0.010525±0.000050 | **0.000297±0.000003** | **0.010140±0.000069** | 0.000276±0.000001 | 0.010114±0.000031 | 0.000268±0.000000 | 0.010779±0.000053 | 0.000359±0.000002 | 0.012135±0.000052 | 0.000227±0.000001 | 0.010457±0.000041 |
| SparseTSF | 0.000584±0.000000 | 0.017272±0.000086 | 0.000331±0.000000 | 0.010316±0.000028 | 0.000420±0.000007 | 0.010710±0.000090 | 0.000611±0.000005 | 0.017037±0.000009 | 0.000708±0.000008 | 0.013342±0.000052 | 0.000481±0.000003 | 0.015915±0.000067 |
| FedFormer | 0.031324±0.001445 | 0.146597±0.002848 | 0.036581±0.002325 | 0.158258±0.004349 | 0.034884±0.001804 | 0.153849±0.003271 | 0.031289±0.001372 | 0.141420±0.002333 | 0.031132±0.001398 | 0.143748±0.002542 | 0.031471±0.001382 | 0.146433±0.002569 |
| DAF (Ours) | **0.000173±0.000006** | **0.008695±0.000311** | 0.000315±0.000007 | 0.012332±0.000101 | **0.000210±0.000006** | **0.009446±0.000522** | **0.000151±0.000009** | **0.009151±0.000009** | **0.000334±0.000008** | **0.009083±0.000318** | **0.000126±0.000008** | **0.008040±0.000352** |
| Improvements | 33.20±2.20 | 17.37±3.60 | -6.06±0.73 | - | 23.91±2.58 | 6.60±5.52 | 43.46±3.40 | 23.31±5.57 | 43.46±1.41 | 25.19±2.72 | 33.67±3.90 | 12.30±1.80 |

### Block 3 (LEMUR-14 … LEMUR-19)

| Dataset / Metric | LEMUR-14 MSE | LEMUR-14 MAE | LEMUR-15 MSE | LEMUR-15 MAE | LEMUR-16 MSE | LEMUR-16 MAE | LEMUR-17 MSE | LEMUR-17 MAE | LEMUR-18 MSE | LEMUR-18 MAE | LEMUR-19 MSE | LEMUR-19 MAE |
|---|---|---|---|---|---|---|---|---|---|---|---|---|
| Time-MoE | 2.111134±0.019578 | 1.086436±0.006624 | 2.080526±0.012200 | 1.058353±0.004776 | 2.274876±0.026810 | 1.205931±0.009422 | 2.261404±0.025480 | 1.198569±0.008118 | 2.244807±0.018854 | 1.191126±0.006491 | 2.245052±0.018549 | 1.190074±0.006559 |
| Timer | 2.676572±0.006479 | 1.309844±0.001399 | 2.836756±0.006953 | 1.330818±0.001415 | 1.362993±0.004773 | 0.977287±0.001923 | 1.489952±0.003728 | 1.022097±0.001359 | 1.714284±0.004241 | 1.098521±0.001309 | 1.675560±0.003524 | 1.083690±0.001558 |
| Timer-XL | 2.435532±0.035400 | 1.251503±0.007971 | 2.548262±0.072837 | 1.257523±0.017217 | 1.360308±0.157926 | 1.127594±0.051692 | 1.848669±0.141633 | 1.142696±0.045826 | 1.953789±0.146346 | 1.174925±0.032763 | 1.931913±0.118189 | 1.167154±0.036919 |
| Time-LLM | 0.961370±0.169408 | 0.758855±0.086889 | 0.960604±0.171720 | 0.744140±0.085583 | 1.076744±0.083483 | 0.852126±0.028843 | 1.050271±0.054553 | 0.844310±0.036219 | 1.034038±0.089045 | 0.832682±0.051460 | 1.042343±0.081873 | 0.834081±0.048179 |
| TTM | 0.065451±0.036144 | 0.172857±0.061592 | 0.062000±0.034083 | 0.162619±0.056587 | 0.500855±0.302321 | 0.543279±0.231766 | 0.439351±0.263780 | 0.508052±0.215003 | 0.355037±0.211292 | 0.456215±0.189389 | 0.344382±0.206961 | 0.447947±0.188896 |
| TimesFM | 0.001991±0.000232 | 0.030413±0.000959 | 0.002332±0.000512 | 0.031460±0.002392 | 0.001229±0.000097 | 0.026050±0.001114 | 0.001331±0.000116 | 0.027008±0.000144 | 0.001354±0.000144 | 0.027126±0.001302 | 0.001305±0.000136 | 0.026641±0.001258 |
| MOMENT | 0.111822±0.002247 | 0.152506±0.001683 | 0.112604±0.003988 | 0.151079±0.002476 | 0.117557±0.000329 | 0.177427±0.001592 | 0.117849±0.000728 | 0.175170±0.002237 | 0.115537±0.001290 | 0.169477±0.002359 | 0.114891±0.001303 | 0.169470±0.002532 |
| MOIRAI | 0.009282±0.002417 | 0.067767±0.009505 | 0.009220±0.002232 | 0.056258±0.008451 | 0.033558±0.015131 | 0.132521±0.034433 | 0.029265±0.013147 | 0.123895±0.031883 | 0.022706±0.000957 | 0.109588±0.027280 | 0.023176±0.010066 | 0.023176±0.027251 |
| GRU | 0.003553±0.002804 | 0.044641±0.017367 | 0.005773±0.004131 | 0.055007±0.019265 | 0.001585±0.001288 | 0.029277±0.011823 | 0.001333±0.000925 | 0.027225±0.009528 | 0.001078±0.000536 | 0.025490±0.006315 | 0.001738±0.000370 | 0.031734±0.003812 |
| LSTM | 0.003279±0.001145 | 0.044785±0.008413 | 0.004974±0.001614 | 0.055167±0.008877 | 0.004805±0.000921 | 0.052205±0.006345 | 0.004026±0.000952 | 0.047577±0.006255 | 0.002444±0.000654 | 0.037305±0.006255 | 0.002019±0.000646 | 0.033829±0.007025 |
| DLinear | 0.000337±0.000003 | 0.011672±0.000061 | 0.000184±0.000000 | **0.008536±0.000013** | 0.001675±0.000027 | 0.029744±0.000266 | 0.001097±0.000016 | 0.023190±0.000197 | 0.000380±0.000004 | **0.012707±0.000102** | 0.000431±0.000006 | 0.013849±0.000116 |
| SparseTSF | 0.000500±0.000006 | 0.015707±0.000097 | 0.000305±0.000004 | 0.010037±0.000110 | 0.001675±0.000027 | 0.040721±0.000091 | 0.001468±0.000008 | 0.030152±0.000082 | 0.000431±0.000000 | 0.013996±0.000057 | 0.000552±0.000003 | 0.016385±0.000058 |
| FedFormer | 0.031641±0.001236 | 0.139324±0.001990 | 0.029395±0.000936 | 0.133281±0.001689 | 0.035930±0.002747 | 0.160125±0.005644 | 0.036491±0.002744 | 0.161121±0.005606 | 0.036802±0.002356 | 0.161187±0.004662 | 0.036741±0.002473 | 0.160816±0.004901 |
| DAF (Ours) | **0.000201±0.000006** | **0.008867±0.000267** | **0.000172±0.000010** | 0.008717±0.000027 | **0.000799±0.000007** | **0.021138±0.000354** | **0.000651±0.000040** | **0.018516±0.000350** | **0.000334±0.000006** | 0.013279±0.000213 | **0.000339±0.000008** | **0.013611±0.000199** |
| Improvements | 40.34±1.87 | 24.04±3.04 | 6.52±0.59 | -2.12±1.08 | 34.96±6.92 | 18.84±4.44 | 40.68±3.70 | 20.13±3.14 | 12.12±1.79 | 21.32±1.87 | 21.32±1.87 | 1.72±1.69 |

### Block 4 (LEMUR-20 … LEMUR-25)

| Dataset / Metric | LEMUR-20 MSE | LEMUR-20 MAE | LEMUR-21 MSE | LEMUR-21 MAE | LEMUR-22 MSE | LEMUR-22 MAE | LEMUR-23 MSE | LEMUR-23 MAE | LEMUR-24 MSE | LEMUR-24 MAE | LEMUR-25 MSE | LEMUR-25 MAE |
|---|---|---|---|---|---|---|---|---|---|---|---|---|
| Time-MoE | 2.259580±0.019395 | 1.194716±0.007511 | 2.234611±0.021311 | 1.180151±0.006324 | 2.132125±0.012861 | 1.145349±0.004654 | 2.130991±0.015850 | 1.145063±0.005237 | 2.133341±0.019063 | 1.149455±0.006154 | 2.103846±0.012847 | 1.114647±0.003777 |
| Timer | 1.492539±0.002652 | 1.020331±0.001849 | 1.737170±0.002103 | 1.098364±0.000768 | 2.566492±0.001572 | 1.342195±0.000013 | 2.574307±0.000703 | 1.344317±0.000703 | 2.566029±0.003989 | 1.346359±0.001102 | 2.785524±0.008022 | 1.371023±0.001868 |
| Timer-XL | 1.860303±0.142282 | 1.143329±0.044844 | 1.947461±0.118771 | 1.165727±0.037414 | 2.331246±0.024693 | 1.282782±0.007673 | 2.334111±0.025592 | 1.283502±0.007414 | 2.330428±0.028308 | 1.286506±0.008666 | 2.479396±0.050141 | 1.297072±0.011209 |
| Time-LLM | 1.057141±0.059496 | 0.841003±0.037907 | 1.034240±0.089325 | 0.826910±0.051102 | 0.971972±0.165683 | 0.798693±0.088464 | 0.968492±0.165421 | 0.798101±0.088707 | 0.968778±0.166799 | 0.800360±0.089641 | 0.955286±0.176233 | 0.777648±0.091835 |
| TTM | 0.424488±0.259331 | 0.500179±0.213881 | 0.329847±0.195590 | 0.436801±0.180898 | 0.197843±0.069678 | 0.197843±0.069678 | 0.199432±0.070010 | 0.199432±0.070010 | 0.072225±0.039258 | 0.194185±0.068670 | 0.046676±0.024475 | 0.046153±0.046936 |
| TimesFM | 0.001330±0.000138 | 0.026849±0.001258 | 0.001310±0.000126 | 0.026657±0.001219 | 0.001895±0.000171 | 0.030865±0.000652 | 0.001902±0.000166 | 0.030955±0.000565 | 0.001740±0.000123 | 0.030194±0.000520 | 0.001944±0.000317 | 0.030671±0.001558 |
| MOMENT | 0.117078±0.000514 | 0.173690±0.001354 | 0.115824±0.000485 | 0.168575±0.001924 | 0.111444±0.001218 | 0.158162±0.001075 | 0.113116±0.000490 | 0.160393±0.000230 | 0.113029±0.001270 | 0.159251±0.000995 | 0.111353±0.000763 | 0.155015±0.000649 |
| MOIRAI | 0.028479±0.012602 | 0.122244±0.030838 | 0.022249±0.009684 | 0.107915±0.026765 | 0.010278±0.002928 | 0.074538±0.011418 | 0.010793±0.002951 | 0.074934±0.011346 | 0.010178±0.002835 | 0.074452±0.011178 | 0.009117±0.002247 | 0.068851±0.009113 |
| GRU | 0.001826±0.000253 | 0.031595±0.002745 | 0.001304±0.000455 | 0.027988±0.005224 | 0.002287±0.000268 | 0.037988±0.007210 | 0.008347±0.002777 | 0.068639±0.010859 | 0.008476±0.002730 | 0.069429±0.010970 | 0.001812±0.000337 | 0.083173±0.010733 |
| LSTM | 0.002727±0.000939 | 0.038531±0.008568 | 0.002552±0.000821 | 0.037899±0.007210 | 0.002627±0.000916 | 0.037923±0.007671 | 0.002644±0.000901 | 0.040064±0.008190 | 0.003000±0.001163 | 0.040064±0.008190 | 0.002846±0.001401 | 0.041022±0.011192 |
| DLinear | 0.001404±0.000009 | 0.022588±0.000190 | 0.000337±0.000006 | 0.011649±0.000094 | 0.000269±0.000002 | 0.010624±0.000038 | 0.000375±0.000003 | 0.010785±0.000041 | 0.000275±0.000000 | 0.011920±0.000050 | 0.000216±0.000001 | 0.010187±0.000035 |
| SparseTSF | 0.000500±0.000006 | 0.029367±0.000088 | 0.000375±0.000002 | 0.012785±0.000088 | 0.000483±0.000004 | 0.016230±0.000077 | 0.000517±0.000000 | 0.016491±0.000002 | 0.000497±0.000003 | 0.016291±0.000062 | 0.000383±0.000004 | 0.011383±0.000054 |
| FedFormer | 0.036528±0.002701 | 0.160130±0.005453 | 0.036763±0.002336 | 0.160454±0.004590 | 0.032005±0.001407 | 0.148281±0.002803 | 0.032006±0.001407 | 0.148281±0.002687 | 0.031637±0.001461 | 0.148524±0.002905 | 0.030203±0.001133 | 0.145323±0.002331 |
| DAF (Ours) | **0.000586±0.000039** | **0.017838±0.000381** | **0.000327±0.000004** | **0.012939±0.000157** | **0.000167±0.000004** | **0.008387±0.000432** | **0.000172±0.000005** | **0.008436±0.000475** | **0.000169±0.000005** | **0.008523±0.000419** | **0.000136±0.000009** | **0.007859±0.000204** |
| Improvements | 43.42±3.65 | 21.08±2.21 | 2.91±0.43 | - | 37.99±1.45 | 21.09±4.99 | 37.45±1.53 | 21.76±5.21 | 38.56±1.77 | 21.96±4.65 | 37.04±1.59 | 22.80±2.01 |

Table 19: Zero-shot evaluation on individual satellites of LEMUR dataset with mean and standard deviation in MSE and MAE. "Improvements" presents the relative performance of our model compared to the best baseline for each dataset and metric. A dash ('-') indicates that our model fails to achieve either the best or second-best performance. The best result is highlighted in **bold**, and the second-best is highlighted with underline.

| Dataset | LEMUR-26 | | LEMUR-27 | | LEMUR-28 | | LEMUR-29 | | LEMUR-30 | | LEMUR-31 | |
|---|---|---|---|---|---|---|---|---|---|---|---|---|
| Metric | MSE | MAE | MSE | MAE | MSE | MAE | MSE | MAE | MSE | MAE | MSE | MAE |
| Time-MoE | 2.121033±0.017968 | 1.129477±0.005782 | 2.224868±0.019465 | 1.740337±0.006740 | 2.224142±0.019690 | 1.172933±0.006528 | 2.014658±0.034246 | 1.159439±0.012353 | 2.241046±0.020837 | 1.176901±0.006419 | 2.243558±0.018825 | 1.178247±0.005945 |
| Timer | 2.619498±0.005346 | 1.339852±0.001596 | 1.825030±0.001263 | 1.123826±0.000438 | 1.831859±0.002476 | 1.125406±0.001104 | 3.114342±0.003238 | 1.551045±0.000234 | 1.655690±0.002582 | 1.065836±0.001012 | 1.658779±0.003791 | 1.067136±0.001212 |
| Timer-XL | 2.366910±0.016703 | 1.277683±0.005308 | 1.982044±0.103827 | 1.174113±0.032542 | 1.983460±0.103598 | 1.173575±0.032699 | 2.763300±0.119058 | 1.463361±0.031381 | 1.951797±0.089314 | 1.161127±0.027471 | 1.954101±0.086110 | 1.162239±0.026596 |
| Time-LLM | 0.960871±0.171054 | 0.786256±0.090578 | 1.022424±0.100768 | 0.819939±0.056622 | 1.019334±0.101412 | 0.818692±0.057094 | 0.982970±0.147675 | 0.844308±0.081139 | 1.042206±0.079573 | 0.825912±0.046681 | 1.043380±0.079094 | 0.826710±0.046277 |
| TTM | 0.065502±0.034731 | 0.182815±0.062249 | 0.292633±0.172481 | 0.410378±0.168131 | 0.290765±0.171489 | 0.408756±0.167709 | 0.146105±0.077146 | 0.300024±0.102151 | 0.382214±0.229443 | 0.467452±0.196643 | 0.379988±0.227994 | 0.466246±0.195983 |
| TimesFM | 0.001846±0.000196 | 0.030437±0.000772 | 0.001313±0.000123 | 0.026638±0.001235 | 0.001328±0.000135 | 0.026726±0.001337 | 0.001304±0.000119 | 0.026401±0.001094 | 0.001294±0.000119 | 0.026351±0.001123 | 0.001312±0.000125 | 0.026490±0.001228 |
| MOMENT | 0.111910±0.000826 | 0.157539±0.000675 | 0.115084±0.001541 | 0.166332±0.002664 | 0.114526±0.000972 | 0.165723±0.001837 | 0.148673±0.007597 | 0.243599±0.001409 | 0.117053±0.000841 | 0.169389±0.002084 | 0.116891±0.001432 | 0.169312±0.002453 |
| MOIRAI | 0.009790±0.002634 | 0.072211±0.010417 | 0.019965±0.008458 | 0.102211±0.024450 | 0.019984±0.008462 | 0.102198±0.024429 | 0.017855±0.004262 | 0.102993±0.013349 | 0.024079±0.010674 | 0.111389±0.027920 | 0.024090±0.010633 | 0.111465±0.027862 |
| GRU | 0.010050±0.003043 | 0.075065±0.010604 | 0.001425±0.000465 | 0.029381±0.004559 | 0.001456±0.000471 | 0.029674±0.004557 | 0.001165±0.000216 | 0.025562±0.003318 | 0.001188±0.000214 | 0.025671±0.003280 | 0.001147±0.000313 | 0.026221±0.004387 |
| LSTM | 0.002900±0.000954 | 0.039851±0.008355 | 0.002239±0.000776 | 0.035532±0.007215 | 0.002221±0.000785 | 0.035337±0.007230 | 0.002227±0.000785 | 0.035454±0.004303 | 0.002227±0.000540 | 0.036384±0.004229 | 0.001793±0.000436 | 0.032744±0.004415 |
| DLinear | 0.000236±0.000001 | 0.010340±0.000044 | 0.000235±0.000002 | 0.009208±0.000064 | 0.000243±0.000002 | 0.009155±0.000069 | 0.000477±0.000007 | 0.014579±0.000123 | 0.000479±0.000006 | 0.014581±0.000114 | 0.000248±0.000001 | 0.008802±0.000047 |
| SparseTSF | 0.000517±0.000000 | 0.016414±0.000000 | 0.000262±0.000000 | 0.009282±0.000032 | 0.000316±0.000032 | 0.009752±0.000048 | 0.000511±0.000000 | 0.016675±0.000007 | 0.000501±0.000004 | 0.016683±0.000067 | **0.000224±0.000000** | **0.008329±0.000087** |
| FedFormer | 0.031251±0.001431 | 0.145127±0.002824 | 0.036549±0.002313 | 0.159502±0.004477 | 0.036653±0.002267 | 0.159295±0.004367 | 0.036858±0.002545 | 0.159289±0.004874 | 0.036866±0.002516 | 0.159370±0.004824 | 0.036431±0.002319 | 0.158246±0.004312 |
| DAF (Ours) | **0.000143±0.000008** | **0.008048±0.000412** | 0.000276±0.000054 | 0.011828±0.000054 | 0.000282±0.000006 | 0.011860±0.000039 | **0.000372±0.000013** | **0.014037±0.000238** | **0.000370±0.000013** | **0.013973±0.000237** | 0.000285±0.000007 | 0.011579±0.000054 |
| Improvements | 39.41±5.66 | 22.14±4.07 | — | — | — | — | 22.00±2.77 | 3.72±1.90 | 22.76±2.78 | 4.17±1.91 | 11.97±1.69 | — |

| Dataset | LEMUR-32 | | LEMUR-33 | | LEMUR-34 | | LEMUR-35 | | LEMUR-36 | | LEMUR-37 | |
|---|---|---|---|---|---|---|---|---|---|---|---|---|
| Metric | MSE | MAE | MSE | MAE | MSE | MAE | MSE | MAE | MSE | MAE | MSE | MAE |
| Time-MoE | 2.218436±0.014753 | 1.169537±0.004889 | 2.239745±0.023043 | 1.176389±0.007597 | 2.222412±0.014158 | 1.185953±0.004854 | 2.225195±0.017630 | 1.186908±0.005083 | 2.237482±0.020686 | 1.182892±0.006670 | 2.302668±0.023940 | 1.208285±0.009118 |
| Timer | 1.857513±0.002087 | 1.132575±0.000961 | 1.671649±0.002777 | 1.071700±0.001104 | 1.877230±0.004142 | 1.153600±0.001420 | 1.892360±0.001726 | 1.158444±0.000696 | 1.644210±0.005275 | 1.067608±0.001903 | 1.078125±0.003992 | 0.858843±0.001748 |
| Timer-XL | 2.035758±0.070323 | 1.188725±0.021324 | 1.963695±0.088628 | 1.165078±0.027337 | 2.027605±0.084187 | 1.198765±0.026114 | 1.906315±0.126097 | 1.200962±0.025723 | 1.152474±0.040124 | 1.152474±0.040124 | 1.682789±0.179375 | 1.080026±0.059948 |
| Time-LLM | 1.023887±0.106145 | 0.818402±0.058590 | 1.042172±0.083318 | 0.825640±0.047915 | 1.019998±0.109234 | 0.827756±0.060458 | 1.016870±0.110905 | 0.826838±0.061843 | 1.040887±0.076696 | 0.830123±0.045695 | 1.11193±0.011303 | 0.860283±0.009437 |
| TTM | 0.292285±0.173643 | 0.408825±0.168600 | 0.373388±0.224336 | 0.462235±0.194687 | 0.284365±0.167285 | 0.404999±0.166409 | 0.278767±0.164041 | 0.404847±0.164811 | 0.371707±0.221836 | 0.463220±0.193762 | 0.651558±0.393751 | 0.615372±0.262986 |
| TimesFM | 0.001280±0.000125 | 0.026216±0.001181 | 0.001370±0.000153 | 0.027295±0.001131 | 0.001392±0.000094 | 0.027377±0.001450 | 0.001352±0.000094 | 0.027047±0.001005 | 0.001856±0.000142 | 0.031575±0.001317 | 0.001945±0.000150 | 0.032277±0.001371 |
| MOMENT | 0.114373±0.001052 | 0.165384±0.002320 | 0.115725±0.001728 | 0.168402±0.002836 | 0.114050±0.000946 | 0.166898±0.002291 | 0.114354±0.001224 | 0.166917±0.002169 | 0.117425±0.000844 | 0.171154±0.002411 | 0.122274±0.001218 | 0.185081±0.002386 |
| MOIRAI | 0.019122±0.008041 | 0.099890±0.023568 | 0.023513±0.019787 | 0.110107±0.027541 | 0.019000±0.007985 | 0.100873±0.023955 | 0.018764±0.007985 | 0.100213±0.023582 | 0.024794±0.010961 | 0.113434±0.028553 | 0.045710±0.020194 | 0.152782±0.039358 |
| GRU | 0.001156±0.000207 | 0.025441±0.003305 | 0.001296±0.000280 | 0.028425±0.003111 | 0.001324±0.000296 | 0.028735±0.003268 | 0.001214±0.000589 | 0.026730±0.006786 | 0.003177±0.002322 | 0.040958±0.015485 | 0.003277±0.002347 | 0.041503±0.015490 |
| LSTM | 0.002156±0.000517 | 0.035924±0.004256 | 0.001991±0.000666 | 0.033811±0.006966 | 0.001972±0.000701 | 0.033633±0.007259 | 0.003090±0.000701 | 0.041515±0.007146 | 0.007431±0.000884 | 0.064805±0.005504 | 0.007607±0.000896 | 0.065455±0.005503 |
| DLinear | 0.000431±0.000006 | 0.013737±0.000117 | 0.000223±0.000001 | 0.008598±0.000049 | **0.000224±0.000001** | **0.008439±0.000046** | 0.000553±0.000000 | 0.015543±0.000128 | 0.004242±0.000075 | 0.047880±0.000442 | 0.004358±0.000076 | 0.048492±0.000448 |
| SparseTSF | 0.000541±0.000000 | 0.016340±0.000000 | 0.000238±0.000000 | 0.008844±0.000098 | 0.000316±0.000000 | 0.009808±0.000121 | 0.018307±0.000073 | 0.018307±0.000073 | 0.007200±0.000025 | 0.069829±0.000084 | 0.007270±0.000027 | 0.070218±0.000090 |
| FedFormer | 0.036852±0.002511 | 0.159271±0.004807 | 0.036354±0.002224 | 0.160465±0.004394 | 0.036282±0.002203 | 0.160341±0.004349 | 0.036798±0.002487 | 0.160653±0.004930 | 0.034035±0.001984 | 0.154241±0.003897 | 0.034034±0.001966 | 0.154160±0.003795 |
| DAF (Ours) | **0.000351±0.000009** | **0.013573±0.000193** | 0.000256±0.000004 | 0.011316±0.000036 | 0.000266±0.000006 | 0.011285±0.000069 | **0.000431±0.000020** | **0.014753±0.000300** | **0.001337±0.000042** | **0.027983±0.000135** | **0.001389±0.000043** | **0.028418±0.000146** |
| Improvements | 18.53±2.10 | 1.20±1.52 | — | — | -16.10±2.59 | -33.64±1.07 | 22.00±2.77 | 3.72±1.90 | 22.76±2.78 | 5.08±2.11 | 28.58±2.23 | 11.97±1.69 |

| Dataset | LEMUR-38 | | LEMUR-39 | | LEMUR-40 | | LEMUR-41 | | LEMUR-42 | | LEMUR-43 | |
|---|---|---|---|---|---|---|---|---|---|---|---|---|
| Metric | MSE | MAE | MSE | MAE | MSE | MAE | MSE | MAE | MSE | MAE | MSE | MAE |
| Time-MoE | 2.308832±0.023692 | 1.208857±0.009290 | 2.305339±0.022858 | 1.208267±0.008948 | 2.304087±0.021026 | 1.208467±0.008752 | 2.305200±0.023197 | 1.209374±0.008882 | 2.305842±0.022885 | 1.209031±0.008951 | 2.317955±0.027546 | 1.214072±0.011152 |
| Timer | 1.682402±0.184814 | 1.079726±0.061458 | 1.695550±0.182665 | 1.083466±0.060714 | 1.672118±0.180436 | 1.075245±0.062007 | 1.675768±0.180436 | 1.077623±0.060459 | 1.678699±0.184216 | 1.078530±0.061433 | 1.604684±0.198098 | 1.051343±0.067399 |
| Timer-XL | 1.682402±0.184814 | 1.079726±0.061458 | 1.695550±0.182665 | 1.083466±0.060714 | 1.675768±0.180436 | 1.075245±0.062007 | 1.675768±0.184216 | 1.077623±0.060459 | 1.678699±0.184216 | 1.078530±0.061433 | 1.604684±0.198098 | 1.051343±0.067399 |
| Time-LLM | 1.117808±0.006952 | 0.861796±0.009912 | 1.110061±0.008093 | 0.859130±0.010330 | 1.116355±0.016937 | 0.862264±0.006962 | 1.115380±0.013578 | 0.861516±0.007815 | 1.118824±0.009418 | 0.861964±0.009246 | 1.140709±0.040012 | 0.869599±0.002972 |
| TTM | 0.654428±0.395470 | 0.616692±0.263311 | 0.640631±0.386685 | 0.601005±0.277563 | 0.672607±0.406562 | 0.625470±0.267353 | 0.659575±0.398653 | 0.619226±0.264545 | 0.658020±0.397420 | 0.618401±0.264000 | 0.754064±0.456014 | 0.661313±0.282751 |
| TimesFM | 0.001636±0.000136 | 0.029680±0.001285 | 0.002485±0.000161 | 0.036807±0.001243 | 0.002067±0.000150 | 0.033477±0.001342 | 0.002015±0.000146 | 0.032912±0.001363 | 0.000553±0.000078 | 0.057232±0.000556 | 0.004519±0.000062 | 0.051329±0.000248 |
| MOMENT | 0.123636±0.001532 | 0.185864±0.002616 | 0.122294±0.000905 | 0.183943±0.002491 | 0.123930±0.002353 | 0.187262±0.003303 | 0.122701±0.002042 | 0.185370±0.002825 | 0.123083±0.000951 | 0.186004±0.002056 | 0.126537±0.001335 | 0.192803±0.002727 |
| MOIRAI | 0.045974±0.020236 | 0.153263±0.039310 | 0.044596±0.019787 | 0.151022±0.039028 | 0.048091±0.021230 | 0.156614±0.040513 | 0.046449±0.020615 | 0.153945±0.039890 | 0.046381±0.020526 | 0.153706±0.039710 | 0.056150±0.023934 | 0.168997±0.042707 |
| GRU | 0.003019±0.002254 | 0.039934±0.015352 | 0.003514±0.002456 | 0.043060±0.015615 | 0.003380±0.002413 | 0.042136±0.015570 | 0.003320±0.002384 | 0.041905±0.015601 | 0.004967±0.003030 | 0.051602±0.016209 | 0.004470±0.002861 | 0.048812±0.016116 |
| LSTM | 0.007199±0.000884 | 0.063775±0.005590 | 0.007914±0.000923 | 0.066870±0.005459 | 0.007720±0.000909 | 0.065937±0.005459 | 0.007651±0.000877 | 0.065695±0.005445 | 0.009889±0.000977 | 0.074740±0.005136 | 0.009219±0.000932 | 0.072087±0.005220 |
| DLinear | 0.003976±0.000072 | 0.046282±0.000438 | 0.004791±0.000087 | 0.050889±0.000489 | 0.004507±0.000078 | 0.049333±0.000451 | 0.004431±0.000079 | 0.048892±0.000453 | 0.007188±0.000135 | 0.062542±0.000603 | 0.006364±0.000118 | 0.058742±0.000564 |
| SparseTSF | 0.006651±0.000025 | 0.067097±0.000086 | 0.008265±0.000029 | 0.074715±0.000094 | 0.007647±0.000028 | 0.072035±0.000090 | 0.007508±0.000027 | 0.071319±0.000087 | 0.012950±0.000043 | 0.093697±0.000101 | 0.011511±0.000038 | 0.088247±0.000096 |
| FedFormer | 0.034245±0.002066 | 0.154651±0.004033 | 0.033823±0.001856 | 0.153655±0.003645 | 0.033937±0.001901 | 0.153883±0.003678 | 0.033982±0.001942 | 0.153922±0.003766 | 0.035240±0.001159 | 0.149787±0.002141 | 0.032867±0.001340 | 0.150784±0.002407 |
| DAF (Ours) | **0.001287±0.000041** | **0.027384±0.000098** | **0.001492±0.000098** | **0.029428±0.000279** | **0.001425±0.000044** | **0.028782±0.000190** | **0.001403±0.000045** | **0.028552±0.000172** | **0.002375±0.000064** | **0.036307±0.000607** | **0.001906±0.000085** | **0.033176±0.000648** |
| Improvements | 21.35±2.01 | 7.74±1.71 | 20.01±1.51 | — | -18.75±2.31 | -33.64±1.07 | 22.10±3.78 | 5.08±2.11 | 27.96±2.27 | 11.36±1.67 | 28.58±2.23 | 32.06±2.08 |

| Dataset | LEMUR-44 | | LEMUR-45 | | LEMUR-46 | | LEMUR-47 | | LEMUR-48 | | LEMUR-49 | |
|---|---|---|---|---|---|---|---|---|---|---|---|---|
| Metric | MSE | MAE | MSE | MAE | MSE | MAE | MSE | MAE | MSE | MAE | MSE | MAE |
| Time-MoE | 2.310422±0.023128 | 1.211563±0.009653 | 2.315281±0.021154 | 1.212630±0.009007 | 2.310555±0.024082 | 1.210794±0.010046 | 2.304965±0.024113 | 1.209574±0.010162 | 2.319285±0.025710 | 1.212285±0.010458 | 2.315620±0.028105 | 1.209729±0.011031 |
| Timer | 1.682402±0.184814 | 1.079726±0.061458 | 0.936558±0.006027 | 0.795440±0.002551 | 0.934442±0.006579 | 0.794586±0.002551 | 0.950515±0.006212 | 0.801915±0.002625 | 0.952643±0.006429 | 0.802719±0.003101 | 0.956307±0.006938 | 0.804037±0.002976 |
| Timer-XL | 1.682402±0.184814 | 1.079726±0.061458 | 0.936558±0.182665 | 0.795440±0.002551 | 0.934442±0.006579 | 0.794586±0.002551 | 0.950515±0.006212 | 0.801915±0.002625 | 0.952643±0.006429 | 0.802719±0.003101 | 0.956307±0.006938 | 0.804037±0.002976 |
| Time-LLM | 1.132033±0.034144 | 0.866556±0.001320 | 1.138839±0.036258 | 0.868469±0.001677 | 1.135118±0.031368 | 0.866715±0.001124 | 1.136083±0.027854 | 0.866715±0.001404 | 1.133695±0.031599 | 0.865852±0.001348 | 1.143184±0.041011 | 0.867747±0.003334 |
| TTM | 0.732113±0.442163 | 0.651554±0.277709 | 0.730758±0.441422 | 0.651005±0.277563 | 0.722497±0.436693 | 0.646925±0.276375 | 0.722154±0.436693 | 0.646623±0.276375 | 0.721313±0.435627 | 0.646623±0.275785 | 0.738255±0.450694 | 0.651412±0.283096 |
| TimesFM | 0.004617±0.000045 | 0.051759±0.000164 | 0.004127±0.000059 | 0.048960±0.000338 | 0.004173±0.000058 | 0.049036±0.000293 | 0.004108±0.000084 | 0.048744±0.000450 | 0.005613±0.000115 | 0.057459±0.000725 | 0.005585±0.000086 | 0.057121±0.000526 |
| MOMENT | 0.125614±0.001249 | 0.190772±0.002903 | 0.124748±0.002280 | 0.190367±0.003166 | 0.125902±0.001187 | 0.189116±0.002350 | 0.125902±0.001187 | 0.190185±0.002089 | 0.124233±0.001272 | 0.188993±0.002089 | 0.127741±0.001928 | 0.193865±0.003136 |
| MOIRAI | 0.045454±0.022981 | 0.164797±0.041824 | 0.053791±0.023132 | 0.165333±0.041967 | 0.052668±0.022792 | 0.163543±0.041535 | 0.052675±0.022779 | 0.163370±0.041535 | 0.052767±0.022841 | 0.163688±0.041669 | 0.055747±0.023124 | 0.168397±0.040832 |
| GRU | 0.004614±0.000949 | 0.049518±0.016149 | 0.004366±0.002828 | 0.048320±0.016145 | 0.004413±0.002828 | 0.048508±0.016061 | 0.004413±0.002836 | 0.048502±0.016143 | 0.005895±0.001029 | 0.056502±0.003480 | 0.006011±0.001051 | 0.056817±0.003487 |
| LSTM | 0.009400±0.000949 | 0.072689±0.005187 | 0.009049±0.000947 | 0.071468±0.005244 | 0.009151±0.000967 | 0.071757±0.005244 | 0.009151±0.000967 | 0.071757±0.005244 | 0.007507±0.002878 | 0.063372±0.013383 | 0.007612±0.002863 | 0.063728±0.013396 |
| DLinear | 0.006501±0.000119 | 0.060134±0.000560 | 0.006134±0.000114 | 0.057658±0.000551 | 0.006183±0.000112 | 0.057852±0.000547 | 0.006168±0.000112 | 0.057730±0.000538 | 0.007229±0.000137 | 0.062339±0.000610 | 0.007244±0.000133 | 0.062340±0.000590 |
| SparseTSF | 0.011561±0.000038 | 0.010944±0.000037 | 0.010944±0.000037 | 0.086001±0.000095 | 0.010712±0.000037 | 0.085357±0.000095 | 0.013220±0.000043 | 0.094184±0.000100 | 0.013220±0.000043 | 0.094184±0.000100 | 0.012841±0.000043 | 0.092858±0.000103 |
| FedFormer | 0.032967±0.001295 | 0.150894±0.002314 | 0.033004±0.001353 | 0.151092±0.002412 | 0.033128±0.000400 | 0.151308±0.002343 | 0.033147±0.001326 | 0.149361±0.001221 | 0.032542±0.001221 | 0.149361±0.002161 | 0.032663±0.001254 | 0.149669±0.002177 |
| DAF (Ours) | **0.001983±0.000078** | **0.033564±0.000582** | **0.001791±0.000076** | **0.032370±0.000558** | **0.001830±0.000076** | **0.032531±0.000565** | **0.001833±0.000075** | **0.036505±0.000614** | **0.002434±0.000065** | **0.036505±0.000614** | **0.002452±0.000065** | **0.036589±0.000630** |
| Improvements | 57.02±3.53 | 32.24±2.09 | 39.98±2.16 | 20.01±1.51 | 56.15±0.36 | 32.97±2.06 | 55.39±0.45 | 32.99±2.09 | 56.65±0.36 | 35.36±1.12 | 56.09±0.28 | 35.60±1.11 |

Table 20: Zero-shot evaluation on individual satellites of two cross-domain datasets (LEMUR and SKYSAT) with mean and standard deviation in MSE and MAE. A dash ('-') indicates that our model fails to achieve either the best or second-best performance. "Improvements" presents the relative performance of our model compared to the best baseline for each dataset and metric. The best result is highlighted in **bold**, and the second-best is highlighted with underline.

| Dataset | LEMUR-50 MSE | LEMUR-50 MAE | LEMUR-51 MSE | LEMUR-51 MAE | LEMUR-52 MSE | LEMUR-52 MAE | LEMUR-53 MSE | LEMUR-53 MAE | LEMUR-54 MSE | LEMUR-54 MAE | LEMUR-55 MSE | LEMUR-55 MAE |
|---|---|---|---|---|---|---|---|---|---|---|---|---|
| Time-MoE | 2.262186±0.021645 | 1.177649±0.007740 | 2.270557±0.021837 | 1.181452±0.008054 | 2.270776±0.024840 | 1.181442±0.008499 | 2.263674±0.026764 | 1.177746±0.008657 | 2.269115±0.028305 | 1.180724±0.010799 | 2.245731±0.024363 | 1.171082±0.008491 |
| Timer | 1.456705±0.005265 | 0.993270±0.001740 | 1.390327±0.004634 | 0.969750±0.001738 | 1.409772±0.004637 | 0.976550±0.001670 | 1.449399±0.003715 | 0.989995±0.001483 | 1.418423±0.006092 | 0.979079±0.002248 | 1.596563±0.003556 | 1.039972±0.001411 |
| Timer-XL | 1.874711±0.103555 | 1.131665±0.032705 | 1.852936±0.115626 | 1.124566±0.036549 | 1.858761±0.108848 | 1.124566±0.034176 | 1.872772±0.109992 | 1.130132±0.034541 | 1.864888±0.112155 | 1.127624±0.035398 | 1.932366±0.090335 | 1.148481±0.027956 |
| Time-LLM | 1.066624±0.051435 | 0.832592±0.033500 | 1.072625±0.043632 | 0.835038±0.030418 | 1.073021±0.044481 | 0.835159±0.030646 | 1.067641±0.049129 | 0.832386±0.032874 | 1.068855±0.043839 | 0.833304±0.030766 | 1.050314±0.069734 | 0.824684±0.041809 |
| TTM | 0.369845±0.220385 | 0.463223±0.193351 | 0.619471±0.373393 | 0.605639±0.257098 | 0.671192±0.404603 | 0.629693±0.267395 | 0.664440±0.401077 | 0.626431±0.266567 | 0.000000±0.000000 | 0.000000±0.000000 | 0.000000±0.000000 | 0.000000±0.000000 |
| TimesFM | 0.005239±0.000047 | 0.055316±0.000321 | 0.005000±0.000033 | 0.059064±0.000218 | 0.004694±0.000050 | 0.052256±0.000046 | 0.005029±0.000050 | 0.054245±0.000253 | 0.001283±0.000139 | 0.025946±0.001323 | 0.001254±0.000136 | 0.025788±0.001288 |
| MOMENT | 0.118409±0.000509 | 0.172925±0.001420 | 0.118135±0.000867 | 0.173576±0.001744 | 0.118875±0.001909 | 0.173934±0.002824 | 0.119036±0.001382 | 0.173310±0.002531 | 0.118428±0.000849 | 0.173338±0.002284 | 0.116032±0.001845 | 0.168794±0.002920 |
| MOIRAI | 0.030350±0.013680 | 0.124132±0.031924 | 0.032605±0.014574 | 0.128383±0.032879 | 0.031861±0.014306 | 0.127006±0.032615 | 0.030633±0.013839 | 0.124429±0.032192 | 0.031675±0.014315 | 0.126554±0.032681 | 0.025866±0.011580 | 0.114681±0.029235 |
| GRU | 0.005783±0.001049 | 0.055945±0.003537 | 0.005799±0.001056 | 0.055739±0.003560 | 0.005543±0.001037 | 0.054714±0.003596 | 0.005700±0.001034 | 0.055522±0.003551 | 0.001192±0.000541 | 0.025928±0.006159 | 0.001169±0.000550 | 0.025759±0.006256 |
| LSTM | 0.007353±0.002806 | 0.062792±0.013335 | 0.007348±0.002818 | 0.062605±0.013363 | 0.007079±0.002773 | 0.059607±0.013354 | 0.007255±0.002802 | 0.062337±0.013335 | 0.004252±0.001683 | 0.047894±0.009909 | 0.004195±0.001650 | 0.047495±0.009816 |
| DLinear | 0.006956±0.000129 | 0.061157±0.000587 | 0.006868±0.000125 | 0.060618±0.000571 | 0.006596±0.000121 | 0.059607±0.000574 | 0.006831±0.000125 | 0.060551±0.000574 | 0.002473±0.000042 | 0.035949±0.000324 | 0.002415±0.000041 | 0.035464±0.000326 |
| SparseTSF | 0.012238±0.000902 | 0.090655±0.000102 | 0.012020±0.000039 | 0.089806±0.000039 | 0.011588±0.000039 | 0.088302±0.000099 | 0.012070±0.000004 | 0.090014±0.000098 | 0.003812±0.000017 | 0.050441±0.000086 | 0.003603±0.000017 | 0.049222±0.000090 |
| FredFormer | 0.032808±0.001347 | 0.150009±0.002330 | 0.032888±0.001384 | 0.150242±0.002424 | 0.032888±0.001400 | 0.150371±0.002501 | 0.032786±0.001324 | 0.150025±0.002256 | 0.035419±0.002615 | 0.157611±0.005270 | 0.035441±0.002622 | 0.157664±0.005285 |
| DAF (Ours) | **0.002218±0.000091** | **0.035246±0.000736** | **0.002179±0.000076** | **0.034874±0.000657** | **0.002179±0.000000** | **0.033781±0.000732** | **0.002184±0.000075** | **0.034755±0.000675** | **0.000998±0.000046** | **0.022596±0.000227** | **0.000965±0.000051** | **0.023333±0.000227** |
| Improvements | 57.68±0.56 | 36.27±0.52 | 56.42±0.41 | 35.30±0.38 | 57.39±0.36 | 35.35±0.07 | 57.36±0.59 | 35.92±0.40 | 16.32±4.57 | 9.01±23.93 | 17.45±4.64 | 9.41±24.28 |

| Dataset | LEMUR-56 MSE | LEMUR-56 MAE | LEMUR-57 MSE | LEMUR-57 MAE | LEMUR-58 MSE | LEMUR-58 MAE | LEMUR-59 MSE | LEMUR-59 MAE | LEMUR-60 MSE | LEMUR-60 MAE | LEMUR-61 MSE | LEMUR-61 MAE |
|---|---|---|---|---|---|---|---|---|---|---|---|---|
| Time-MoE | 2.226021±0.023306 | 1.167437±0.007895 | 2.133689±0.017159 | 1.134209±0.005950 | 2.226061±0.016911 | 1.167599±0.006040 | 2.185401±0.020653 | 1.152031±0.007701 | 2.120949±0.015184 | 1.167181±0.004279 | 2.124512±0.022187 | 1.123994±0.007318 |
| Timer | 1.823798±0.002460 | 1.116722±0.000926 | 2.262057±0.005035 | 1.337182±0.000739 | 1.806608±0.001634 | 1.111795±0.000793 | 2.170905±0.000705 | 1.21977±0.000162 | 2.628155±0.006602 | 1.316072±0.001955 | 2.608792±0.005341 | 1.330441±0.001177 |
| Timer-XL | 1.981719±0.104828 | 1.167001±0.031754 | 2.355040±0.003900 | 1.275336±0.002049 | 1.975695±0.108719 | 1.165562±0.032977 | 2.133608±0.064191 | 1.21971±0.019221 | 2.394136±0.014842 | 1.259023±0.002931 | 2.368578±0.009649 | 1.270157±0.001812 |
| Time-LLM | 1.020443±0.100379 | 0.814670±0.056371 | 0.967904±0.166656 | 0.789511±0.088232 | 1.024458±0.098862 | 0.816201±0.055487 | 0.991568±0.141623 | 0.800104±0.075521 | 0.968342±0.168026 | 0.773080±0.085769 | 0.966455±0.169121 | 0.782371±0.089321 |
| TTM | 0.000000±0.000000 | 0.000000±0.000000 | 0.000000±0.000000 | 0.000000±0.000000 | 0.000000±0.000000 | 0.000000±0.000000 | 0.000000±0.000000 | 0.000000±0.000000 | 0.000000±0.000000 | 0.000000±0.000000 | 0.000000±0.000000 | 0.000000±0.000000 |
| TimesFM | 0.001269±0.000134 | 0.025916±0.001280 | 0.001257±0.000132 | 0.025856±0.001282 | 0.001281±0.000138 | 0.026686±0.001328 | 0.001281±0.000138 | 0.026033±0.001329 | 0.001277±0.000142 | 0.026055±0.001371 | 0.001250±0.000141 | 0.025769±0.001362 |
| MOMENT | 0.114297±0.000434 | 0.164950±0.001543 | 0.111667±0.000682 | 0.152592±0.000221 | 0.115577±0.001618 | 0.165987±0.002465 | 0.112348±0.000269 | 0.158975±0.002066 | 0.112153±0.000269 | 0.154634±0.000113 | 0.113387±0.001505 | 0.156844±0.000776 |
| MOIRAI | 0.019855±0.008316 | 0.101452±0.023908 | 0.010085±0.004574 | 0.072949±0.011058 | 0.012338±0.008391 | 0.102338±0.023998 | 0.011798±0.008451 | 0.084949±0.016666 | 0.009600±0.002470 | 0.070019±0.009777 | 0.009676±0.002659 | 0.071177±0.010537 |
| GRU | 0.001162±0.000527 | 0.025675±0.006099 | 0.001182±0.000538 | 0.025860±0.006117 | 0.001194±0.000578 | 0.025818±0.006414 | 0.001164±0.000532 | 0.025686±0.006125 | 0.001143±0.000541 | 0.025513±0.006214 | 0.001164±0.000543 | 0.025630±0.006267 |
| LSTM | 0.004171±0.001645 | 0.047210±0.009841 | 0.004195±0.001659 | 0.047486±0.009845 | 0.004278±0.001575 | 0.047920±0.009846 | 0.004278±0.001575 | 0.046891±0.009846 | 0.004062±0.001616 | 0.046708±0.009810 | 0.004151±0.001638 | 0.047140±0.009843 |
| DLinear | 0.002365±0.000039 | 0.035145±0.000311 | 0.002401±0.000039 | 0.035282±0.000312 | 0.002298±0.000039 | 0.036021±0.000313 | 0.002298±0.000039 | 0.034483±0.000313 | 0.002228±0.000038 | 0.034385±0.000312 | 0.002330±0.000039 | 0.034738±0.000316 |
| SparseTSF | 0.003443±0.000017 | 0.048161±0.000087 | 0.003509±0.000016 | 0.048563±0.000087 | 0.003407±0.000016 | 0.047616±0.000088 | 0.003458±0.000016 | 0.047616±0.000090 | 0.003407±0.000090 | 0.047967±0.000090 | 0.003412±0.000017 | 0.047889±0.000090 |
| FredFormer | 0.035521±0.002661 | 0.157894±0.005366 | 0.035536±0.002684 | 0.157907±0.005447 | 0.035580±0.002779 | 0.158399±0.005354 | 0.035586±0.002652 | 0.157920±0.005354 | 0.035536±0.002652 | 0.157920±0.005409 | 0.035522±0.002679 | 0.157886±0.005409 |
| DAF (Ours) | **0.000941±0.000056** | **0.023609±0.000291** | **0.000983±0.000054** | **0.023352±0.000260** | **0.001139±0.000061** | **0.023881±0.000183** | **0.000959±0.000061** | **0.022995±0.000310** | **0.000931±0.000053** | **0.022857±0.000286** | **0.000931±0.000054** | **0.023106±0.000236** |
| Improvements | 18.99±4.83 | 10.15±23.92 | 16.83±4.66 | 9.67±4.74 | 57.39±0.36 | 35.35±0.07 | 16.32±4.57 | 10.47±24.15 | 18.55±5.10 | 10.36±24.12 | 17.45±5.10 | 9.57±24.01 |

| Dataset | LEMUR-62 MSE | LEMUR-62 MAE | SKYSAT-1 MSE | SKYSAT-1 MAE | SKYSAT-2 MSE | SKYSAT-2 MAE | SKYSAT-3 MSE | SKYSAT-3 MAE | SKYSAT-4 MSE | SKYSAT-4 MAE | SKYSAT-5 MSE | SKYSAT-5 MAE |
|---|---|---|---|---|---|---|---|---|---|---|---|---|
| Time-MoE | 2.136297±0.015111 | 1.142362±0.006551 | 2.111134±0.019578 | 1.086436±0.006624 | 2.080526±0.012200 | 1.058353±0.004776 | 2.286254±0.026995 | 1.217689±0.010161 | 2.340615±0.035415 | 1.220377±0.013052 | 2.233311±0.024398 | 1.195440±0.008174 |
| Timer | 2.600266±0.004903 | 1.348083±0.000715 | 2.676572±0.006479 | 1.309843±0.001399 | 2.865756±0.006953 | 1.330818±0.001415 | 1.267273±0.006282 | 0.946987±0.002342 | 0.775558±0.005642 | 0.719273±0.002583 | 1.732426±0.002538 | 1.110940±0.000999 |
| Timer-XL | 2.363068±0.007367 | 1.287237±0.002673 | 2.435532±0.035400 | 1.251503±0.007971 | 2.548262±0.072837 | 1.257523±0.017217 | 1.765544±0.168296 | 1.123196±0.055624 | 1.567822±0.236398 | 1.187701±0.079871 | 1.954305±0.120633 | 1.182754±0.038175 |
| Time-LLM | 0.000000±0.000000 | 0.000000±0.000000 | 0.000000±0.000000 | 0.000000±0.000000 | 0.000000±0.000000 | 0.000000±0.000000 | 1.087394±0.020468 | 0.862316±0.021249 | 1.156895±0.062922 | 0.875046±0.011415 | 1.034235±0.089226 | 0.837520±0.052013 |
| TTM | 0.000000±0.000000 | 0.000000±0.000000 | 0.000000±0.000000 | 0.000000±0.000000 | 0.000000±0.000000 | 0.000000±0.000000 | 0.543411±0.327916 | 0.570112±0.243046 | 0.797171±0.483974 | 0.680548±0.291897 | 0.321564±0.190893 | 0.437896±0.181363 |
| TimesFM | 0.004652±0.000025 | 0.052094±0.000023 | 0.001231±0.000090 | 0.026190±0.001019 | 0.009429±0.000441 | 0.075920±0.001804 | 0.001311±0.000121 | 0.027051±0.001286 | 0.001293±0.000117 | 0.026274±0.001249 | 0.001277±0.000101 | 0.026694±0.001158 |
| MOMENT | 0.111939±0.000867 | 0.158019±0.000471 | 0.111822±0.000471 | 0.152536±0.001683 | 0.112604±0.002988 | 0.151079±0.002476 | 0.120210±0.000851 | 0.182154±0.001790 | 0.129533±0.002483 | 0.199487±0.003743 | 0.116812±0.001569 | 0.171588±0.002162 |
| MOIRAI | 0.009893±0.002762 | 0.072845±0.010937 | 0.009282±0.002417 | 0.067767±0.009505 | 0.009220±0.002232 | 0.066258±0.008451 | 0.037525±0.016833 | 0.140818±0.036409 | 0.062878±0.025770 | 0.179428±0.043562 | 0.022394±0.009745 | 0.109753±0.027158 |
| GRU | 0.005524±0.001026 | 0.054680±0.003599 | 0.001947±0.001504 | 0.031954±0.012274 | 0.006260±0.000655 | 0.058182±0.005067 | 0.001536±0.000573 | 0.030375±0.006029 | 0.002052±0.000788 | 0.035136±0.006507 | 0.001978±0.000772 | 0.034376±0.006244 |
| LSTM | 0.007050±0.002751 | 0.061519±0.013338 | 0.005685±0.000929 | 0.056436±0.005814 | 0.009211±0.003178 | 0.073850±0.012879 | 0.002568±0.000824 | 0.037603±0.006848 | 0.001787±0.000609 | 0.031681±0.006143 | 0.001892±0.000687 | 0.032641±0.006800 |
| DLinear | 0.006558±0.000122 | 0.059372±0.000569 | 0.002425±0.000039 | 0.036103±0.000315 | 0.010053±0.000193 | 0.073689±0.000719 | 0.000404±0.000004 | 0.012550±0.000095 | **0.000186±0.000002** | **0.008072±0.000052** | **0.000151±0.000001** | **0.007542±0.000048** |
| SparseTSF | 0.011516±0.000039 | 0.087970±0.000089 | 0.003646±0.000017 | 0.049856±0.000092 | 0.018016±0.000059 | 0.104446±0.000115 | **0.000372±0.000002** | **0.012955±0.000054** | 0.000236±0.000002 | 0.008419±0.000094 | 0.000236±0.000094 | 0.008146±0.000118 |
| FredFormer | 0.032949±0.001401 | 0.150427±0.002415 | 0.035484±0.002682 | 0.159637±0.005553 | 0.031533±0.000981 | 0.146373±0.002211 | 0.036786±0.002361 | 0.162311±0.004800 | 0.036352±0.002233 | 0.159841±0.004465 | 0.036295±0.002248 | 0.161210±0.004531 |
| DAF (Ours) | **0.001968±0.000091** | **0.033520±0.000691** | **0.001046±0.000059** | **0.024278±0.000310** | **0.004205±0.000068** | **0.044834±0.000338** | 0.000379±0.000008 | 0.013720±0.000168 | 0.000236±0.000003 | 0.011063±0.000035 | 0.000337±0.000004 | 0.010730±0.000015 |
| Improvements | 57.69±1.21 | 35.66±0.30 | 14.99±2.31 | 7.16±3.91 | 32.83±1.22 | 22.97±3.19 | 17.58±5.11 | 10.47±24.15 | -1.88±6.57 | - | 0.47±1.90 | - |

| Dataset | SKYSAT-6 MSE | SKYSAT-6 MAE | SKYSAT-7 MSE | SKYSAT-7 MAE | SKYSAT-8 MSE | SKYSAT-8 MAE | SKYSAT-9 MSE | SKYSAT-9 MAE | SKYSAT-10 MSE | SKYSAT-10 MAE | SKYSAT-11 MSE | SKYSAT-11 MAE |
|---|---|---|---|---|---|---|---|---|---|---|---|---|
| Time-MoE | 2.217263±0.019278 | 1.180680±0.006047 | 2.220786±0.019890 | 1.190728±0.006394 | 2.231420±0.021367 | 1.184158±0.007101 | 2.222228±0.022333 | 1.186060±0.005789 | 2.214315±0.017316 | 1.186603±0.006443 | 2.235809±0.016941 | 1.190592±0.005571 |
| Timer | 1.868431±0.001307 | 1.146680±0.000489 | 1.881500±0.002464 | 1.159767±0.000875 | 1.779060±0.003022 | 1.116707±0.001046 | 1.895952±0.001073 | 1.161599±0.000432 | 1.928632±0.000609 | 1.171489±0.000760 | 1.770373±0.001781 | 1.119076±0.000597 |
| Timer-XL | 2.022161±0.109123 | 1.195439±0.033480 | 2.015142±0.101245 | 1.202413±0.031837 | 1.982984±0.114786 | 1.181703±0.035840 | 2.026253±0.107151 | 1.203152±0.033102 | 2.044122±0.077595 | 1.207883±0.023977 | 1.976385±0.097424 | 1.184527±0.030514 |
| Time-LLM | 1.020665±0.107893 | 0.825362±0.059994 | 1.018410±0.109446 | 0.830455±0.061419 | 1.026822±0.097467 | 0.826892±0.055510 | 1.015196±0.111022 | 0.827580±0.062461 | 1.016400±0.115919 | 0.827222±0.064141 | 1.028865±0.094042 | 0.832256±0.054035 |
| TTM | 0.262203±0.154364 | 0.392900±0.160408 | 0.260168±0.153143 | 0.393403±0.160617 | 0.297830±0.176471 | 0.417995±0.172618 | 0.252920±0.148528 | 0.387470±0.157627 | 0.263772±0.154379 | 0.394640±0.150090 | 0.330730±0.195906 | 0.441205±0.181703 |
| TimesFM | 0.001353±0.000123 | 0.027105±0.001201 | 0.001323±0.000119 | 0.027026±0.001222 | 0.001380±0.000158 | 0.027333±0.001374 | 0.001416±0.000159 | 0.027717±0.001348 | 0.001363±0.000146 | 0.027130±0.001281 | 0.001381±0.000149 | 0.027337±0.001343 |
| MOMENT | 0.115609±0.001189 | 0.167293±0.002537 | 0.115505±0.000942 | 0.168345±0.001918 | 0.115788±0.001407 | 0.168648±0.001783 | 0.114001±0.001090 | 0.167108±0.001893 | 0.114527±0.000926 | 0.169935±0.002177 | 0.115862±0.000765 | 0.167337±0.001343 |
| MOIRAI | 0.018938±0.007857 | 0.100585±0.023619 | 0.018712±0.007785 | 0.100753±0.023721 | 0.021146±0.009000 | 0.105818±0.025625 | 0.018596±0.007743 | 0.100201±0.023629 | 0.018055±0.007464 | 0.098607±0.022953 | 0.021611±0.009421 | 0.107302±0.026582 |
| GRU | 0.001704±0.000642 | 0.032151±0.006237 | 0.002081±0.000810 | 0.035320±0.006422 | 0.001387±0.000810 | 0.029504±0.006422 | 0.001108±0.000421 | 0.025952±0.004935 | 0.001214±0.000298 | 0.027419±0.003415 | 0.001113±0.000446 | 0.026060±0.005216 |
| LSTM | 0.002122±0.000698 | 0.034060±0.006331 | 0.001785±0.000668 | 0.032245±0.006331 | 0.001785±0.000668 | 0.032254±0.007242 | 0.001785±0.000668 | 0.032254±0.007242 | 0.002010±0.000573 | 0.034000±0.006573 | 0.002284±0.000652 | 0.036137±0.006422 |
| DLinear | 0.000324±0.000003 | 0.010737±0.000080 | 0.000167±0.000001 | 0.008237±0.000048 | 0.000167±0.000002 | 0.007638±0.000052 | **0.000167±0.000002** | **0.007638±0.000052** | 0.000224±0.000002 | 0.008741±0.000056 | **0.000320±0.000006** | **0.011169±0.000078** |
| SparseTSF | 0.000386±0.000088 | 0.011475±0.000009 | 0.000250±0.000001 | 0.008562±0.000120 | 0.000257±0.000001 | 0.009233±0.000142 | 0.000345±0.000000 | 0.011399±0.000081 | 0.000268±0.000002 | 0.008726±0.000062 | 0.000338±0.000000 | 0.011751±0.000013 |
| FredFormer | 0.036710±0.002314 | 0.160432±0.004544 | 0.036086±0.002178 | 0.160505±0.004390 | 0.036086±0.002178 | 0.160174±0.004305 | 0.036743±0.002331 | 0.161355±0.004633 | 0.036505±0.002290 | 0.160577±0.004539 | 0.036728±0.002329 | 0.161271±0.004628 |
| DAF (Ours) | 0.000337±0.000005 | 0.012767±0.000097 | 0.000260±0.000005 | 0.011116±0.000034 | **0.000214±0.000004** | **0.010520±0.000094** | 0.000261±0.000005 | 0.011421±0.000058 | 0.000261±0.000005 | 0.013075±0.000122 | 0.000320±0.000005 | 0.012798±0.000193 |
| Improvements | -4.01±0.86 | - | 14.69±2.31 | 7.16±3.91 | 32.83±1.22 | 22.97±3.19 | -16.52±2.51 | - | 0.00±1.17 | - | 0.00±1.56 | - |

Table 21: Zero-shot evaluation on individual satellites of SKYSAT dataset with mean and standard deviation in MSE and MAE. "Improvements" presents the relative performance of our model compared to the best baseline for each dataset and metric. A dash ('-') indicates that our model fails to achieve either the best or second-best performance. The best result is highlighted in **bold**, and the second-best is highlighted with underline.

| Dataset | SKYSAT-12 | | SKYSAT-13 | | SKYSAT-14 | | SKYSAT-15 | |
|---|---|---|---|---|---|---|---|---|
| Metric | MSE | MAE | MSE | MAE | MSE | MAE | MSE | MAE |
| Time-MoE | 2.236464±0.017642 | 1.190623±0.005391 | 2.240123±0.017810 | 1.193063±0.005769 | 2.222521±0.019208 | 1.183334±0.006306 | 2.187511±0.016989 | 1.176568±0.005243 |
| Timer | 1.773883±0.002530 | 1.120311±0.000622 | 1.738289±0.001117 | 1.109440±0.000759 | 1.870675±0.001904 | 1.147982±0.000934 | 2.157144±0.001933 | 1.241508±0.000774 |
| Timer-XL | 1.979871±0.097502 | 1.185919±0.030516 | 1.961218±0.097415 | 1.180946±0.030776 | 2.020076±0.108422 | 1.195464±0.033309 | 2.133858±0.074213 | 1.237540±0.022912 |
| TTM | 0.329865±0.195286 | 0.440593±0.181416 | 0.346014±0.205206 | 0.451804±0.186507 | 0.262402±0.154509 | 0.393302±0.160666 | 0.158082±0.090164 | 0.305651±0.118242 |
| TimesFM | 0.001435±0.000168 | 0.027763±0.001373 | 0.001407±0.000161 | 0.027686±0.001365 | 0.001317±0.000122 | 0.026855±0.001296 | 0.001418±0.000091 | 0.028006±0.000827 |
| MOMENT | 0.115225±0.000494 | 0.169201±0.001709 | 0.116160±0.000915 | 0.170774±0.002496 | 0.114590±0.001089 | 0.167235±0.002416 | 0.113804±0.000451 | 0.163318±0.001895 |
| MOIRAI | 0.021122±0.009072 | 0.106200±0.025822 | 0.022382±0.009738 | 0.109233±0.026976 | 0.019134±0.008025 | 0.101078±0.023960 | 0.013851±0.005079 | 0.087202±0.017715 |
| GRU | 0.001125±0.000394 | 0.026258±0.004584 | 0.001105±0.000493 | 0.025847±0.005708 | 0.002014±0.000767 | 0.034824±0.006402 | 0.003570±0.001394 | 0.045992±0.008385 |
| LSTM | 0.002181±0.000615 | 0.035355±0.006311 | 0.002395±0.000616 | 0.036996±0.006034 | 0.001860±0.000630 | 0.032204±0.006240 | 0.001476±0.000497 | 0.029140±0.006224 |
| DLinear | **0.000306±0.000003** | **0.010729±0.000078** | 0.000399±0.000003 | 0.012524±0.000086 | **0.000228±0.000002** | 0.008665±0.000052 | 0.000224±0.000001 | 0.009644±0.000039 |
| SparseTSF | 0.000408±0.000001 | 0.012135±0.000031 | 0.000351±0.000001 | **0.012471±0.000046** | 0.000233±0.000001 | **0.008454±0.000088** | 0.000419±0.000007 | 0.015087±0.000172 |
| FredFormer | 0.036736±0.002332 | 0.161403±0.004648 | 0.036799±0.002329 | 0.161707±0.004668 | 0.036341±0.002247 | 0.159876±0.004398 | 0.034907±0.001832 | 0.157590±0.003574 |
| **DAF (Ours)** | 0.000314±0.000002 | 0.012669±0.000110 | **0.000366±0.000008** | 0.013558±0.000193 | 0.000269±0.000007 | 0.011493±0.000078 | **0.000179±0.000011** | **0.009352±0.000686** |
| Improvements | -2.55±0.76 | - | 8.26±2.22 | -8.23±1.80 | - | - | 20.09±6.43 | 3.03±7.13 |

