# OpenReview forum: "Decomposed Attention FredFormer: Large Time-series Prediction Model for Satellite Orbit Prediction"
_ICLR.cc/2026/Conference — Submitted to ICLR 2026_

### Official Review · Reviewer_qjRj · 2025-10-25

**Soundness:** 3
**Presentation:** 3
**Contribution:** 2
**Rating:** 4
**Confidence:** 3

**Summary:**

This paper presents Decomposed Attention FredFormer (DAF), a large time-series forecasting model designed for satellite orbit prediction.  Built upon FredFormer, DAF introduces three main modifications:
+ replacing DFT with RFFT/IRFFT to leverage the symmetry of real-valued inputs.
+ removing patching and layer normalization while adding positional embeddings to better capture orbital periodicity.
+ integrating Tensor Train Decomposition (TTD) into multi-head attention for parameter compression.

Trained on large-scale Starlink data, DAF achieves strong zero-shot performance. The method demonstrates high efficiency and promising generalization across diverse orbital regimes.

**Strengths:**

+ Clear motivation and practical relevance. The paper addresses a critical and under-explored domain: satellite orbit prediction using large time-series models. This setting highlights real-world importance and contributes to the intersection of space situational awareness and AI for scientific computing.
+ Effective and task-aligned efficiency. The method thoughtfully exploits the characteristics of orbital time-series data, using real-valued frequency transforms and tensorized attention to substantially reduce model parameters while preserving both predictive accuracy and inference efficiency.
+ Extensive and well-validated experiments. The paper conducts comprehensive experiments across multiple satellite constellations and real-world datasets, including detailed zero-shot evaluations and ablation studies.  The breadth of baselines and the consistency of improvements provide strong empirical support for the proposed approach

**Weaknesses:**

+ Limited theoretical insight. While the paper presents clear empirical gains, it lacks a deeper theoretical analysis explaining why tensor decomposition and real-valued Fourier transforms improve performance.  The benefits appear primarily empirical, without discussion on the induced inductive biases or representational properties.

+ Incremental methodological novelty. The proposed components, such as RFFT substitution, LayerNorm removal, and tensorized attention, are all established techniques.  Their combination is well-engineered but not conceptually groundbreaking, making the contribution more engineering-oriented than methodological.

+ Lack of comparison with traditional baselines. The introduction motivates the work by contrasting data-driven and physics-based orbit prediction methods. However, the experiments only compare against neural and transformer-based baselines, without including classical orbital propagators or numerical integration schemes (e.g., SGP4). This weakens the empirical link between the stated motivation and the reported results.

**Questions:**

+ How sensitive is the model’s performance to the choice of Tensor-Train rank? Have you observed any trade-off between compression and accuracy?
+ The paper demonstrates strong cross-constellation performance, suggesting promising generalization. However, has the proposed model been validated on other time-series datasets beyond orbital dynamics, or is its performance limited to this specific domain?
+ The paper argues that physics-based propagators such as Orekit are slow and inaccurate, yet these same methods are used to generate the interpolated training and ground-truth data. This raises a conceptual concern:
    - If Orekit is inaccurate, wouldn’t the model inherit its systematic bias through the labels?
    - And if the model still requires interpolated inputs from Orekit during inference, does this not reintroduce the very computational bottleneck the method aims to eliminate?
    - Could the authors clarify how the interpolation process affects both training and inference, and whether the proposed model can operate independently of the physical propagators?

---

> ### Author Response · Authors · 2025-11-28
>
> ## **W1 & W2: Theoretical Depth and Methodological Novelty**
> Thank you to the reviewer for the constructive comments and helpful feedback. We agree with your assessment regarding the theoretical depth and the nature of our methodological contribution. However, through our proposed approach, we demonstrated:
> * Enhanced capability to capture the inherent periodicity of satellite orbits (via RFFT, **in new Section 4.5**).
> * Significant model compression with minimal performance degradation (via Tensor Decomposition).
>
> These results are critical for achieving practical and stable orbit prediction in real-world scenarios. Furthermore, recent studies in AI-based orbit prediction [1-6] have similarly adopted this approach, adapting established deep learning techniques to address domain-specific challenges. Our work aligns with this research direction, focusing on the engineering synthesis required for this specific application. We sincerely thank you for this valuable insight.
> ***
> REFERENCES
> [1] Salleh, N., et al.: An adaptation of deep learning technique in orbit propagation
> model using long short-term memory. In: ICECCE. pp. 1–6. IEEE (2021)
> [2] Curzi, G., et al.: Two-line-element propagation improvement and uncertainty es-
> timation using recurrent neural networks. CEAS Space Journal 14(1), 197–204
> (2022)
> [3] Chen, X., et al.: Modelling and prediction of atmospheric drag coefficients in leo
> satellite orbit determination and prediction with bi-lstm approach. Advances in
> Space Research 75(3), 2874–2888 (2025)
> [4] Ren, H., et al.: Research on satellite orbit prediction based on neural network
> algorithm. In: HPCCT. pp. 267–273 (2019)
> [5] Osama, A., et al.: Satellite orbit prediction based on recurrent neural network using
> two line elements. In: ICCI. pp. 298–302. IEEE (2022)
> [6] Shin, Y., et al.: Selective tensorized multi-layer lstm for orbit prediction. In: CIKM.
> pp. 3495–3504 (2022)
> ***
> ## **W3. Comparison with Physics-Based Baselines**
> Thank you for raising this important point regarding the comparison with traditional orbital propagators. We intentionally excluded traditional propagators (SGP4, Skyfield, Orekit) from the main comparison tables due to a fundamental methodological inconsistency. Classical propagators require a single instantaneous epoch, specifically a Two-Line Element (TLE), to propagate orbits forward. In contrast, our deep learning architecture relies on a sliding look-back window of historical observations to capture temporal dependencies, making direct comparison challenging within the same experimental setup.
>
> However, to address this concern and demonstrate DAF's advantages over classical methods, we conducted an additional evaluation by adapting the experimental setup. We converted the position and velocity vectors used as inputs for our model back into TLE elements, which were then fed into the physics-based propagators. The evaluation was performed on the same 56 test satellites used for benchmarking the neural baselines. The results are as follows:
>
> | Method | MSE (Norm) | MAE (Norm) | MSE (Denorm) | MAE (Denorm) |
> |--------|------------|------------|--------------|--------------|
> | SGP4 | 1.94 |1.12 |15011455.90 |2208.77 |
> | Skyfield | 1.92 | 1.11 | 14726191.26 | 2176.72 |
> | Orekit | 1.95 | 1.13 | 15115396.31 | 2217.08 |
>
> These results demonstrate that classical propagators significantly underperform compared to our deep learning approach (**Table 3**), validating the motivation for data-driven methods and confirming DAF's advantages in terms of both normalized metrics and reduced physical position errors.

---

> > ### Author Response · Authors · 2025-11-28
> >
> > ## **Q1. Sensitivity Analysis of Tensor-Train Rank**
> > Thank you for this important question. As specified in **Appendix D**, DAF uses a fixed TT-Rank of [1, 3, 32, 1]. We have conducted experiments with different TT-Rank configurations to investigate the trade-off between compression and accuracy.
> >
> > **The results confirm a trade-off between compression and accuracy:** lower ranks yield fewer parameters but degrade performance. Specifically, reducing the rank from 32 to 8 decreases parameters by 11.0% but increases MSE by 8.5%. We selected [1, 3, 32, 1] as it achieves the best accuracy while still providing substantial compression. We did not experiment with higher ranks (e.g., [1, 3, 64, 1]) because the rank would exceed the original matrix dimensions, resulting in more parameters than the uncompressed FredFormer (Layer Variant) and thus defeating the purpose of tensor decomposition.
> >
> > | TT-Rank | Parameters | MSE | MAE |
> > |---------|------------|-----|-----|
> > | [1, 3, 8, 1] | 199,980 | 0.001064 | 0.024991 |
> > | [1, 3, 16, 1] | 208,172 | 0.001030 | 0.024833 |
> > | [1, 3, 32, 1] | 224,556 | **0.000981** | **0.023853** |
> >
> > ***
> > ## **Q2. Generalization to General Time-Series Domains**
> > Thank you for this insightful suggestion. While the current study focuses on orbital dynamics, we anticipate that the proposed model has strong potential for broader application due to its frequency-domain architecture. Specifically, we expect that DAF's advantages in handling periodicity and long-range dependencies will extend effectively to other quasi-periodic domains such as meteorology, energy consumption, and signal processing. Incorporating your feedback, we have updated **Section 5 (Lines 493-497)** to explicitly discuss our future plan to apply the proposed framework to these general time-series tasks where capturing frequency-domain features is critical.

---

> > > ### Author Response · Authors · 2025-11-28
> > >
> > > ## **Q3. Propagator Accuracy, Bias, and Computational Independence**
> > > We appreciate the reviewer's conceptual concern regarding the reliance on physics-based propagators. We would like to clarify our stance on the accuracy of these propagators and the computational efficiency of our proposed framework during inference.
> > >
> > > **1. Clarification on Accuracy and Bias**
> > > First, we would like to clarify a misunderstanding regarding the premise that Orekit is inaccurate. In our manuscript (**Section 1**), the mention of significant inaccuracy ("*kilometer-level prediction error*") referred specifically to SGP4, not Orekit.
> > > As detailed in **Section 2.2** and supported by **Table 2 and Table 10**, we validated SGP4, Skyfield, and Orekit against high-precision ephemeris data (ground truth). Our experiments demonstrated that Orekit significantly outperforms SGP4 (**Table 10**) and provides the lowest error rates. Consequently, we selected Orekit as the most reliable physics-based model available to interpolate irregular TLE data into high-precision, 1-minute interval training labels.
> > > Therefore, the model learns from the highest-fidelity physics-based data available in the absence of dense ground truth, rather than learning from inaccurate labels. Furthermore, to prove that our model is not dependent on specific propagator biases, we also conducted experiments using the KOMPSAT dataset. Since this dataset consists of real-world telemetry already provided at 1-minute intervals, it does not require propagator-based interpolation (**Table 5**). The DAF model demonstrated superior performance on this dataset as well, confirming its robustness.
> > >
> > > **2. Computational Bottleneck and Independence (Training vs. Inference)**
> > > Regarding the concern about computational bottlenecks and the reintroduction of propagator latency, we would like to distinguish clearly between the training and inference phases:
> > >
> > > **Training Phase (Offline):** Orekit is used exclusively for offline dataset construction (**Figure 1(a)**). Due to security and operational constraints, publicly available datasets often feature large sampling intervals (e.g., exceeding 8 hours, as shown in **Table 1**). To overcome this data sparsity and train a high-precision model, we used Orekit to interpolate these irregular TLEs into a dense, physically consistent dataset at 1-minute intervals. This is a one-time offline cost incurred only before training.
> > >
> > > **Inference Phase (Online & Independent):** Crucially, the proposed model operates completely independently of physical propagators during inference, thus avoiding the reintroduction of any computational bottleneck. In real-world operations, the data sparsity observed in public datasets is primarily a result of security policies rather than a lack of measurements. While high-resolution data is restricted from the public domain, satellite operators possess high-frequency internal telemetry (e.g., 1-minute intervals) for their assets. Consequently, operators feed this dense internal data directly into the model, eliminating the need for propagator-based interpolation. This is evidenced by the KOMPSAT dataset specifications in **Table 1**, which serves to demonstrate that real-world telemetry is indeed collected at 1-minute intervals, thereby confirming that interpolation is unnecessary in operational settings. Even when starting from TLEs, only a lightweight vector conversion is needed, avoiding full orbital propagation.
> > >
> > > In summary, the DAF model uses Orekit only to generate high-quality data for offline training. During online inference, the model operates completely independently of any physical propagator. Thus, the computational bottleneck associated with propagators is not reintroduced, ensuring the model remains lightweight and suitable for real-time operations.

---

### Official Review · Reviewer_Q1tf · 2025-10-31

**Soundness:** 3
**Presentation:** 3
**Contribution:** 4
**Rating:** 8
**Confidence:** 3

**Summary:**

The authors propose a large time-series model for satellite orbit prediction based on FredFomer. The authors introduce several modifications to the architecture and demonstrate significant improvements in the performance on a large variety of datasets.

**Strengths:**

S1: The problem is increasingly important to solve

S2: The authors performed several thoughtful (RFFT and TTD, plus several tweaks (e.g. removing patching and layer normalization)) changes to the FredFormer architecture in order to attain higher performance. These changes reflect the domain-specific knowledge about the available data

S3: The resulting model is lightweight, which is crucial for keeping the cost of these operations low.

**Weaknesses:**

W1: About the model being lightweight, the authors simply report the parameter count. For this application speed is also crucial, it would be very nice if the authors could also report throughput/FLOPs or any other metrics that would give an idea of the speed at which predictions can be made. This should be reported in comparison with all the baselines.

**Questions:**

Q1: Given in other domains there's a substantial trend of augmenting the real-world training datasets with synthetic data, did the authors consider augmenting the Starlink dataset with synthetic data? (e.g. from Kessler [1]). In particularly it could be interesting to know:

Q1.1. In case the failures of the current model can be identified as gaps in the training distributions, can the addition of synthetic data alleviate the issue?

Q1.2. Is it possible to mix synthetic and real data in order to obtain better performance? Or is it detrimental?

Q2: The authors mention performing normalization and its impact on the reported errors (e.g. a 0.001 error can translate to tens of kilometers).

Q2.1. Different perdicted physical variables have different sensitivity, would it be possible for the authors report a breakdown of the MSE/MAE for each predicted variable, to identify whether the errors are evenly distributed or if there's some variables that are harder to predict and could be an area for further improvement for future models?

Q2.2. I didn't get the chance to check the code, but is the normalization performed per-dataset or across all datasets? Is it possible that a global normalization may render some data less useful in training? Or alternatively, is it possible that a per-dataset normalization may amplify errors on some datasets?

Q3: have the authors considered using techniques that could produce not only predictions but also error estimates (e.g. in terms of variance around the predicted value)?

[1] https://github.com/kesslerlib/kessler

---

> ### Author Response · Authors · 2025-11-28
>
> ## **W1: Comprehensive Computational Efficiency (Throughput/FLOPs)**
> We appreciate the reviewer's suggestion to report throughput or FLOPs to better assess the model's speed.
>
> **1. Clarification on Inference Time (Existing in Table 9)**
> Regarding the wall-clock speed, we would like to highlight that the inference time was already reported in **Table 9** of the original manuscript. These measurements represent the total inference time for the Starlink dataset and 10 cross-domain datasets on an NVIDIA GeForce RTX 3090 GPU.
> As shown in **Table 9**, DAF (3.86s for Starlink) matches the speed of traditional lightweight models like GRU (3.86s) and LSTM (3.84s), while being orders of magnitude faster than large foundation models like TimesFM (756.06s). This existing metric demonstrates the practical, lightweight nature of our model in a real-world deployment scenario.
>
> **2. Additional Metric: *GFLOPs* per Sequence**
> However, we agree that providing *FLOPs* is essential to evaluate the architectural efficiency independently of hardware. Therefore, as requested, we have newly calculated the *GFLOPs per sequence* for all baselines. To ensure a fair comparison, the batch size was fixed at 16. For univariate foundation models (TimesFM and Time-MoE), the *GFLOPs* were scaled by the number of input features.
>
> These results have been incorporated into the **Table 9** of the revised manuscript. The summary is as follows:
> | Model | GFLOPs per sequence |
> | :--- | :---: |
> | **Time-MoE** | 5353.2000 |
> | **Timer** | 25.1552 |
> | **Timer-XL** | 25.1552 |
> | **Time-LLM** | 954.4704 |
> | **TTM** | 62.1760 |
> | **TimesFM** | 517.2000 |
> | **MOMENT** | 7.0992 |
> | **MOIRAI** | 7.1008 |
> | **GRU** | 0.4368 |
> | **LSTM** | 0.5824 |
> | **FredFormer** | 0.0240 |
> | **DLinear** | 0.0096 |
> | **SparseTSF** | 0.0016 |
> | **DAF (Ours)** | 0.0144 |
>
> ***
> ## **Q1: Synthetic Data Augmentation**
> We appreciate the reviewer's suggestion to explore synthetic data augmentation (e.g., using Kessler [1]). We agree that this is a promising direction, and we address the specific sub-questions as follows:
>
> ### **Q1.1. - In case the failures of the current model can be identified as gaps in the training distributions, can the addition of synthetic data alleviate the issue?**
> Yes, we strongly believe that adding synthetic data can effectively alleviate these issues. The primary failure modes of our current model, such as prediction errors during sudden maneuvers (e.g., collision avoidance) or extreme space weather events, stem from the fact that these anomalies are extremely rare or absent in the Starlink training dataset [2]. This creates clear gaps in the training distribution.
> By utilizing high-fidelity simulators (like Kessler) to generate synthetic data specifically for these rare scenarios, we can expose the model to these out-of-distribution patterns during training. This would enable the model to learn latent representations of anomalous dynamics, thereby significantly enhancing its robustness against edge cases that rarely occur in nominal operations.
>
> ### **Q1.2. - Is it possible to mix synthetic and real data in order to obtain better performance? Or is it detrimental?**
> We consider the mixing of synthetic and real data to be highly beneficial rather than detrimental, provided it is done correctly.
> * **Real data (Starlink)** serves as the baseline. It allows the model to learn the nominal orbital dynamics and standard perturbations (e.g., atmospheric drag, solar radiation pressure) grounded in physical reality.
> * **Synthetic data** acts as an effective data augmentation strategy to fill the gaps for edge cases or stress test scenarios that real data lacks.
>
> However, a critical caveat is the mixing methodology. Merely injecting synthetic data points randomly or treating them as simple noise could be detrimental. The integration must be physics-based and context-aware. Synthetic segments must respect the temporal continuity and orbital mechanics of the orbit. Instead of random injection, synthetic maneuvers or anomalies should be seamlessly blended into the time-series to preserve the physical consistency of the orbit [3]. Under these conditions, we expect the hybrid dataset to significantly improve both the model's generalization and robustness.
>
> ***
> REFERENCES
> [1] https://github.com/kesslerlib/kessler
> [2] Baireddy, Sriram, et al. "Spacecraft time-series online anomaly detection using deep learning." 2023 IEEE Aerospace Conference. IEEE, 2023.
> [3] Baudier, Stéfan, et al. "Synthetic Dataset of Maneuvering Low Earth Orbit Satellite Trajectories for AI Analysis." Proceedings of SPAICE2024: The First Joint European Space Agency/IAA Conference on AI in and for Space. Zenodo, 2024.

---

> > ### Author Response · Authors · 2025-11-28
> >
> > ## **Q2. Normalization Strategy and Detailed Error Analysis**
> > We appreciate the reviewer's detailed inquiry regarding the breakdown of errors per variable and the normalization strategy. We address each sub-question below.
> >
> > ### **Q2.1. - Report a breakdown of the MSE/MAE for each predicted variable to identify whether the errors are evenly distributed. Are there some variables that are harder to predict and could be an area for further improvement for future models?**
> > Thanks for the great suggestion! We agree that the normalized metrics averaged across all variables might obscure the specific behaviors of individual physical components. To address this, we have computed the *de-normalized MSE, MAE, and RMSE* for each of the 6 state variables ($x, y, z, vx, vy, vz$). These results represent the actual *physical errors in kilometers (km)* for position and *kilometers per second (km/s)* for velocity. The breakdown for the baselines and our proposed model (DAF) is as follows:
> >
> > **(1) TimesFM**
> > | Variable | MSE (Pos: $km^2$, Vel: $km^2/s^2$) | MAE (Pos: $km$, Vel: $km/s$) | RMSE (Pos: $km$, Vel: $km/s$) |
> > | :---: | :---: | :---: | :---: |
> > | **x** | 18,119.7764 | 94.9481 | 133.9801 |
> > | **y** | 16,792.0257 | 91.4305 | 126.5273 |
> > | **z** | 23,878.3529 | 116.4781 | 154.4132 |
> > | **vx** | 0.0300 | 0.1282 | 0.1713 |
> > | **vy** | 0.0455 | 0.1584 | 0.2104 |
> > | **vz** | 0.0271 | 0.1272 | 0.1644 |
> >
> > **(2) GRU**
> > | Variable | MSE (Pos: $km^2$, Vel: $km^2/s^2$) | MAE (Pos: $km$, Vel: $km/s$) | RMSE (Pos: $km$, Vel: $km/s$) |
> > | :---: | :---: | :---: | :---: |
> > | **x** | 43,118.8100 | 152.4500 | 207.6400 |
> > | **y** | 31,898.5900 | 128.6500 | 178.1300 |
> > | **z** | 32,304.1300 | 129.6400 | 179.6300 |
> > | **vx** | 0.0482 | 0.1611 | 0.2194 |
> > | **vy** | 0.0369 | 0.1391 | 0.1915 |
> > | **vz** | 0.0367 | 0.1381 | 0.1915 |
> >
> > **(3) DLinear**
> > | Variable | MSE (Pos: $km^2$, Vel: $km^2/s^2$) | MAE (Pos: $km$, Vel: $km/s$) | RMSE (Pos: $km$, Vel: $km/s$) |
> > | :---: | :---: | :---: | :---: |
> > | **x** | 32,935.0300 | 134.8300 | 181.4700 |
> > | **y** | 40,525.6800 | 144.6300 | 201.3100 |
> > | **z** | 43,672.2100 | 155.3300 | 208.9700 |
> > | **vx** | 0.0431 | 0.1574 | 0.2077 |
> > | **vy** | 0.0534 | 0.1707 | 0.2311 |
> > | **vz** | 0.0511 | 0.1621 | 0.2261 |
> >
> > **(4) SparseTSF**
> > | Variable | MSE (Pos: $km^2$, Vel: $km^2/s^2$) | MAE (Pos: $km$, Vel: $km/s$) | RMSE (Pos: $km$, Vel: $km/s$) |
> > | :---: | :---: | :---: | :---: |
> > | **x** | 67,789.4200 | 212.9700 | 260.3600 |
> > | **y** | 67,521.6300 | 212.4100 | 259.8500 |
> > | **z** | 63,398.1900 | 205.6100 | 251.7900 |
> > | **vx** | 0.0816 | 0.2333 | 0.2856 |
> > | **vy** | 0.0811 | 0.2327 | 0.2848 |
> > | **vz** | 0.0761 | 0.2261 | 0.2758 |
> >
> > **(5) DAF (ours)**
> > | Variable | MSE (Pos: $km^2$, Vel: $km^2/s^2$) | MAE (Pos: $km$, Vel: $km/s$) | RMSE (Pos: $km$, Vel: $km/s$) |
> > | :---: | :---: | :---: | :---: |
> > | **x** | 15,860.0300 | 95.7300 | 125.9200 |
> > | **y** | 15,866.1500 | 95.4500 | 125.9500 |
> > | **z** | 15,148.6500 | 94.3800 | 123.0600 |
> > | **vx** | 0.0190 | 0.1047 | 0.1379 |
> > | **vy** | 0.0188 | 0.1044 | 0.1373 |
> > | **vz** | 0.0186 | 0.1042 | 0.1362 |
> >
> > **Uniformity in Proposed Model:** As demonstrated in the tables, DAF exhibits a remarkably uniform error distribution across all spatial dimensions ($x, y, z$). The RMSE values for position are nearly identical (ranging from 123.06 to 125.95 km), and velocity errors are similarly balanced (approx. 0.136–0.137 km/s). This indicates that our model effectively captures the coupled 3D orbital dynamics without biasing towards any specific planar projection.
> >
> > **Comparison with Baselines:** In contrast, several baselines show uneven distributions. For instance, TimesFM and DLinear exhibit noticeably higher errors in the $z$-axis compared to the $x$ or $y$ axes (e.g., TimesFM $z$-RMSE is ~22% higher than $y$-RMSE). This suggests that existing models may struggle with specific orbital components, whereas DAF maintains robustness across all variables.
> >
> > **Future Improvement:** Since there is no single "hard-to-predict" variable in our model, future improvements should focus on reducing the overall magnitude of errors across all dimensions rather than targeting specific component-wise deficiencies.

---

> > > ### Author Response · Authors · 2025-11-28
> > >
> > > ### **Q2.2. - The normalization performed per-dataset or across all datasets? Is it possible that a global normalization may render some data less useful in training? Or is it possible that a per-dataset normalization may amplify errors on some datasets?**
> > > We employed a *global normalization strategy*. As explicitly stated in **Section 2.2** of the manuscript, we calculated the mean and standard deviation solely from the pre-training split (Starlink) and applied these fixed statistics to normalize the training, validation, and test sets. Crucially, these same statistics were applied to all 10 zero-shot datasets (**Tables 4 and 5**).
> > > We address your specific questions regarding the potential risks of normalization strategies as follows:
> > >
> > > **1. Is it possible that global normalization renders some data less useful?**
> > > We acknowledge the reviewer's valid point that *global normalization* runs the risk of rendering some out-of-distribution data less useful due to distributional shifts. However, the strong zero-shot performance demonstrated in **Tables 4 and 5** suggests that DAF is highly robust to such distributional shifts. It effectively learns the underlying orbital dynamics even when the input data is normalized by the source domain's (Starlink) statistics.
> > >
> > > **2. Is it possible that a per-dataset normalization may amplify errors?**
> > > Yes, we believe per-dataset normalization can introduce significant issues in this context:
> > > * **Loss of Physical Context:** Per-dataset normalization forces distinct orbital regimes into the same standard normal distribution ($N(0,1)$). This removes the absolute scale information that helps the model distinguish between orbital regimes, potentially amplifying errors when the model attempts to reconstruct the physical orbit.
> > > * **Realistic Zero-shot Setting:** More critically, using per-dataset normalization implies prior knowledge of the target dataset's statistics (mean/std). In a realistic zero-shot setting (e.g., deploying to a new satellite like KOMPSAT), these statistics are unknown at the time of deployment. Using them during testing would violate the strict zero-shot assumption inherent to real-world operations.
> > >
> > > By sticking to *global normalization*, we ensure a rigorous evaluation of the model's true generalization capability without relying on target-domain statistics.
> > > ***
> > > ## **Q3. Error Estimation and Variance**
> > > We appreciate the reviewer's insightful suggestion regarding the provision of error estimates (e.g., variance). We agree that for real-world satellite operations, such as collision avoidance, estimating the variance or confidence intervals around a prediction is as critical as the deterministic value itself.
> > > Recognizing this as a crucial direction, we have revised **Section 5 (Discussion and Future Work)** to explicitly address the development of methods for estimating variance and reliability. The added text is as follows **(Lines 489-493)**:
> > > *"To address this, we will explore methods to transition from deterministic to probabilistic forecasting, such as modifying the output layer to estimate Gaussian distribution parameters (mean and variance) using negative log-likelihood loss or employing quantile loss to provide confidence intervals. Finally, we will validate the model's practical applicability by conducting experiments on a broader set of high‑precision, real‑world satellite orbit datasets."*

---

### Official Review · Reviewer_ixVa · 2025-11-01

**Soundness:** 3
**Presentation:** 3
**Contribution:** 3
**Rating:** 4
**Confidence:** 3

**Summary:**

The paper tackles satellite orbit forecasting and proposes DAF (Decomposed Attention in the Frequency domain): raw time-series are RFFT-tokenized (real/imag concatenated), fed to an attention block factorized via a low-rank/tensor-train style decomposition, and trained without patching (and with global z-scoring). Across multiple orbital datasets, DAF outperforms strong transformer and large-FM baselines while being inference-efficient on a single GPU. The method’s design is motivated by orbital dynamics’ quasi-periodicity and long-range dependencies.

**Strengths:**

-  Frequency-domain tokenization is a natural fit for quasi-periodic orbital signals, helping the model capture long-range structure with fewer tokens.
- The decomposed attention reduces memory/time complexity versus dense attention at long contexts, enabling practical inference on commodity GPUs.
- DAF consistently beats competitive transformer baselines and several large time-series FMs on orbital forecasting benchmarks.
- Clear experimental protocol. Common training settings across models and transparent dataset preparation (interpolation choice justified and tested) improve reproducibility and interpretability of results.

**Weaknesses:**

- The paper reports parameter counts but lacks training-time wall-clock (hours), or FLOPs per step or per sequence. Without these, it’s hard to assess whether gains come from architectural merit vs. compute budget or implementation. Please unify timing/compute reporting across all baselines.
- Frequency-domain positional embeddings need justification. Inputs are RFFT’d and concatenated, then positional embeddings are added before attention. The physical meaning of these PEs in the frequency domain is unclear. Provide ablations vs. time-domain PEs (or none), and discuss why frequency-space tokens need positional encoding and what “position” represents there.

**Questions:**

- Many baselines/FMs that the paper compared to are fairly old. There are many recent time series foundation models that provide a much stronger performance. Consider inclusing them in results such as ([Chronos](https://arxiv.org/abs/2403.07815)) and ([TOTO](https://arxiv.org/abs/2505.14766))?
- You validate interpolation with SGP4/Skyfield/Orekit. Can you also include a propagation-based physics baseline (same splits) in the forecasting tables to contextualize ML gains?
- Orbit specific metrics: Since small MAE/MSE differences can correspond to kilometer-scale position errors, is there a better metric that you can use to evaluate such as 3D position error in km ?
- Given DAF’s strong results on satellite orbit prediction, what assumptions or inductive biases make it particularly suitable to orbital dynamics (e.g. spectral sparsity, long-range dependencies)? Do you expect the same advantages to hold for general time series, and under what conditions?

---

> ### Author Response · Authors · 2025-11-28
>
> ## **W1: Comprehensive Computational Efficiency Reporting (GFLOPs)**
> We appreciate the reviewer's valuable feedback regarding the need for comprehensive computational reporting. As noted, our initial manuscript focused primarily on inference latency (**Table 9**) and parameter counts (**Tables 3 and 7**) to highlight the deployability of the model on resource-constrained onboard systems (as discussed in **Section 5**).
>
> However, we agree that providing FLOPs is essential to assess whether performance gains stem from architectural merit or compute resources. Accordingly, we have calculated the GFLOPs per sequence for all baselines. To ensure a fair comparison, the batch size was fixed at 16 for all experiments. For TimesFM and Time-MoE, the GFLOPs were simply scaled by the number of input features to align with the multivariate setting of the other baselines. These results have been incorporated into the revised **Table 9 (Lines 432-444)**. As shown below, DAF (Ours) demonstrates exceptional computational efficiency compared to large-scale foundation models and remains competitive with lightweight baselines:
> | Model | GFLOPs per sequence |
> | :--- | :---: |
> | **Time-MoE** | 5353.2000 |
> | **Timer** | 25.1553 |
> | **Timer-XL** | 25.1557 |
> | **Time-LLM** | 954.4704 |
> | **TTM** | 62.1760 |
> | **TimesFM** | 517.2000 |
> | **MOMENT** | 7.0992 |
> | **MOIRAI** | 7.1008 |
> | **GRU** | 0.4368 |
> | **LSTM** | 0.5824 |
> | **FredFormer** | 0.0240 |
> | **DLinear** | 0.0096 |
> | **SparseTSF** | 0.0016 |
> | **DAF (Ours)** | 0.0144 |
> ***
> ## **W2: Justification and Ablation of Frequency-Domain Positional Embeddings**
> Thank you for raising this important point. We clarify that since we removed the patching mechanism from the original FredFormer, we introduced positional embedding (PE) as a lightweight replacement to preserve sequential structure, rather than to encode physical frequency-bin relationships.
>
> The original FredFormer uses patching to capture multi-scale temporal patterns. However, satellite orbital data exhibit strong periodicity with a relatively simple structure, reducing the need for local pattern learning. We therefore removed patching and adopted PE to preserve the sequential structure of frequency components with minimal overhead. The results show that *DAF (ours)* reduces MSE by 3.1% compared to *DAF (no PE)* while using 3,328 fewer parameters, demonstrating that PE is a more efficient alternative to patching. Furthermore, *DAF (ours)* outperforms *DAF (no RFFT)* by 23.5%, confirming that RFFT is effective for accurate orbit prediction. The slight parameter increase (256 params) in *DAF (ours)* over *DAF (no RFFT)* is due to RFFT producing $N+2$ dimensions ($N/2+1$ complex numbers represented as concatenated real and imaginary parts) compared to $N$ dimensions in the time domain. We have added these experiments to **Section 4.5 (Lines 451-468)** of the revised manuscript.
>
> | Model Variant                           | MSE         | MAE         | Parameter |
> |-----------------------------------------|-------------|-------------|------------|
> | DAF-timePE (no RFFT)    | 0.001281  | 0.027065  | 224,300 |
> | DAF-noPE (no PE)         | 0.001012  | 0.024515  | 227,884 |
> | DAF (ours)          | 0.000981    | 0.023853    | 224,556 |

---

> > ### Author Response · Authors · 2025-11-28
> >
> > ## **Q1. Comparison with Recent Foundation Models (Chronos 2 & TOTO)**
> >
> > We appreciate the reviewer's comment regarding the selection of baseline models. We respectfully wish to clarify the recency of the baselines selected for this study. The impression that the baselines are fairly old may stem from the inclusion of LSTM, GRU, and DLinear. These were included solely to serve as standard historical benchmarks, which is a common practice in time-series forecasting research.
> >
> > In contrast, the TSFMs we employed as primary baselines represent the SOTA, with most published in *2024 or later*:
> > * **2024 Publications:** SparseTSF, FredFormer, TimesFM, MOIRAI, TTM, MOMENT, and Timer.
> > * **2025 Publications:** Timer-XL and Time-MoE were presented at *ICLR 2025*, with Time-MoE notably selected as a *Spotlight paper*.
> >
> > Given that the ICLR 2026 submission deadline was in *September 2025*, our selection of baselines strictly adhered to the most recent SOTA standards available at the time of writing, ensuring the fairness and currency of our comparative analysis. Regarding the specific models you suggested:
> > * **Chronos:** While accepted to *TMLR in 2024* [1], it was initially prioritized lower than the *ICLR 2025* models.
> > * **TOTO:** The initial version appeared on *arXiv in July 2024* [2], and the updated version was released in *May 2025* [3].
> >
> > **Additional Experiments on TOTO and Chronos 2**
> > We agree with your suggestion to compare against the very latest architectures to validate the competitiveness of our proposed method. We have conducted additional experiments using **TOTO** [3] and **Chronos 2** (released *October 2025* [4]).
> >
> > The experiments followed the identical setup described in **Section 4.1**, including fine-tuning on the Starlink dataset.
> > **Performance Comparison with TOTO and Chronos 2**
> > | Metric | TOTO [3] | Chronos 2 [4] |
> > | :--- | :---: | :---: |
> > | **Normalized MSE** | 0.4501 | 0.1007 |
> > | **Normalized MAE** | 0.5794 | 0.0806 |
> > | **FLOPs (G)** | 119.79 | 126.11 |
> > | **3D Pos. RMS (km)** | 4,711.60 | 2,164.10 |
> > | **3D Pos. Mean (km)** | 4,585.34 | 654.23 |
> > | **3D Pos. Median (km)** | 4,682.61 | 79.28 |
> > | **3D Pos. Std (km)** | 1,083.45 | 2,062.84 |
> > | **x-MSE ($km^2$)** | $7.38 \times 10^6$ | $1.82 \times 10^6$ |
> > | **y-MSE ($km^2$)** | $7.66 \times 10^6$ | $1.56 \times 10^6$ |
> > | **z-MSE ($km^2$)** | $7.16 \times 10^6$ | $1.31 \times 10^6$ |
> >
> > The results indicate that while **Chronos 2** outperforms **TOTO**, our proposed model (**DAF**) demonstrates superior performance, achieving significantly lower errors *(MSE 0.0009 vs. 0.1007)* with orders of magnitude greater computational efficiency.
> > ***
> > REFERENCES
> > [1] Ansari, Abdul Fatir, et al. "Chronos: Learning the Language of Time Series." Transactions on Machine Learning Research.
> > [2] Cohen, Ben, et al. "Toto: Time series optimized transformer for observability." arXiv preprint arXiv:2407.07874 (2024).
> > [3] Cohen, Ben, et al. "This Time is Different: An Observability Perspective on Time Series Foundation Models." arXiv preprint arXiv:2505.14766 (2025).
> > [4] Ansari, Abdul Fatir, et al. "Chronos-2: From univariate to universal forecasting." arXiv preprint arXiv:2510.15821 (2025).

---

> > > ### Author Response · Authors · 2025-11-28
> > >
> > > ## **Q2. Benchmarking Against Physics-Based Propagators**
> > > Thank you for your suggestion regarding the inclusion of physics-based propagators (SGP4, Skyfield, Orekit) as baselines. While such benchmarks are essential for contextualizing our model's performance, we intentionally excluded physics-based propagators from the main forecasting tables due to a fundamental inconsistency in experimental setup and input requirements. Standard propagators require a single, instantaneous input epoch, specifically a corresponding Two-Line Element (TLE), to project the orbit forward. This methodology contrasts sharply with our deep learning architecture, which requires the time-series context provided by a sliding look-back window of historical observations.
> > >
> > > However, to provide a contextually relevant comparison and demonstrate DAF's advantages beyond classical models, we adapted the experimental setup specifically for this comparison. We converted the position and velocity vectors used as inputs to our deep learning model back into TLEs, which were then used as inputs for the physics-based propagators. The evaluation was conducted on the same 56 satellites used for testing the deep learning models.
> > > | Propagator | MSE (norm) | MAE (norm) | MSE (de-norm) | MAE (de-norm) |
> > > | :--- | :---: | :---: | :---: | :---: |
> > > | **SGP4** | 1.94 | 1.12 | 15011455.90 | 2208.77 |
> > > | **Skyfield** | 1.92 | 1.11 | 14726191.26 | 2176.72 |
> > > | **Orekit** | 1.95 | 1.13| 15115396.31 | 2217.08 |
> > >
> > > The results of this adapted comparison confirm the substantial gains achieved by our proposed architecture DAF (**Table 3**), which learns corrections and patterns beyond classical physics models.
> > > ***
> > > ## **Q3. Physical Interpretation of Errors (3D Position Metrics)**
> > > We appreciate the reviewer's valuable insight regarding the physical interpretation of normalized metrics. While our main analysis relies on standard MSE/MAE benchmarks, we acknowledge that checking *3D position errors* provides useful context for real-world applicability. To address this, we calculated the *de-normalized 3D position errors* using the identical experimental settings as the Main Results. We report the *RMS, Mean, Median, and Standard Deviation (Std) of the position errors* in kilometers. The results are summarized in the table below:
> > > | Model | RMS (km) | Mean (km) | Median (km) | Std (km) |
> > > | :--- | :---: | :---: | :---: | :---: |
> > > | **TimesFM** | 241.41 | 205.44 | 189.22 | 126.45 |
> > > | **GRU** | 327.41 | 268.19 | 231.99 | 187.51 |
> > > | **DLinear** | 342.24 | 283.90 | 239.06 | 191.12 |
> > > | **SparseTSF** | 445.77 | 410.50 | 434.94 | 173.77 |
> > > | **DAF (Ours)** | **216.48** | **186.41** | **172.53** | **109.98** |
> > >
> > > As demonstrated, our proposed model (DAF) achieves the lowest error rates across all metrics.
> > > ***
> > > ## **Q4. Inductive Biases and Generalization to Other Domains**
> > > We gratefully acknowledge the reviewer's constructive feedback and insightful questions. DAF's suitability for orbital dynamics have  the following *assumptions*:
> > > **1. Emphasis on Periodicity (RFFT Backbone):** DAF uses the RFFT/IRFFT (Real Fast Fourier Transform) as its backbone (**Section 3.2**). This provides a strong bias towards processing time-series data in the frequency-domain rather than the time domain. Satellite orbits inherently possess very strong quasi-periodicity (though slightly varied by perturbations). DAF efficiently captures and models this periodic characteristic by transforming the data into its frequency spectrum.
> > > **2. Global Pattern and Long-range Dependencies through Transformer:** We hypothesize (**Section 3.3**) that because satellite orbit data exhibits strong periodicity and a relatively simple structure, the importance of learning local patterns is reduced. We therefore removed the patching mechanism used in the original FredFormer. Instead, we use the Transformer attention (TMHA) to focus on capturing Long-range Dependencies and global relationships among frequency components across the entire look-back window. This approach is well-suited for understanding the dynamics of the orbit as a whole.
> > > **3. Assumption of Efficient Representation (TTD):** Applying Tensor Train Decomposition (TTD) (**Section 3.4**) to the attention weights compresses the model. This embeds an inductive bias that the complex interactions between frequency components can actually be efficiently represented by a lower-rank structure.
> > >
> > > We expect that DAF's advantages (data with periodicity and long-range dependencies) will hold true and work well for other general time-series data that exhibit strong periodicity or quasi-periodicity. Examples include seasonal weather data, daily/weekly patterns in power consumption data, or signals in audio/signal processing. Conversely, for time-series where aperiodic and stochastic noise or local patterns are significantly more important than periodicity (e.g., stock market fluctuation data), DAF might need additional evaluations to accurately gauge the performance.

---

### Official Review · Reviewer_CeYR · 2025-11-02

**Soundness:** 3
**Presentation:** 3
**Contribution:** 3
**Rating:** 8
**Confidence:** 4

**Summary:**

The paper proposes the decomposed attention FredFormer (DAF), a transformer (FredFormer)-based architecture that replaces DFT by  real fast Fourier transform and Tensorized MHA (TMHA) using Tensor Train Decomposition for efficient compression. DAF eliminates patching and layer normalization to better model the periodic nature of satellite orbits, significantly reducing parameters while maintaining strong generalization. Trained on a large-scale Starlink satellite dataset. DAF is evaluated on seven cross-domain and three real-world satellite datasets.

**Strengths:**

The paper targets operational challenges in orbit prediction e.g., long-term forecasting, limited computational resources, and model scalability across satellites making it valuable for real-world deployment.
The authors applied robust preprocessing, ablation studies showing the effects of RFFT, positional embedding, and tensorization and comparison with multiple baselines.
It also demonstrates how parameter-efficient transformer architectures can be adapted for safety-critical, data-sparse scientific domains.

**Weaknesses:**

All satellites share similar orbital mechanics and control schemes. So, zero-shot generalization to other constellations might be overstated. Did you include satellites with distinct perturbation dynamics? If no, what can you say about the adaptability.

It is not clear what is the inference latency? Whether it can be used on low-resource satellite onboard systems like CubeSats, etc.

The model is benchmarked mainly on mean error metrics (MSE/MAE), is this sufficient to evaluate or evaluation should include orbital physics consistency metrics?

Paper does not include potential failure cases.

Section 3.2 and 3.1 presenting the same stuff.

**Questions:**

same as limitations

---

> ### Author Response · Authors · 2025-11-28
>
> ## **W1: Generalizability and Adaptability to Distinct Dynamics**
> We thank the reviewer for the constructive comments. We agree that satellites generally share similar orbital mechanics. However, we clarify that in practice, their dynamical behaviors differ depending on the operating organization, mission objectives, hardware specifications, and operational strategies. We have organized our response to clarify the necessity of generalization and address the concerns regarding perturbation dynamics.
>
> **1. Clarification on Generalization**
> Even when satellites follow similar control schemes, their orbital dynamics operate across vast spatial scales. Consequently, practical application requires a model that can adapt to these variations without retraining. As shown in **Table 3**, zero-shot generalization is essential to ensure reliable operational performance across diverse satellite systems in real-world scenarios.
>
> **2. Distinct Perturbation Dynamics**
> Regarding the adaptability to distinct dynamics, we included constellations operating in different regimes:
> * **Starlink:** Operates at approximately **550 km** altitude [1].
> * **KINEIS:** Operates primarily around **650 km** altitude [2].
>
> The successful performance in the zero-shot setting, where the model was trained on Starlink and tested on KINEIS, demonstrates the model's capability to generalize across different orbital altitudes and constellation characteristics.
> ***
> REFERENCES
> [1] SpaceX Starlink Satellites: https://www.space.com/spacex-starlink-satellites.html
> [2] Kineis Constellation: https://kineis.com/en/satellite-constellation
> ***
>
> ## **W2: Inference Latency and Onboard Applicability**
>
> We thank the reviewer for the very interesting question. Indeed, onboard processing in CubeSat is an interesting scenario. However, the onboard scenario is not the current scope of our work. Our proposed work is designed for ground stations for real-time and low-latency data management and analysis systems.  In particular, we show that the processing time of our approach is minimal and sufficient for real-time applications **(Table 8)**. Nevertheless, we plan to actively consider further optimized models for low-resource environments as part of our future research.

---

> > ### Author Response · Authors · 2025-11-28
> >
> > ## **W3: Evaluation Metrics and Physical Consistency**
> > We thank the reviewer for the valuable feedback and helpful questions. In this study, we primarily employed MSE and MAE as our quantitative benchmarks. These metrics are not only the standard for general time-series forecasting but are also widely adopted in the aerospace and satellite orbit prediction literature to evaluate model accuracy [1-6].
> >
> > However, we agree that evaluating performance in terms of physical metrics provides greater insight into the mode's applicability to real-world aerospace missions. Regarding orbital physics consistency, calculating metrics such as kinetic energy requires specific information that is currently unavailable in our study. We believe that obtaining this data and incorporating these metrics in future work will lead to more robust research. Nevertheless, to address the need for physical evaluation, we conducted an additional analysis of *de-normalized 3D position errors in kilometers*, comparing our proposed model against the high-performing baselines identified in our Main Results. The table below summarizes the *RMS, Mean, Median, and Standard Deviation (Std)* of the position errors for each model:
> > | Model | RMS (km) | Mean (km) | Median (km) | Std (km) |
> > | :--- | :---: | :---: | :---: | :---: |
> > | **TimesFM** | 241.41 | 205.44 | 189.22 | 126.45 |
> > | **GRU** | 327.41 | 268.19 | 231.99 | 187.51 |
> > | **DLinear** | 342.24 | 283.90 | 239.06 | 191.12 |
> > | **SparseTSF** | 445.77 | 410.50 | 434.94 | 173.77 |
> > | **DAF (Ours)** | **216.48** | **186.41** | **172.53** | **109.98** |
> >
> > As demonstrated, our proposed method (DAF) consistently achieves the lowest error rates across all metrics, confirming its superior performance not only in normalized statistical metrics but also in terms of actual physical position accuracy.
> > ***
> > REFERENCES
> > [1] Li, Bin, et al. "A machine learning-based approach for improved orbit predictions of LEO space debris with sparse tracking data from a single station." IEEE Transactions on Aerospace and Electronic Systems 56.6 (2020): 4253-4268.
> > [2] Peng, Hao, and Xiaoli Bai. "Comparative evaluation of three machine learning algorithms on improving orbit prediction accuracy." Astrodynamics 3.4 (2019): 325-343.
> > [3] Peng, Hao, and Xiaoli Bai. "Artificial neural network–based machine learning approach to improve orbit prediction accuracy." Journal of spacecraft and rockets 55.5 (2018): 1248-1260.
> > [4] Peng, Hao, and Xiaoli Bai. "Exploring capability of support vector machine for improving satellite orbit prediction accuracy." Journal of aerospace information systems 15.6 (2018): 366-381.
> > [5] Peng, Hao, and Xiaoli Bai. "Machine learning approach to improve satellite orbit prediction accuracy using publicly available data." The Journal of the astronautical sciences 67.2 (2020): 762-793.
> > [6] Lin, Chusen, and Chen Junyu, “Using long short-term memory neural network for satellite orbit prediction based on two-line element data.” IEEE Transactions on Aerospace and Electronic Systems (2025).
> > ***
> > ## **W4: Discussion on Potential Failure Cases**
> > We appreciate the reviewer's suggestion to clarify the potential failure cases. We have updated **Section 5 (Lines 477-482)**, to explicitly address scenarios related to sudden maneuvers and environmental changes. The details are as follows:
> > * **Sudden Orbital Maneuvers (Added):** We have clarified that the model may struggle to predict sudden orbit changes, such as collision avoidance maneuvers, if such events are absent from the training data.
> > * **Extreme Environmental Variations (Added):** We added a discussion on how severe fluctuations in the space environment (e.g., rapid changes in atmospheric density due to intense solar activity) can create out-of-distribution scenarios, potentially increasing prediction errors.
> >
> > Satellite system failures are also an important scenario. We agree on its significance and recognize it as a promising avenue for future work. This is because obtaining failed satellite orbit data operationally is currently challenging, and such analysis aligns more closely with anomaly detection research, which extends beyond the scope of our current orbit prediction task.
> > ***
> > ## **W5: Structural Refinement of Sections 3.1 and 3.2**
> > We sincerely appreciate the reviewer's detailed assessment and the constructive points raised. Originally, **Sections 3.1 and 3.2** independently introduced DAF, RFFT, and IRFFT, which led to some repetition. Following your suggestion, rather than merging the sections, we have streamlined **Sections 3.1 and 3.2** by removing redundant descriptions to improve readability and clarity. The revised content can now be found in the updated manuscript.

---

### Meta-Review · Area_Chair_CPrg · 2026-01-01

**Summary:**

This paper presents a large time series forecasting model for satellite orbit prediction. Given the significant divergence among reviewer opinions, I have conducted an independent assessment. My main concerns are outlined below:

1. Although the title indicates a focus on satellite orbit prediction, the proposed method appears to be a general time series model without specific adaptations for this domain. Moreover, no experimental results outside satellite orbit prediction are provided to demonstrate its broader applicability.

2. All experiments are conducted solely within the narrow domain of satellite orbit prediction, which raises the question of what is meant by “cross-domain” capability as mentioned in the paper.

3. The methodological novelty of the paper—a key criterion for this conference—is limited. While I acknowledge the practical importance of satellite orbit prediction, the core technical contribution does not appear to advance the state of the art in time series forecasting.

**Reviewer Concerns:**

While the author’s rebuttal provided a comprehensive response to several issues raised by the reviewer, a number of substantive concerns remain outstanding.

1 (Reviewer CeYR):  All satellites share similar orbital mechanics and control schemes. So, zero-shot generalization to other constellations might be overstated.

2 (Reviewer Zongzhe Xu): Do you expect the same advantages to hold for general time series, and under what conditions?

3 ( Reviewer qjRj): Limited theoretical insight. While the paper presents clear empirical gains, it lacks a deeper theoretical analysis explaining why tensor decomposition and real-valued Fourier transforms improve performance. The benefits appear primarily empirical, without discussion on the induced inductive biases or representational properties.

4 ( Reviewer qjRj): Incremental methodological novelty. The proposed components, such as RFFT substitution, LayerNorm removal, and tensorized attention, are all established techniques. Their combination is well-engineered but not conceptually groundbreaking, making the contribution more engineering-oriented than methodological.

**Reviewer Scores:**

NA

---

### Decision · Program_Chairs · 2026-01-26

Reject